

# GRiMeDB: The global river database of methane concentrations and fluxes

Emily H. Stanley[1], Luke C. Loken[2], Nora J. Casson[3], Samantha K. Oliver[2], Ryan A. Sponseller[4] Marcus B. Wallin[5] Liwei Zhang[6] Gerard Rocher-Ros[7,8]

[1]Center for Limnology, University of Wisconsin-Madison, Madison WI, USA 53706
[2]U.S. Geological Survey, Upper Midwest Science Center, Madison, WI USA 53726
[3]Department of Geography, University of Winnipeg, Winnipeg, MB Canada R3B 2E9
[4]Department of Ecology and Environmental Science, Umeå University, Umeå, Sweden 90736
[5]Department of Aquatic Sciences and Assessment, Swedish University of Agricultural Sciences, Uppsala, 75007, Sweden
[6]Sino-French Institute for Earth System Science, College of Urban and Environmental Sciences, Peking University, 100871, Beijing, China
[7]Department of Forest Ecology and Management, Swedish University of Agricultural Sciences, 90183 Umeå, Sweden
[8]Integrative Freshwater Ecology Group, Centre for Advanced Studies of Blanes (CEAB-CSIC), 17300 Girona, Spain

*Correspondence to*: Emily H. Stanley (ehstanley@wisc.edu)

**Abstract.** Despite their small spatial extent, fluvial ecosystems play a significant role in processing and transporting carbon in aquatic networks, which results in substantial emission of methane ($CH_4$) to the atmosphere. For this reason, considerable

effort has been put into identifying patterns and drivers of $CH_4$ concentrations in streams and rivers and estimating fluxes to the atmosphere across broad spatial scales. Yet progress toward these ends has been slow because of pronounced spatial and temporal variability of lotic $CH_4$ concentrations and fluxes and by limited data availability across diverse habitats and physicochemical conditions. To address these challenges, we present the first comprehensive database of $CH_4$ concentrations and fluxes for fluvial ecosystems along with broadly relevant and concurrent physical and chemical data. The Global River

Methane database (GriMeDB; https://doi.org/10.6073/pasta/b7d1fba4f9a3e365c9861ac3b58b4a90) includes 24,024 records of $CH_4$ concentration and 8,205 flux measurements from 5,037 unique sites that were extracted from publications, reports, data repositories, and other outlets published between 1973 and 2021. GriMeDB also includes 17,655 and 8,409 concurrent measurements of concentrations and 4,444 and 1,521 of fluxes for $CO_2$ and nitrous oxide ($N_2O$) respectively. Most observations are date-specific (i.e., not site averages) and many are supported by data for 12 physicochemical variables and 6

site variables. Site variables include codes to characterize marginal channel types (e.g., springs, ditches) and/or presence of human disturbance (e.g., point source inputs, upstream dams). Overall, observations in GRiMeDB encompass a broad range of the climatic, biological, and physical conditions that occur among world river basins, although some geographic gaps remain (e.g., arid regions, tropical regions, high latitudes and altitude systems). The global median $CH_4$ concentration (0.20 μmol L$^{-1}$) and diffusive flux (0.44 mmol m$^{-2}$ d$^{-1}$) in GRiMeDB are lower than estimates from past, site-averaged compilations, although

ranges and standard deviations are greater from this larger and more temporally-resolved database. Available flux data are dominated by diffusive measurements despite the recognized importance of ebullitive and plant-mediated $CH_4$ fluxes. Despite these limitations, GriMeDB provides a comprehensive and cohesive resource for examining relationships between $CH_4$ and environmental drivers, estimating the contribution of fluvial ecosystems to $CH_4$ emissions, and to contextualize site-based investigations.

## 1 Introduction

Despite their small areal extent, running-water (fluvial) ecosystems play a significant role in processing and transporting carbon (C) in and through aquatic networks, including the production, consumption, transport, and evasion of carbon dioxide ($CO_2$) and methane ($CH_4$). The profound planetary warming effects of $CH_4$ in the atmosphere, its erratic but accelerating rate

of increase over recent years (NOAA, 2022), the significant contributions of natural sources to the growing atmospheric pool (Turner et al., 2019), and improvements in gas measurement technologies have all contributed to a rapid increase in studies of $CH_4$ dynamics in aquatic environments in general, and fluvial ecosystems in particular. These studies reveal widespread supersaturation of $CH_4$ in running waters that underlies their larger than expected contribution to the atmospheric pool (Stanley et al., 2016).

Efforts to quantify fluvial $CH_4$ dynamics at regional, continental, and global scales have been fraught with uncertainty, reflecting the inherent variability of this gas in surface waters combined with a notable limitation in data availability. Sources and sinks of $CH_4$ are often unevenly distributed over space and time within drainage systems and, as a result, concentrations can vary over 3-4 orders of magnitude over short time periods or relatively small spatial extents (e.g., Anthony et al., 2012; Crawford et al., 2017; Bretz et al., 2021; Robison et al., 2021). Similarly, several drivers or predictors of $CH_4$ have been

identified in the literature and these properties also have variable spatial and temporal distributions. Thus, efforts to estimate the total emissions from world rivers have relied on relatively small data sets composed of site-specific values that have been averaged over time, and then have employed upscaling strategies based on Monte Carlo techniques or extrapolations using predictor variables that have little or no significant statistical relationships with large-scale patterns of gas concentrations or fluxes (Hutchins et al. 2020). Consequently, current global scale estimates of riverine emissions are poorly constrained and

highly uncertain (Saunois et al., 2020; Rosentreter et al., 2021)

The combination of rapidly increasing atmospheric concentrations of $CH_4$, the significant role of fluvial systems in emitting this gas, and, critically, current difficulties in explaining or predicting concentrations and fluxes with reasonable certainty inspired the central goal of this paper: to assemble a comprehensive database of $CH_4$ concentrations and fluxes for fluvial ecosystems that includes broadly relevant concurrent physical and chemical data. This effort expands upon a prior compilation

of $CH_4$ and $CO_2$ data (MethDB; Stanley et al., 2015) that was constructed to emphasize among-site differences and included 1,496 concentration records and 532 flux records from 1,080 sites. In this more comprehensive Global River Methane database (GRiMeDB), most data are date-specific (i.e., not averaged over time), the breadth of site types is expanded to include marginal fluvial habitats as well as disturbed and artificial waterways, and $CH_4$ data are supported by a broad suite of site-specific physical and chemical attributes along with concurrent measurements of $CO_2$ and $N_2O$ where available. Given the more finely

resolved scale of the data and the growth of the field in the past decade, GRiMeDB represents a significant expansion beyond MethDB. Building GRiMeDB with greater detail and breadth of data was done with the intent of increasing opportunities to identify and predict spatial and temporal variation in $CH_4$, to test hypotheses related to greenhouse gas dynamics, and reduce uncertainty in future upscaled estimates of gas emissions. In this paper, we (1) provide a detailed description of the components of the database and its construction; (2) summarize some basic patterns of gas concentrations and fluxes from GRiMeDB; and

(3) highlight critical data gaps and possible future research opportunities for improving current understanding of $CH_4$ dynamics in streams and rivers.

## 2 Database components and assembly

GRiMeDB is composed of four tables that contain information related to (1) data sources, (2) sites, (3) gas concentrations and supporting physicochemical data, and (4) gas fluxes. All tables are linked by unique data source identifiers, and all concentration

and flux observations are also linked to unique site numbers. Data included in GRiMeDB were gathered from scientific journals,

government reports, public data repositories, theses, dissertations, and unpublished data sets provided by individual investigators. Sources were discovered via searches of bibliographic databases and data repositories (Web of Science, Google Scholar, ProQuest Dissertations & Theses Global, China National Knowledge Infrastructure, Environmental Data Initiative, USGS ScienceBase, NERC Environmental Information Data Centre, Arctic Data Center, and PANGAEA) using the keywords: methane and stream*

or river* or ditch* or canal*, and searches were repeated numerous times between 2018 and December 2021 for completeness. We also used informal 'word of mouth' approaches to discover additional, often unpublished data sets.

All potential data sources were first screened to determine their appropriateness for inclusion in GRiMeDB. Several criteria were established *a priori* to ensure the usability of the data and that it was derived from inland running water systems. Coastal sites with

>1 ppt salinity were considered estuarine and thus were excluded. Similarly, sites that were situated in reservoirs, dam spillways, or lake outlets or were subject to experimental manipulation were omitted. We did not enter fluxes derived from chambers attached to collars or inserted into sediments because we could not be certain that such measurements were capturing air-water fluxes. Sources that reported gas concentrations or fluxes only as ranges (i.e., minimum and maximum values only) were not included. Finally, rates expressed on an annual basis were also excluded to avoid introducing uncertainty associated with different upscaling

assumptions and methods.

### 2.1 Sources Table

The Sources Table contains the list of all sources used to build GRiMeDB, a unique identification number (*Source_ID*) for each $CH_4$ data source, and basic bibliographic information for the data source (*Title, Author, Source*, publication year [*Pub_year*], and digital object identifiers [DOI] or other persistent identifier; all column titles for this table are defined in Table

A1). In several cases, data sources were supplemented with additional supporting information (e.g., associated physicochemical data) from separate sources (described further in Sect. 2.3) or additional or corrected information from authors (Fig. 1). In the latter case, we contacted authors if questions arose regarding their data (e.g., clarification regarding units) and/or to request supporting information or site- or date-specific concentrations or fluxes if published values were aggregated. Inclusion of additional unpublished data from authors is noted in the Sources Table along with a description of the addition or



correction. If supporting data from separate published sources were used, the DOI or other persistent identifier for the secondary source was listed in a separate column.

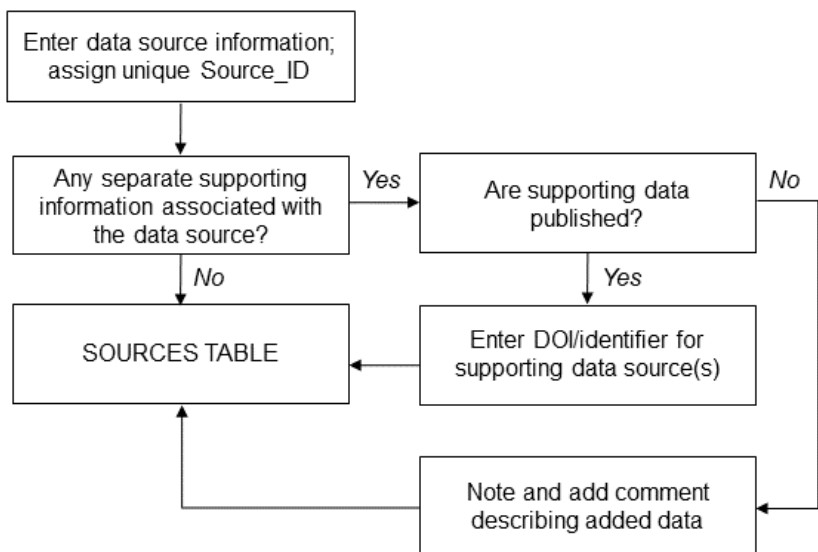

**Figure 1: Workflow for entering data into the Sources Table of GRiMeDB.**

## 2.2 Site Table

The Sites Table reports basic information on attributes for all sites where $CH_4$ was sampled. Each site has a unique identification code (*Site_ID*) and name (usually taken directly from the data source) and is linked to the Sources Table via the *Source_ID* (see Table A2 for detailed descriptions of all columns in the Sites Table). What composes a site varies among data sources and includes discrete sampling points, discrete study reaches, and aggregations of points and/or reaches across larger areas such as a drainage

basin. Because gas data for sites in this third category are averages from locations that may vary with respect to land use, channel order, slope, etc., we included fields to indicate if a site was aggregated and if so, the number of locations in the aggregation (if available). We also limited the resolution of latitude and longitude for these sites to < 3 decimal places. At the opposite extreme, gas sampling at points very close to one another (a 'high density site' *sensu* Fig. 2) has the potential to create ambiguities for site delineation and data analysis. To avoid these pitfalls, we combined points with slightly different latitude-longitude values to

represent a single site for three specific cases. First, multiple samples collected at different points and/or depths within a channel cross-section were averaged to form a single site. Second, some drainages or regions were surveyed repeatedly (particularly the Congo River basin and streams in Pennsylvania, USA) and it was not always clear if closely situated points from different surveys were intended to be a repeated sampling of the same location or sampling of discrete sites. Some judgment was involved in choosing between these two possibilities, and in a subset of cases, points in close proximity to one another that were sampled on separate

dates were treated as a single site. Finally, three data sources had extremely high sampling densities within discrete reaches (50 -

>20,000 samples per reach; Crawford et al., 2016; Call et al., 2018; Loken et al., 2018). Because closely adjacent gas samples can be spatially autocorrelated (Crawford et al., 2017) and including all individual values from these studies would have resulted in their over-representation in the database, individual point measurements were treated as within-reach replicates.

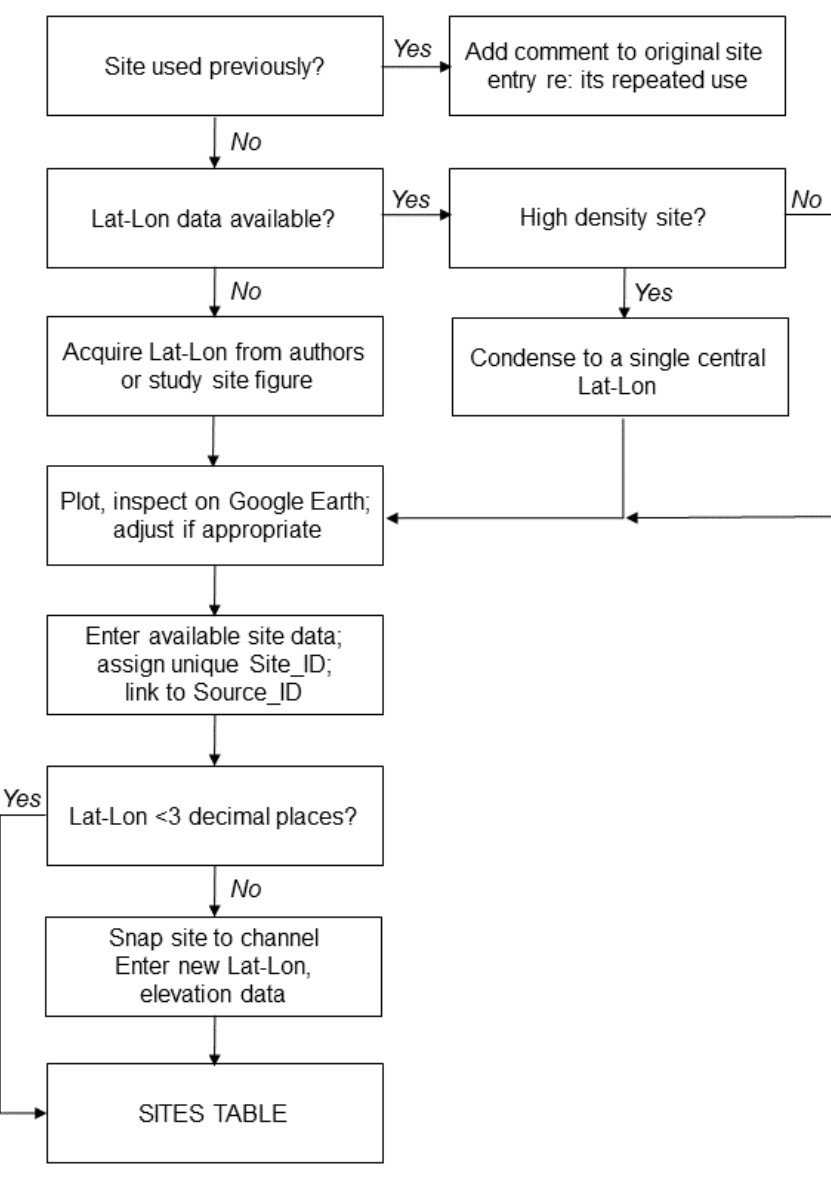

**Figure 2: Workflow for entering and checking data for the GRiMeDB Sites Table. 'Lat-Lon' is an abbreviation for latitude and longitude.**

For a site used in multiple studies, the *Site_ID* was assigned to the earliest paper and a comment was added to the site entry noting its use in other data sources (Fig. 2). Latitude and longitude coordinates were available for most sites; however, in



several cases, location information was acquired from authors or estimated from study site figures using Google Earth (© Google Earth 2020). Sites were plotted on Google Earth and inspected (Fig. 2) to identify and correct data errors. If a site's
coordinates were immediately adjacent to, but not on a channel, the coordinates were adjusted to fall on the channel and this modification was noted in the *Comments* field. If available, additional variables drawn from the data sources were entered to characterize the site, including stream name, basin or region name, elevation, channel slope, Strahler order, basin area, and codes denoting distinct channel or site types (described below). We also estimated elevation for all sites except aggregated sites or sites with poorly-resolved coordinates (less than 3 decimal places for both latitude and longitude) after snapping
coordinates to the nearest stream channel using the following procedure. First, we downloaded a digital elevation model (DEM) for each site using the function *get_elev_raster()* from the package "elevatr" (version 0.4.2; Hollister et al., 2021) at a resolution of 6-9 m depending on the location in the globe. Second, the DEM was processed for hydrological correctness using the package "whitebox" (version 1.2.0, Wu, 2020) by filling single cell pits (*fill_single_cell_pits()* function) and breaching depressions (*breach_depressions()* function) to obtain a flow direction model (*d8_pointer()* function). Finally, we calculated
a flow accumulation model (*d8_flow accumulation()* function). If the coordinates reported in the data source had a flow accumulation less than 10 cells (indicating that were not located in a preferential flow path), the new coordinates were assigned to the cell with the highest flow accumulation within a 50 m radius. If the initial site had a high flow accumulation value (>10 cells), we assumed the site was in a stream channel.

Many studies of $CH_4$ dynamics have been undertaken to determine if and how specific phenomena such as presence of upstream reservoirs, point source discharges, thermokarst features, or oil and gas extraction potentially affect fluvial $CH_4$ (and other constituents), usually with an expectation of a net enhancement of concentrations and fluxes. Similarly, other studies have examined sites that may be expected to be enriched in $CH_4$, but whose fluvial identity might be considered marginal or ambiguous (e.g., springs, floodplain backwaters, ditches, canals). Inclusion of such 'methane hunting' studies has the potential to bias the dataset
toward higher values (Stanley et al., 2016). Nonetheless we included these studies in GRiMeDB because they provide an opportunity to investigate the consequences of human activity and gain a more comprehensive understanding of fluvial $CH_4$ dynamics (e.g., see Alshboul et al., 2016; Peacock et al., 2021). However, to accommodate future analyses in which use of such data might be unsuitable, or alternatively, when these sites might be the sole focus of a study, we generated a set of channel codes to identify targeted site types (Table 1). Information about four of the codes was not consistently available among data sources and
thus their assignment often involved judgment calls. The first case involved determining if the presence of an upstream dam (code DD) was or was not relevant for sites of varying downstream distances. We used a value of distance of 7 km as a cut-off for this category, although the zone of influence of small or large dams may be far shorter or extend far beyond this distance (Kemenes et al., 2007), respectively. To provide some context for this code, a site's distance from a dam was acquired from the data source or estimated in Google Earth using the Path tool and reported in the *Comments* field whenever possible. The second case involved
straight, symmetrical channels that are common in many agricultural and urban areas. Frequently, it was not known if this unnatural geometry was due to channelization (straightening) of a stream (code CH) or creation of a new channel (ditches and canals; codes

DIT and CAN). In the absence of specific information, straight channels were classified as CH. Third, channels draining or passing through wetlands (WS) were often difficult to identify, particularly given seasonal variation in wetland appearance in tropical systems with wet-dry climates. Finally, floodplain channels presented a distinct challenge because of the complex nature of these

environments and their potential to be classified as either riverine or wetland systems. We used the FP code to indicate habitats that were described or appeared to be lentic (i.e., backwaters or connected floodplain lakes) but were persistently connected to the main river channel and thus were part of the fluvial system. Given these ambiguities, we recommend that these four codes be viewed and used with care.

## 2.3 Concentrations and Fluxes Tables

The Concentrations and Fluxes Tables contain the primary gas data central to GRiMeDB, and the Concentrations Table also hosts physical and chemical variables associated with concentration and/or flux observations (see Tables A3 and A4 for the full list of Concentrations Table and Fluxes Table columns and their descriptions). The vast majority of concentration and flux data were extracted from tables within data sources, data repositories, or provided by authors. However, in some cases, values were acquired

from figures using graphical digitizing software (WebPlotDigitizer (https://automeris.io/WebPlotDigitizer/), GetData (http://getdata-graph-digitizer.com/), or DigitizeIt (https://www.digitizeit.xyz/)). Plots with log scales or that were difficult to interpret were not digitized. The accuracy and consistency of this method were evaluated by comparing data generated by different individuals digitizing a set of common figures and by comparing digitized results to known results. Agreement both between both comparisons was strong (average slope = 0.994, average $R^2$ = 0.9996 for 5 comparisons between individuals digitizing the same

dataset, and average slope = 0.998, average $R^2$ =0.997 for digitized versus actual data for 7 datasets; see Table S1 for further details), demonstrating the reliability of this method of data gathering.

Whenever possible, concentrations and fluxes were entered as values for individual sites on individual days (i.e., not averaged across sites or days) (Fig. 3). Because 1 day represented the lowest level of temporal resolution in GRiMeDB, repeated

measurements made on a sub-daily scale were averaged and expressed as a daily value and were not considered to be aggregated over time. If multiple replicates were collected at different times on the same day (e.g., a study of diurnal gas dynamics), this was noted in the *Comments* fields and measurements prior to and after 12:00 a.m. (local time) were entered as separate, consecutive days. Observations resolved to the daily scale can be identified using either a "No" in the *Aggregated_Time* field or by having the same reported starting (*Date_start*) and ending (*Date_end*) dates. If the specific start and end dates were not specified in the data

source, we entered the day as the 15th of the month and noted this approximation in the *Comments* field. If available, we also reported minimum and maximum values and standard deviations (SD) for entries that were aggregated over space and/or time. SDs, but not minima and maxima were reported for replicates from non-aggregated sampling when available, except for reach-averaged entries with multiple within-reach measurements and diel studies with multiple within-day values. In these cases, minima and maxima were also included.






**Table 1. Codes denoting distinct site or channel attributes or presence of conditions that potentially affect CH₄ concentrations or fluxes. Assignment of codes to a site is based on information provided in the data source and/or physical appearance of a site and a site may have more than one code. Codes are reported in the *Channel_type* field of the Sites Table.**

| Code | Definition |
|------|------------|
| CAN | Canal or other artificial channel with hardened channel boundaries |
| CH | Channelized; a channel that has long straight-line sections and changes in channel direction are typically distinct angular features rather than curves |
| DC | Channel in a river delta |
| DD | Downstream (within 7 km) of a dam. Samples from spillways were not included. |
| DIT | Ditch, typically for agricultural drainage, without channel hardening |
| FP | Site in a floodplain water body connected to the main channel that appears lentic or is described as a floodplain lake or backwater. This category does not include braided river side-channels within floodplains or tributary channels transecting a floodplain |
| GT | Site below the toe or terminus of a glacier |
| IMP | Presence of multiple and typically small impoundments in a site's vicinity (e.g., various European rivers, Mississippi River) |
| PI | Permafrost influenced; this refers specifically to sites at or immediately below thermokarst outflows and not to sites in areas underlain with permafrost |
| PS | Immediately (<1 km) downstream of a point source discharge |
| SP | Spring channel; this does not include sites characterized as seeps (features with low flow volume adjacent to channels) |
| TH | Site receiving inputs of thermogenic CH₄, either naturally or as a result of mining, fracking, oil extraction, and other related activities. |
| WS | Wetland stream; site is in a wetland or immediately downstream from the outlet of a wetland |

Dealing with concentration data reported as a negative value, zero, or below a detection limit (BDL) is problematic because of inconsistencies in detection limits and reporting practices, and any decision about handling these records introduces some bias (Stow et al., 2018). For example, using a non-numerical format such as BDL or <0.01 is likely to lead to the elimination of these 205 entries during data analysis and thus would introduce a bias against low-value observations. Alternatively, converting any such value to zero would introduce a bias in the opposite direction. As a compromise solution, concentrations recorded as zero in the original data source were entered as zero in GRiMeDB and other below-detection values were entered as -999999. In this latter

case, the original data entry format was noted in the *Comments* column. For fluxes, negative and zero values were entered without modification or comment.


The Fluxes Table reports diffusive, ebullitive, and total $CH_4$ fluxes along with $CO_2$ and $N_2O$ diffusive fluxes. Given the diverse strategies for measuring each of the three $CH_4$ flux pathways and potential biases associated with different approaches (Lorke et al., 2015), values are accompanied by brief categorical descriptions of methods used for each $CH_4$ flux type as well as for $CO_2$ fluxes and the gas exchange coefficient $k$. For a small number of entries, $CH_4$ fluxes were not directly reported in the data source

but information was available (dissolved gas concentration, temperature, and a corresponding gas exchange coefficient ($k$)) that allowed us to calculate these fluxes. We also entered BDL values for flux for one data source in which fluxes had been calculated from concentration, but fluxes associated with BDL concentrations had been omitted from the results. Finally, a small number of observations listed diffusive and ebullitive but not total fluxes, so diffusion and ebullition were summed and entered as total flux. In all cases, the added calculations are noted in the *Comments* field.


The GRiMeDB Concentrations Table includes physicochemical measurements in support of concentration and flux observations (Fig. 3, Table A3). Availability of this supplemental information varied widely among data sources, and was limited to data collected concurrently with gas samples. For data sources with gas fluxes and physicochemical data but not gas concentrations, we created rows in the Concentrations Table to capture the supporting data. These records are identified by a "Yes" in the

*FluxYesNo* column, *SampleCount* = 0, and NA in the *CH4mean* column. Finally, water temperature was estimated for entries if it was needed to convert gas units and entered in the *WaterTemp_degC_estimated* column. Estimates were typically based on values from the same or adjacent sites or similar times (e.g., averages of temperature from the prior and subsequent dates, or from the same month in a prior year).

Following completion of all data entry, gas and physicochemical variables were converted to 'new' standard units (Tables A3, A4). The identities of the new and original units are included in both the Concentrations and Fluxes Tables for clarity. Elevation was used to estimate atmospheric pressure if needed for unit conversions. We used Henry's Law, water temperature, and atmospheric pressure to convert dissolved gas values reported in ppm, ppb, μatm, and % saturation (~13% of observations). For observations that reported gas values as percent saturation (<1%), we also used the global average $CH_4$, $CO_2$, and the $N_2O$

atmospheric concentrations from the NOAA Global Monitoring Laboratory (https://gml.noaa.gov/ccgg/) for the year 2013, which corresponds to median observation year in the database.

**Figure 3: Workflow for entering and checking data for the GRiMeDB Concentrations and Fluxes Tables.**

## 2.4 Assessment of representativeness

We assessed the representativeness of sites in GRiMeDB relative to the global distribution of biological, physical, and climatic properties following van den Hoogen et al. (2021). Briefly, we first assigned each site to a corresponding river reach in HydroSHEDS (Linke et al., 2019), which is a global hydrological network database that contains spatial data for a wide array





of hydrological, physiographical, climatic, land cover, geological, edaphic and anthropogenic variables for each river reach. HydroSHEDS thus provides a multidimensional characterization of global rivers that is well suited for assessing how representative GRiMeDB sites are in terms of key biophysical and anthropogenic features. After excluding non-numerical variables (e.g., biome) and variables with monthly values (e.g., monthly precipitation), we performed a principal component

analysis (PCA) on all HydroSHEDS subcatchments using all possible combinations of the 54 remaining HydroSHEDS variables. From this, we selected all principal components (PCs) needed to explain 90% of the variance in the PCA, which corresponded to 28 PCs and 378 possible bivariate combinations of these PCs. For each unique PC pair, we computed the convex hull of all sampled sites to determine the distribution of these sites relative to all global river subcatchments for the specified PCs (Fig. 4). Each HydroSHEDS subcatchment was then assigned a value of 1 or 0 if it fell within or outside the

convex hull, respectively. This process was repeated for each of the 378 possible PC combinations. To collapse this information, we calculated the fraction of cases that a given subcatchment fell within the convex hull for all PC combinations to obtain summary value ranging from 0 to 1. A subcatchment with a value of 1 for this index of "representativeness" means that it fell within the convex hull for 100% of the PC combinations, indicating that its overall characteristics are well captured in the database. It is important to note that this analysis only captures average catchment properties of relatively large river

reaches (average subcatchment area: 130 km$^2$). Given the strong local controls on $CH_4$ concentrations and fluxes, interpretations from this analysis should be made with some caution.

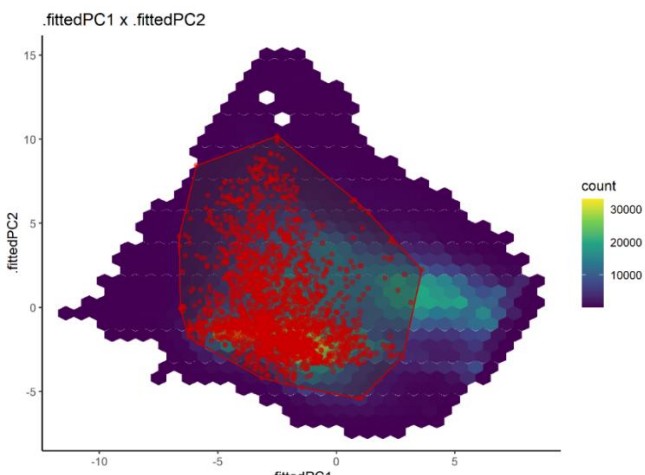

**Figure 4: Example of a representative PCA hexagon plot based on variability in HydroSHEDS river subcatchment attributes. Hexagon colour indicates the number of subcatchments per hexagon. Subcatchments hosting GRiMeDB sites are plotted in red and contained within the convex hull delineated by red lines. Subcatchments that fall within this polygon are assigned a value of 1 and those outside the perimeter are given a value of 0 to indicate the representativeness of sampled reaches for this pair of PC axes. See Sect. 2.4 for further explanation.**






## 2.5 Data checking and data analysis

Several approaches were taken to check the accuracy of data in GRiMeDB. This included evaluation of the reliability of digitized data (Sect. 2.3) along with several additional inspection steps. Entries were error checked by a co-author other than the individual who entered the data, including confirmation of site location information, validating units for all variables, and spot- or complete
checking of entered gas data (independent units and data check in Fig. 3), depending on dataset length and if data were manually entered or imported directly from a file. Once values had been converted to standard units, all variables were plotted to identify outliers (outlier check; Fig. 3), and extreme values were checked against the original data source. In cases in which errors were present in the original data, if possible, authors were contacted for clarification. In the few rare cases in which issues could not be resolved, the data were excluded. These and all other calculations were performed in R (version 4.2, R Core Team 2021), using
the "dplyr" package (version 1.0.7, Wickham et al., 2021) for data analysis, "sf" package (version, 1.0, Pebesma, 2018) for spatial data processing, and "ggplot2" (version 3.3.5, Wickham, 2016) and patchwork (Pedersen, 2020) packages for visualization.

## 3. RESULTS

### 3.1 Overview of GRiMeDB data

GRiMeDB includes 24,024 records of $CH_4$ concentration and 8,205 $CH_4$ flux values from 5,037 unique sites, along with 17,655 and 8,409 concurrent measurements of concentration and 4,444 and 1,521 of flux of $CO_2$ and $N_2O$, respectively (Table S2). Although the first concentration and flux values in GRiMeDB were published in 1973 (Lamontagne et al., 1973) and 1987 (de Angelis and Lilley, 1987), respectively, over 70% of all $CH_4$ concentrations and 80% of flux observations became available after 2015 (the year of publication of MethDB; Fig. 5). This growth in data availability has occurred predominantly along the
spatial axis, as almost two thirds of all sites were added in or after 2015 and over half of all sites in the database have a single concentration and/or flux observation. Conversely, long timeseries are rare, with only 8% of the 5,037 sites having > 10 concentration observations and 4% having >10 diffusive flux records (Fig. 5). The longest concentration record includes 590 observations distributed over 28 years (Toolik Inlet, *Site_ID* 9025; Kling, 2019a, 2022) while the longest flux record has 82 observations of diffusive flux over 4 years (*Site_ID* 3644; Aho et al. 2021). Further, among the 15 sites with time series > 5
years, 12 are situated in either the Toolik Lake region of Alaska, USA (Kling, 2019a, 2019b, 2022) or within the Krycklan watershed in Sweden (Wallin et al., 2018, Wallin, unpublished).



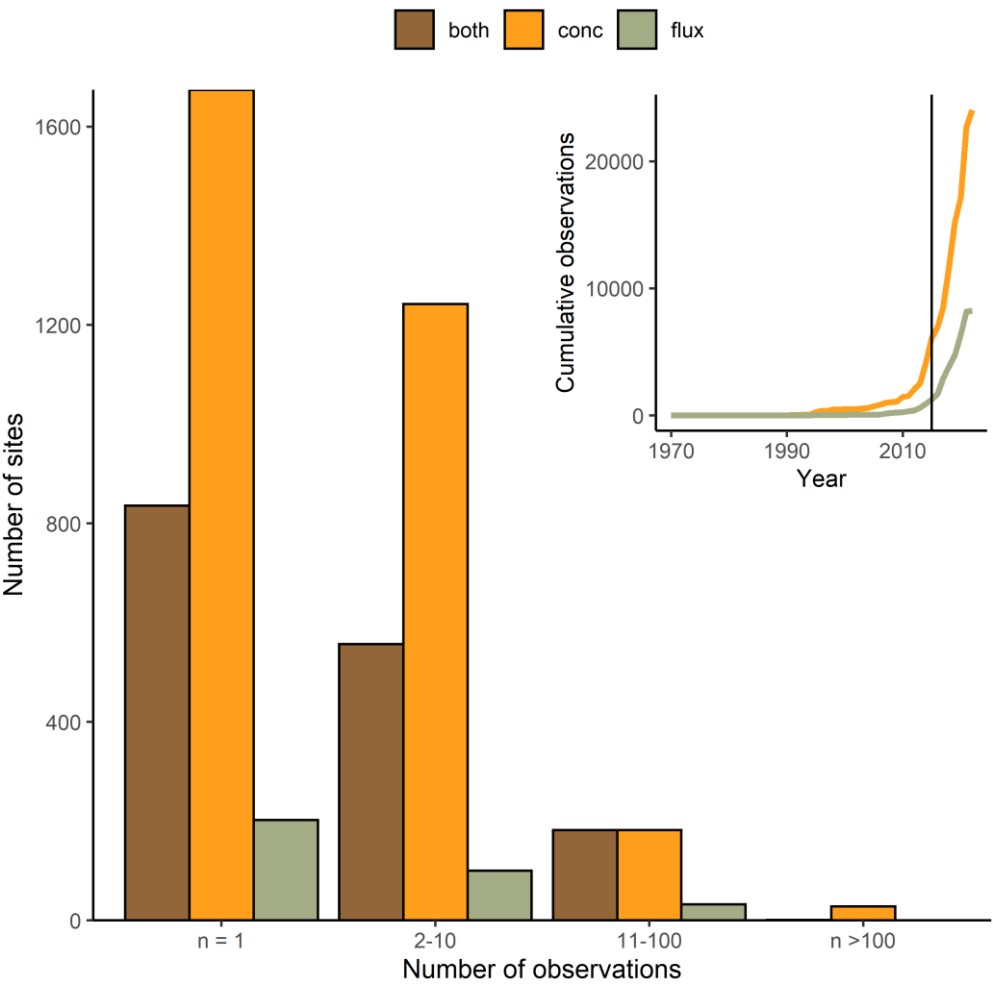

**Figure 5: Distribution of the number of observations per site. Brown bars indicate sites with both concentration and flux observations. Orange and green bars show sites with only concentration and only flux observations, respectively. Inset: Cumulative observations of $CH_4$ concentration and flux data based on the year of publication of the data source. The vertical line (2015) indicates the year of MethDB (Stanley et al., 2015) publication.**

## 3.2 Spatial and temporal distribution of data

Spatially, 40% of all sites and 52% of all $CH_4$ concentration observations are in North America, followed by Europe (25% of all sites and 26% of all $CH_4$ observations; Table S2). Conversely, there are vast geographic areas with moderate to high channel densities with few or no observations, such as central Canada, Central America, South America beyond the Amazon mainstem area, most of Russia, central and western Asia, New Zealand, and the Malay Archipelago (Fig. 6a). Despite these gaps, there is surprisingly good representation in terms of the range of hydrological, physiographical, climatic, land cover, geological, edaphic, and anthropogenic conditions that exist globally (Fig. 6b). Areas that are poorly represented are characterized by very



low channel density associated with arid or polar climates as well as high altitude regions (Greenland, northern Canada, northern Africa, central Australia, Middle Eastern nations, western China, Mongolia, Chile, southern Argentina). Evaluating the distribution or representativeness of sites in terms of system size is difficult given the limited availability of relevant information such as Strahler stream order or basin area, which were reported for only 26-28% of all sites (Table S2). For sites with these data, counts of observations decline with increasing stream order (Fig. 7) in a log-linear fashion ($R^2$ = 0.92 for

concentration and 0.90 for flux; $P$ <0.0005 for both regressions after excluding zero-order counts), consistent with Horton's Law of Stream Numbers (Horton, 1945). Thus, other than the extreme under-representation of zero-order channels, this predictable decline suggests reasonable representation by order, although this result should be interpreted with caution given the scarcity of relevant data. The distribution of counts by basin size follows a similar pattern of under-representation of sites draining very small basins and also indicates a potential over-representation of some large basin sizes (Fig. 7; e.g., basins of

ca. 10,000 km$^2$).

The distribution of observations among months illustrates seasonal sampling regimes dominated by summer sampling in northern (> 40˚) and southern (< -20˚) latitudes contrasted by even or erratic sampling at mid-latitudes (Fig. 8). Consistent with the lower representation of southern hemisphere rivers and streams, several months lack concentration and/or flux measurements south of -10˚ latitude, particularly during winter months. Beyond these gaps, the only months missing data in

the northern hemisphere are fluxes in January and February at sites north of 60˚ latitude and several missing months north of 70˚, presumably due to pervasive ice and snow cover.



**Figure 6: (a) Global distribution of observations in the database, colour coded for sites with concentration data only, flux data only, or both concentration and flux data. Top and right panels show, respectively, longitudinal and latitudinal patterns of the density of CH$_4$ observations (grey bars) and the density of river area (blue bars). These bars have been aggregated at a 1 latitudinal or longitudinal degree and rescaled from 0 to 1 for this visualization. River area was obtained from BasinAtlas (Linke et al. 2019). (b) Representativeness of the database based on a wide array of biological, physical, hydrological and land cover variables (see Sect. 2.4 for details). Values close to 1 indicate a high representativeness, with only 4% of the global surface below a threshold of 0.9.**







**Figure 7: Number of sites with concentration (top) or diffusive flux (bottom) observations as a function of stream order (left) and basin size (right) for the subset of sites with channel order and/or basin size information.**

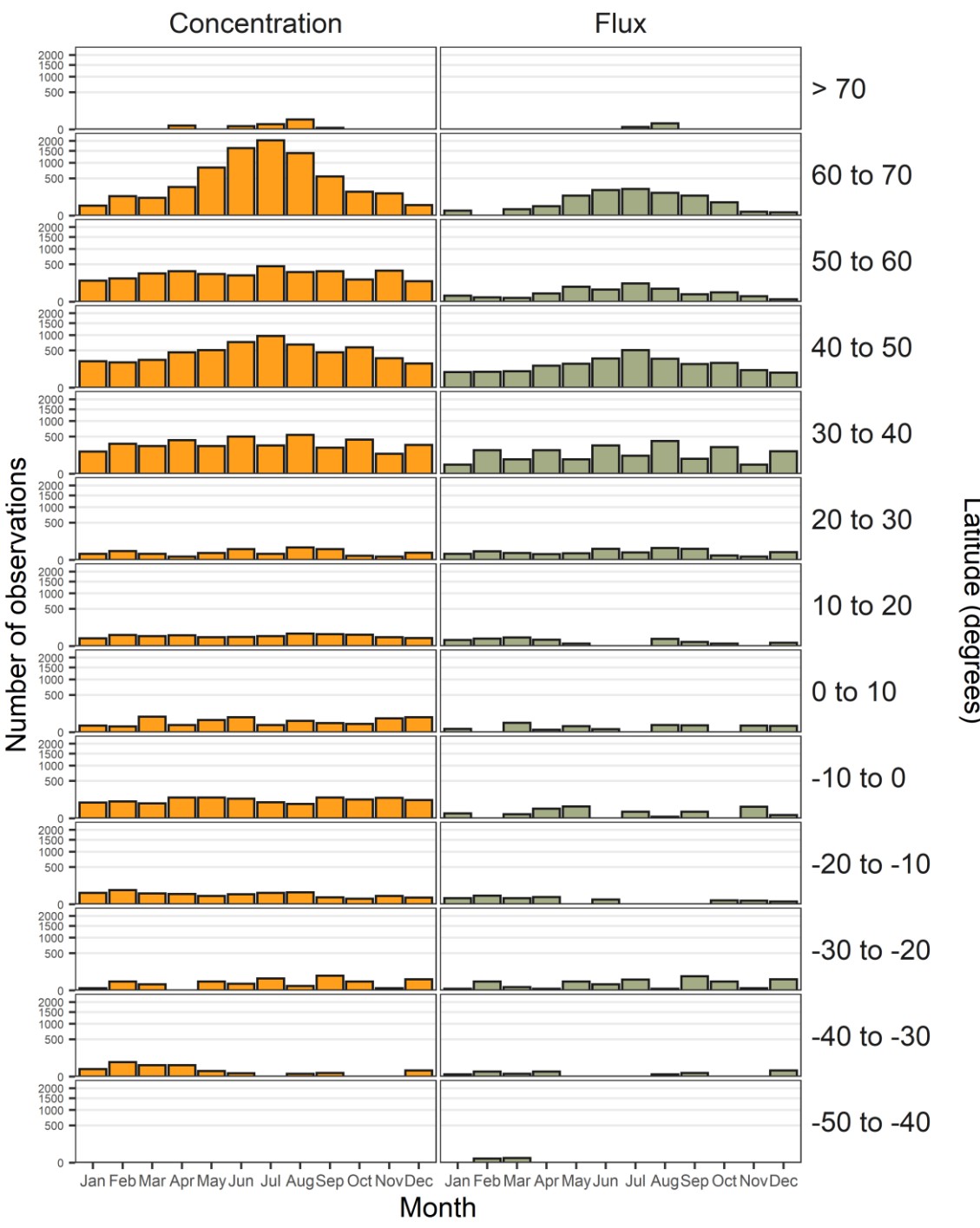

**Figure 8: Number of observations of concentration (left) and flux (right) by month for 10˚ latitude bands.**





### 3.3 CH$_4$ flux methodology

Records of CH$_4$ flux are dominated by diffusive flux measurements, which represent 85% of all flux values in the database,
with ebullition (8%) and total flux (7%) accounting for the remaining entries (Fig. 9). Not surprisingly, a variety of methods
have been used to quantify each flux type, although diffusive flux methods are dominated by calculations based on dissolved
gas concentration and a gas exchange coefficient ($k$) (74% of all observations), while chamber-based methods are most
common for quantifying total flux (93% of all observations). Similarly, the gas exchange coefficient $k$ is most commonly
estimated via physical models ($n = 3, 188$). Several models have been employed for this calculation, as indicated by >25
different references for $k$ model sources listed in GRiMeDB.

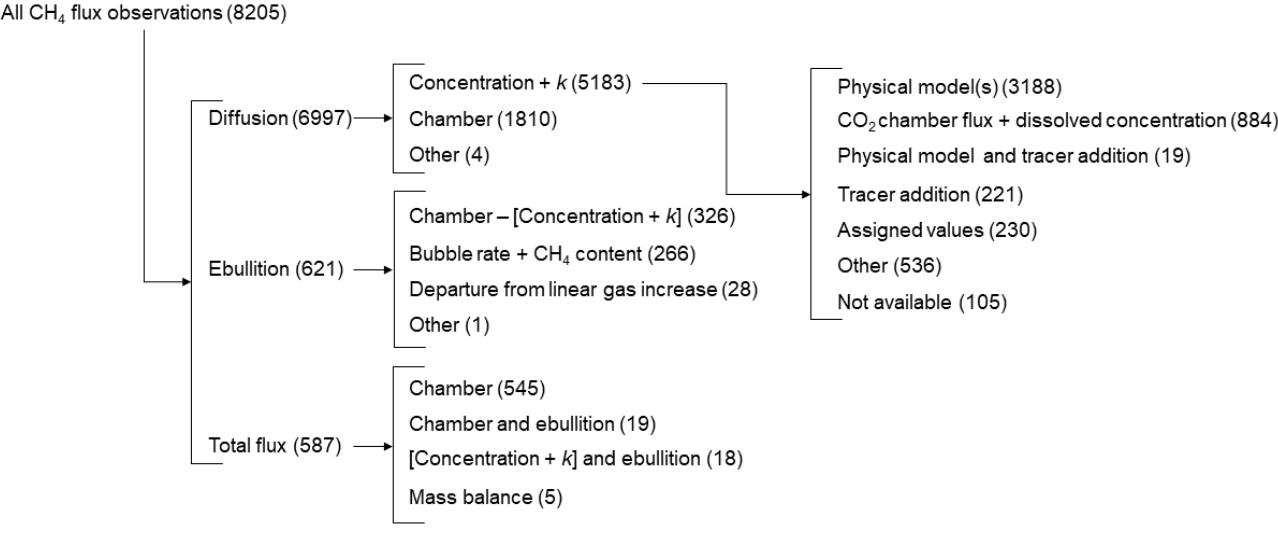

**Fig. 9: Counts of CH$_4$ flux observations by type (left), by major methodological categories for each pathway (middle), and for method
type used to estimate the gas exchange coefficient $k$ (right). For clarity, the chamber category includes all chamber types and patterns
of gas increase in the chamber unless specified; more resolved methodological data are presented in the GRiMeDB Fluxes Table.
See Table A4 for further details about category definitions.**

### 3.4 Overview of concentration and flux data

Concentrations and fluxes of all three gases are characterized by log-normal distributions that range across several orders of
magnitude (Fig. 10) and large coefficients of variation (CVs) for CH$_4$ and especially N$_2$O (Table 2). The vast majority (~95%)
of CH$_4$ and CO$_2$ concentrations appear to be supersaturated, in contrast to N$_2$O concentrations in which 67% of observations
were above this threshold. Reports of concentrations below detection are scarce for all gases, including N$_2$O (Table 2). For
fluxes,






**Figure 10: Histograms of gas concentrations and fluxes in GRiMeDB, excluding values reported as below detection or zero; counts of these values are reported in Table 3. Dashed vertical lines in the concentration histograms indicate the 100% saturation concentration based on the median estimated elevation (250 m) and water temperature for all sites and atmospheric concentrations of 1.83, 400, and 0.325 ppm for CH₄, CO₂, and N₂O, respectively.**




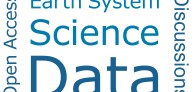

the fraction of observations with zero, below detection, or negative fluxes (5, 5, and 19% for diffusive $CH_4$, $CO_2$, and $N_2O$ fluxes, respectively). corresponded reasonably well with the frequency of subsaturated concentrations. At the other extreme, the highest $CH_4$ concentrations (> 200 µmol $L^{-1}$) paradoxically occur in either anthropogenically-influenced large rivers of the warm tropics (e.g., Amazon basin: Kemenes et al., 2007; Ganges, Mekong: Begum et al., 2021) or in small boreal headwater streams (e.g., Campeau et al., 2018; Wallin et al, 2018).

There were no meaningful univariate relationships between variables that may be used for upscaling (latitude, basin area, and stream order) and mean site concentration or flux (Fig. 11, Table S3). Although regressions were significant for latitude, latitude accounted for a very small percent of the variation in both concentration ($R^2 = 0.0025$) and flux ($R^2 = 0.004$) among sites. Similarly, concentration and flux among stream orders were significantly different for concentration (Kruskal-Wallis tests: Kruskal-Wallis $\chi^2 = 46.072$, $P < 0.001$) and marginally different for flux ($\chi^2 = 14.796$, $P = 0.06$). However, corrected pairwise comparisons (using the method of Benjamini and Hochberg, 1995) revealed no significant differences among orders for flux, and differences ($P < 0.05$) only between 7th order channels and all other orders, and between 6th vs 1st order sites for concentration, indicating an absence of a consistent change in $CH_4$ magnitude across channel orders. In contrast, variability decreased with increasing order and basin size, although this pattern is likely influenced by the accompanying decrease in sample size across this gradient.

**Table 2. Summary statistics for $CH_4$, $CO_2$, and $N_2O$ concentrations and fluxes. The %BDL column reports the percent of all observations that are below detection limits (including values reported as zero) for concentration. See Table S2 for counts and Table S3 for statistical summaries for all other variables.**

| Gas | Variable | Mean | Median | Max | Min | SD | CV | %BDL |
|---|---|---|---|---|---|---|---|---|
| Concentration (µmol $L^{-1}$) | | | | | | | | |
| | $CH_4$ | 1.49 | 0.20 | 456 | 0 | 10.69 | 718 | 3.2 |
| | $CO_2$ | 135 | 81.7 | 5,479 | 0 | 174.8 | 130 | 0.05 |
| | $N_2O$ | 0.058 | 0.017 | 32.9 | 0 | 0.602 | 1,042 | 0.59 |
| | | | | | | | | |
| Flux (mmol $m^{-2}$ $d^{-1}$) | | | | | | | | |
| | $CH_4$-diffusive | 7.31 | 0.44 | 4,057 | -136 | 86.4 | 1,182 | |
| | $CH_4$-ebullitive | 4.65 | 0.26 | 366 | 0 | 22.75 | 490 | |
| | $CH_4$- total | 7.62 | 0.62 | 366 | -0.05 | 28.5 | 375 | |
| | $CO_2$ | 319 | 128 | 23,749 | -1625 | 770 | 242 | |
| | $N_2O$ | 0.082 | 0.008 | 31.3 | -11.3 | 0.981 | 1,199 | |

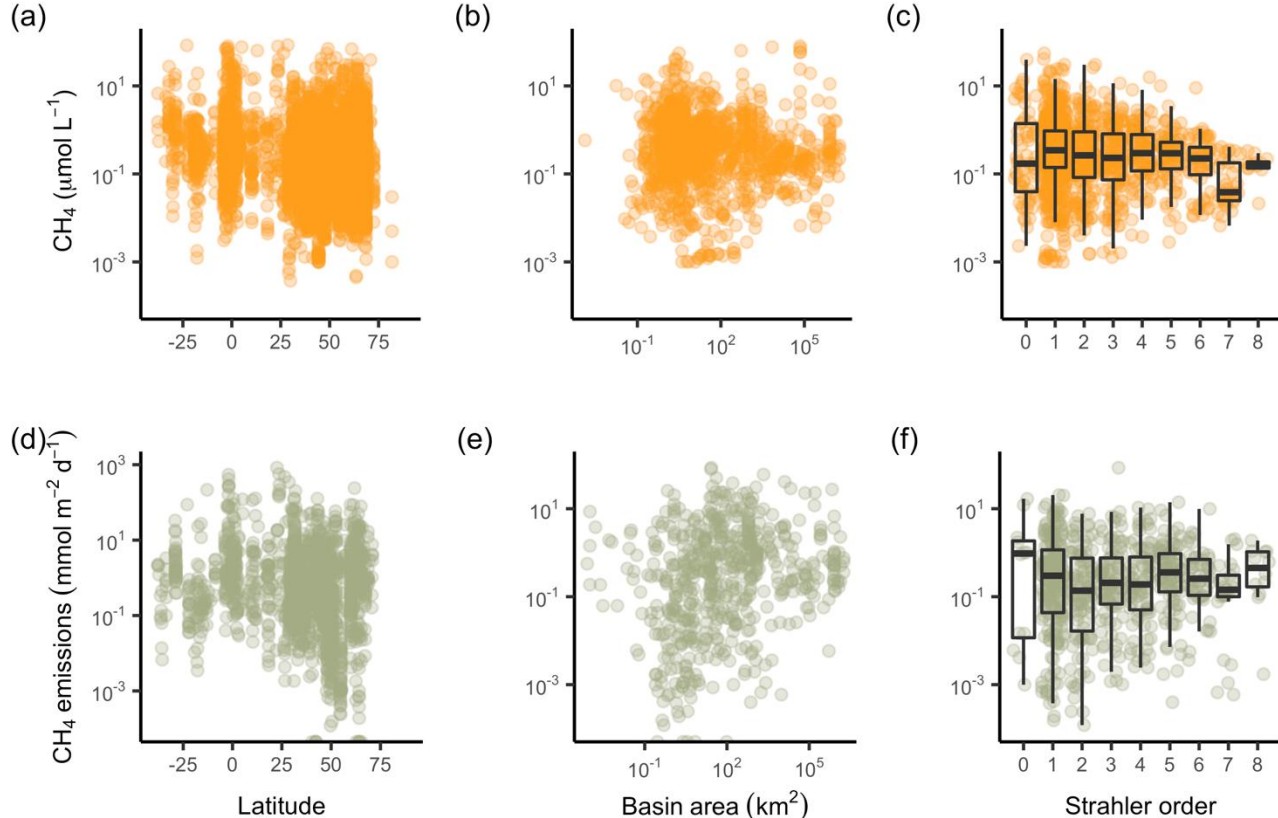

**Figure 11: Site-average CH$_4$ concentrations (a-c) and flux (d-f) as function of latitude, basin area, and Strahler stream order. For boxplots, the upper and lower edges of each box are the 25$^{th}$ and 75$^{th}$ percentiles and whiskers are drawn up to 1.5 times the interquartile range.**


As with relationships between CH$_4$ physical site attributes, relationships between CH$_4$ concentration or flux and water chemistry parameters are also characterized by substantial variability. Representative examples indicate increasing, decreasing, and ambiguous relationships between CH$_4$ concentrations and fluxes and selected chemical constituents (Fig. 12). One source of variation shown in Fig. 12 can be attributed to differences among sites, as is illustrated for the case of CH$_4$ concentration

versus discharge (Fig. 13). The cluster of all points in this plot by itself does not suggest an obvious linear relationship between concentration and discharge; however, resolving the data to the site level for sites with multiple observations reveals several significant trends. Among 57 sites with >30 observations, 42 had significant relationships ($P < 0.05$) between concentration and discharge and 30 of these 42 trends were negative.

Median site concentrations and fluxes for most categories of targeted channels (Fig. 14) were significantly different than "normal" (NORM) sites (Kruskal-Wallis test $\chi^2 = 460.1$, df = 12, $P < 0.0001$). Pairwise Wilcoxon comparisons adjusted to





account for multiple comparisons (Benjamini and Hochberg, 1995) indicated that springs (SP) and delta channels (DC) did not differ from NORM sites ($P > 0.4$) and IMP sites were marginally different ($P = 0.053$). Concentrations in channels at glacial termini (GT) and floodplain backwaters (FP) were lower ($P < 0.0001$), whereas all other site types had significantly

higher site average $CH_4$ concentrations than NORM sites. Fluxes among channel type were also significantly different (Kruskal-Wallis test $\chi^2 = 143.8$, df = 12, $P < 0.0001$), and similar to concentration, fluxes in delta channels, permafrost-influenced channels (PI), and springs were similar to NORM channels. Pairwise comparisons indicated that all other site types differed from NORM sites. Further, fluxes at floodplain sites were significantly higher ($P < 0.02$) than NORM sites, in contrast to the significantly lower concentrations for this site type. However, sample sizes were very small for FP, PI, as well as GT

sites (in addition to an absence of flux data for TH sites), so comparisons for these sites should be viewed very cautiously.

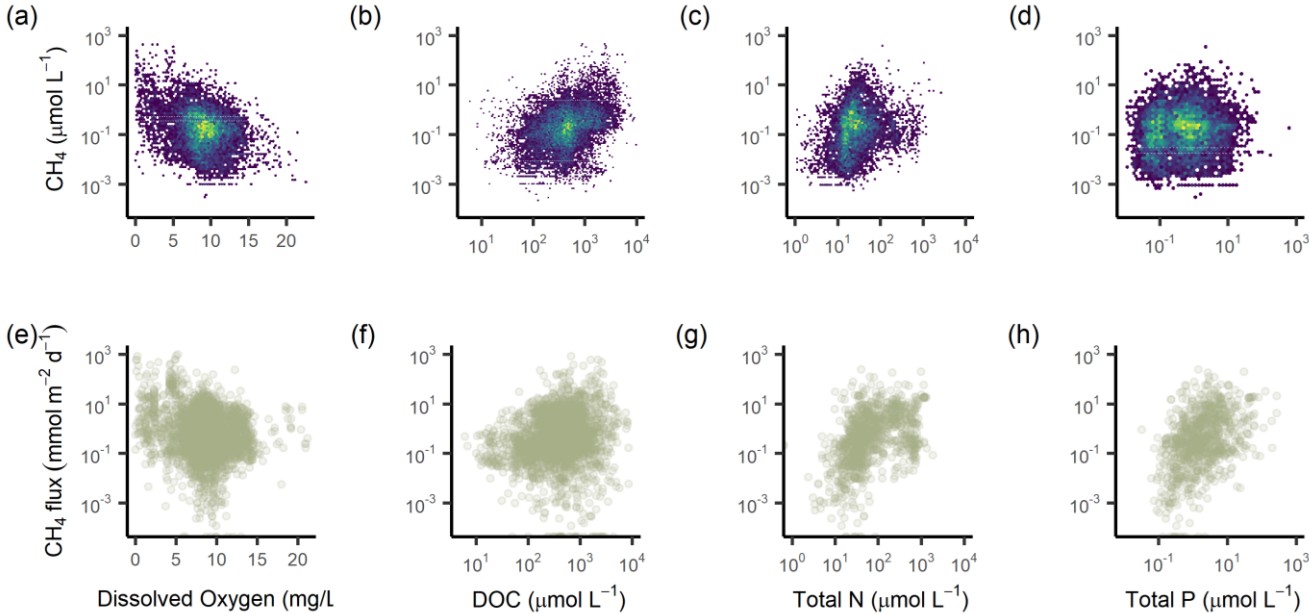

**Figure 12: CH₄ concentration (top row) and diffusive flux (bottom row) versus concurrent measures of dissolved oxygen ($n$ = 8,529 and 2,316 for concentration and flux, respectively), dissolved organic carbon (DOC; $n$ = 14,441 and 1,901), total nitrogen (Total N; $n$ = 8,378 and 467) and total phosphorus (Total P; $n$ = 6,904 and 240). Three outliers were excluded from the DOC plots, and because of the log scale for CH₄, negative and zero values have been omitted. For concentration plots, colours represent number of observations per polygon, varying from 1 (dark blue) to 30 (yellow).**


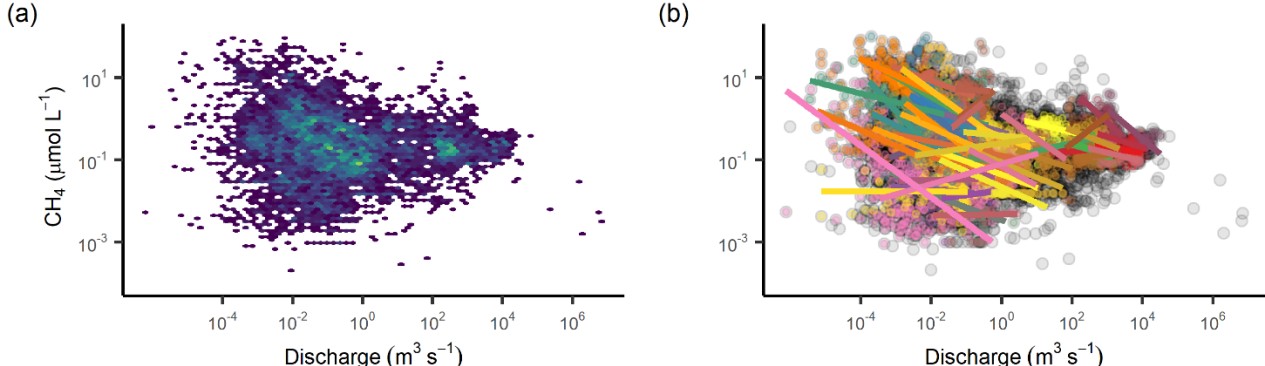

**Figure 13: CH₄ concentration versus concurrent measures of discharge for (a) all sites and (b) sites with >30 observations (57 sites) with trend lines denoting within-site relationships between concentration and discharge. Each site is represented by a separate colour. Because of the log scale for CH₄, negative and zero values are omitted.**




**Fig. 14: Boxplots of site-averaged CH₄ concentration (a) and diffusive flux (b) for channel type categories. Channel categories are defined in Table 1, but briefly are as follows: NORM- non-targeted sites; CAN-canals; CH- channelized streams; DC-river delta channels; DD- downstream of dams; DIT- ditches; FP- floodplain backwaters; GT- glacial outflows; IMP- impounded reaches; PI-permafrost (thermokarst) influenced; PS- point source influenced; SP- springs; TH- thermogenic CH₄ inputs; WS- wetland streams. Number of sites per channel type are listed on the right side of each plot. The vertical black line denotes the median concentration and flux for non-targeted (NORM) sites. Because a log-scale is used in these plots, zeros and negative values were excluded. The actual median for non-targeted sites represented by the vertical line is therefore slightly different than the median displayed in the corresponding box plot because of this exclusion. The upper and lower edges of each box are the 25th and 75th percentiles, whiskers are drawn up to 1.5 times the interquartile range, and points are plotted if beyond the whiskers.**




## 4. Discussion

The rapid increase in availability of aquatic $CH_4$ (as well as $CO_2$ and $N_2O$) data over the past 5-10 years has been remarkable and creates new opportunities for examining patterns and drivers of these gases across broad spatial scales in lotic ecosystems. Similarly, constructing GRiMeDB provided us with an unprecedented opportunity to identify tendencies in when, where, and

how $CH_4$ has been sampled in streams and rivers. Examination of such data collection tendencies can reveal important biases and gaps within a field (Stanley et al. 2019, Gomez-Gener et al. 2021b) and thus points to future research needs and opportunities. Below, we discuss the distribution of sampling efforts and methodological issues, preliminary data analyses, and consider questions that GRiMeDB can help to answer.

## 4.1 When and where: sampling effort considerations

The growth of GHG studies in flowing water systems in the past decade includes a geographic expansion beyond the large body of historic and current work in temperate regions of North America and Europe. In particular, recent research in Africa, Australia, and especially southeast Asia has greatly improved the global coverage of available data. However, studies in arid drainages remain scarce- even beyond what would be expected given their small river surface area. A possible explanation for

the limited study of $CH_4$ in these systems may be the pervasive focus on the contribution of streams and rivers to the global atmospheric $CH_4$ pool, and the corresponding assumption that arid land systems play a minor role in this context. Yet we suggest that limited study in arid and semi-arid drainages represents a missed opportunity to understand metabolism and carbon cycling in a set of streams and rivers that drain nearly half of the global land surface, are increasingly stressed by growing human water demands (e.g., Sabo et al., 2010; Lian et al., 2021; Stringer et al., 2022), and support ecosystem process rates

that are amplified by warm temperatures and highly variable flow regimes (Fisher et al., 1982; Ran et al., 2021). Beyond arid and semi-arid basins, further research emphasis in tropical and high-latitude regions would also be beneficial even given recent improvements in data availability and geographic representation of both areas. Existing data for tropical forests and grasslands are dominated by studies of African rivers (especially the Congo drainage) and the Amazon River system. In fact, observations from tropical areas of the Indomalayan and northern Australasian region represent <3% of all sites, and Central America is

represented by a single study. Tropical drainages are frequently characterized by high $CH_4$ concentrations and fluxes, along with rapid changes in land use and river regulation that are affecting C cycling and GHG dynamics (Park et al., 2018; Flecker et al. 2022). However, understanding or detecting the magnitude and consequences of these anthropogenic changes on fluvial $CH_4$ is constrained by these current sampling limitations. Finally, while high latitude regions (north of the Arctic Circle) are well represented in GriMeDB with >3,600 concentration observations, more than 80% of these values are derived from studies

in the vicinity of the Toolik Field Station in Alaska, USA, and thus do not capture the full biophysical diversity of Arctic biomes (Metcalfe et al. 2018). Given that climate change at high latitudes is progressing faster than elsewhere on the planet (IPCC, in press), and that the global north stores massive quantities of C in soils (Hugelius et al., 2014), more extensive coverage of $CH_4$ across Arctic drainage systems is warranted.



Although the spatial coverage of $CH_4$ data has improved markedly over the past decade, expansion across temporal dimensions has lagged. The predominant mode of sample collection has been and continues to be through surveys that yield one or a few observations from individual sites (e.g., Bouillon et al., 2012; Kuhn et al., 2017; Jin et al., 2018; Ho et al., 2022) and studies characterizing seasonal dynamics or responses to a site-specific environmental change are limited. Indeed, long-term (>5 years) $CH_4$ datasets in general are extremely rare (Leng et al., 2021); no such data are currently available for fluxes and most long-

term concentration records are derived from just a few clustered locations. Determining the consequences of changes in land use or habitat attributes on fluvial $CH_4$ dynamics have instead relied on space-for-time substitutions (e.g., Smith et al., 2017; Gatti et al., 2018; Woda et al. 2020) rather than on direct observations of change over time. Although this strategy has been successful in revealing variation in GHG dynamics among different site types, current knowledge about how gases vary over time and respond to perturbations is poorly developed because of these data limitations. This deficit may be particularly

consequential in the case of climate change, as the broad scope of this phenomenon will inevitably limit the effectiveness of spatial sampling approaches.

  The discussion above regarding the 'when' and 'where' of sampling emphasizes large spatial and relatively long temporal scales, consistent with the extent of GRiMeDB. However, another current deficit in our understanding relates to the degree of

heterogeneity of this gas at fine spatial and temporal scales, and thus if current sampling strategies are missing meaningful variation. Recent studies of $CO_2$ provide a cautionary tale in this context, as failure to account for diurnal variation in this gas results in a consistent under-estimation of fluvial emissions that is quantifiable at regional (Attermeyer et al., 2021) and global (Gómez-Gener et al., 2021b) scales. Similar questions may arise for spatial variation; that is, what is the minimum grain size or appropriate spatial scale for sampling of $CH_4$ in running waters (Crawford et al., 2017; Lupon et al., 2019)? The potential

to examine very short-term variation is not possible using GRiMeDB data because of our decision to average of within-day measurements given the current small number (ca. 20) of these temporally-detailed studies. Assessment of fine-scale spatial variation is also limited because of limited fine-scale sampling in general, as well as by decisions made both by investigators and during database construction. For example, geomorphologically distinct units (e.g., an individual riffle or pool) are often used as a basic sampling unit and results are presented as averages of replicates collected at different points within the study

reach (e.g., Hlaváčová et al., 2006; Smith et al., 2017). In general, information about replication was frequently omitted, or if reported, information about variability among replicates was frequently absent. In addition to this limitation, our decision to combine replicates taken at different points in a channel cross-section and within individual channel units that had hundreds to thousands of datapoints also constrains the opportunity to examine variation at fine spatial scales. However, we anticipate that this situation will change over the next few several years, as *in situ* sensors or other devices capable of collecting high-

frequency/high density gas measurements become more widely available. Recent papers signal this new frontier and have highlighted the presence (e.g., Lamarche-Gagnon et al., 2019; Smith and Bohlke, 2019; Chen et al., 2021; Taillardat et al., 2022) and absence (e.g., Castro-Morales et al., 2022; Rovelli et al., 2022; Zhang et al., 2021) of predictable diel variation in





CH$_4$ concentrations and fluxes, and varying degrees of within-reach spatial variability (Crawford et al., 2016; 2017; Call et al., 2018; Bussman et al., 2022).


### 4.2 How: methodological considerations

Measuring dissolved GHG concentrations or fluxes involves multiple steps and calculations. Field and laboratory protocols vary widely in the literature, and methodological variety is particularly conspicuous for flux determination. Ironically, even though many studies of lotic CH$_4$ dynamics are framed in terms of understanding the contribution of these ecosystems to the

rapidly increasing atmospheric CH$_4$ pool, flux measurements lag far behind those of concentration, and the vast majority (ca. 85%) of flux data quantify only the diffusive pathway. Ebullition measurements are notably scarce despite the potential of this pathway to account for a large fraction of total emissions in some streams (e.g., from 30-90% of total CH$_4$ emissions; Baulch et al., 2011; Crawford et al., 2014; Chen et al., 2021). The conventional approach to quantifying ebullition involves a combination of capturing bubbles just below the water surface to determine the area and time-specific rate of bubble volume

reaching the surface and measuring CH$_4$ content of recently-erupted bubbles. The episodic nature and extreme spatial heterogeneity of ebullition (Crawford et al., 2014; Spawn et al., 2015; Robison et al., 2021) requires good replication of bubble traps that need to be deployed over multiple days to generate reliable measurement. Given the logistic challenges and labour-intensive work involved, indirect approaches are becoming more common. These approaches typically use the difference between a chamber-based measurement of flux, which is assumed to represent total flux (diffusion + ebullition) and diffusion

calculated from dissolved CH$_4$ and $k$ (i.e., the 'chamber – [concentration + $k$]' method in Fig. 9) to estimate ebullition (e.g., Campeau et al., 2014; Zhang et al., 2020; Ran et al., 2021). We suggest that this approach should be used cautiously, however. For example, this strategy is arguably inappropriate for situations in which the chamber gas content increases in a linear fashion, consistent with the occurrence of diffusive flux alone. Further, relatively short chamber deployments are likely to miss or incompletely capture bubble releases, while long-term deployments are vulnerable to sampling artefacts associated with

altered concentration gradients within, and/or turbulence around the chamber (Sawakuchi et al., 2014; Lorke et al., 2015). Given these challenges, it is not altogether surprising that comparisons between direct and indirect measurements of ebullition can yield substantially different results (e.g., Yang et al., 2012; Bednařík et al., 2017; Chen et al., 2021).

The final and most profound knowledge gap in the collection of flux data is the absence of measurements of plant-mediated emissions. Plant-mediated fluxes can account for a substantial fraction of total emissions from wetlands and shallow lake

habitats (Bodmer et al., 2021) but the contribution of this pathway is unknown in fluvial systems. Indeed, we did not include plant mediated fluxes in GRiMe DB because we encountered only two papers that had explicitly quantified this pathway in streams (Sanders et al., 2007; Wilcock and Sorrell, 2008). Although aquatic macrophytes are sparse or absent from many streams and rivers, they can be abundant in low-gradient, low-disturbance environments (Riis and Biggs, 2003; Gurnell et al., 2010) where diffusive fluxes would be constrained by low gas exchange rates. Sediment trapping and venting by macrophytes

enhances both methanogenesis and methane emission in these systems (Sanders et al., 2007), but the significance of such



processes and the contribution of plant-mediated fluxes at larger spatial scales remain to be determined for fluvial systems (Bodmer et al., 2021).

**4.3 Concentration and flux patterns**

Not surprisingly, the massive increase in data availability have led to differences in averages and measures of variability for $CH_4$ concentrations and fluxes compared our previous efforts. Median values for all three $CH_4$ flux pathways in GRiMeDB are 1.2-2.2 times lower than those reported by Stanley et al. (2016), as well as those from Rosentreter et al. (2021). Conversely, measures of variability (SD, CV) in GRiMeDB are almost 3-fold greater than previous estimates, undoubtedly due to the far larger number of observations, the associated expansion of geographic scope and channel types, and the higher temporal

resolution of the data. For any sampling effort, the standard deviation increases with increasing sample size, but eventually reaches a plateau that indicates a sample size sufficient to capture the true population variability. It is not yet clear if the sample sizes are sufficient to capture the true global-scale variability of fluvial concentrations and fluxes, and future database updates should be used to examine this relationship.

Despite the slight lowering of median values, supersaturated concentrations and positive fluxes are the norm for $CH_4$ as well as for $CO_2$ and $N_2O$. However, it is likely that $CH_4$ concentrations and fluxes below detection limits (BDLs) are under-reported, as is common with environmental data in general (Stow et al., 2018), so current averages may be slight overestimations of true population medians. Even given the modest number of zero or undetectable $CH_4$ concentrations in GriMeDB (<2.5%), decisions about handling BDLs can have a small but detectable effect on the estimation of global averages. For example, if

these observations are excluded, median $CH_4$ concentrations for all other observations increases from 1.49 to 1.51 µmol L$^{-1}$. If we keep all of these observations and assign them a value of zero (an unlikely scenario, but used here to provide a lower limit for this example), then the overall median declines to 1.46 µmol L$^{-1}$. Although this difference is relatively small, it would likely be consequential for upscaling estimates. At a minimum, we urge GRiMeDB users to be aware of how these values are handled and encourage future researchers to determine and report detection limits and include samples that fall below these

limits in their results.

A goal of assembling GRiMeDB was to centralize $CH_4$ data to foster future research efforts. To this end, we also included information about habitat conditions that allows the exploration of relationships between $CH_4$ and potential explanatory variables and covariates. To demonstrate this opportunity, we provided a limited number of graphic examples of $CH_4$ versus

variables that have been identified as potential predictors or drivers of $CH_4$ production, concentration, or flux (Figs. 11-13), and these plots suggest both the presence and absence of relationships. For example, increasing $CH_4$ concentrations have been associated with low or decreasing dissolved oxygen and/or increasing organic carbon (e.g., Borges et al., 2018; Jin et al. 2018; Begum et al., 2021) and these relationships are recognizable across the entirety of the GRiMeDB data. Similarly, increases in discharge have been linked to declines in gas concentration, likely due to source limitation (i.e., dilution) of terrestrial supply



(Aho et al., 2021; Gómez- Gómez-Gener et al., 2021a) and/or greater water turbulence, which increases gas exchange and in turn can reduce supersaturated $CH_4$ stocks (Billett and Harvey, 2013; Kokic et al. 2018).  This relationship is not obvious when all data were considered en masse, but became more apparent when examining within-site dynamics. In contrast to these three confirmatory examples, although latitude and channel size have also been identified as determinants of $CH_4$ concentrations or used to extrapolate site-specific gas measurements to larger (even global) scales (e.g., Bastviken et al., 2011; Li et al., 2020),

evidence for such relationships is not apparent from our analysis. Further, even for the former examples that indicated relationships between $CH_4$ concentration and DO, DOC, or discharge, there is substantial variability present in these relationships, the strength of these predictors is likely to vary across scales, and explains little of the variability for diffusive fluxes. In short, substantial opportunities exist to identify multivariate relationships between different predictors and $CH_4$ concentrations and fluxes across different scales, and pursuit of these opportunities should be improved by the substantial

increase in data for both gases and potential predictor variables.

The disproportionate contribution of streams and rivers to atmospheric inputs together with the utility of $CH_4$ as an indicator of anthropogenic influences on drainage systems have inspired several studies that focus on fluvial habitats that are expected to have high concentrations and fluxes. Many of these 'methane hunting' studies have demonstrated significant increases in

$CH_4$ concentrations and/or fluxes associated with phenomena such as point source inputs, ditch and canal construction, oil and gas extraction, or passage through wetlands. Such signals persist at the global scale (Fig. 14), highlighting widespread human enhancement of $CH_4$ emissions from lotic ecosystems. Not all targeted sites are $CH_4$-rich however.  Low concentrations in glacial outflows (GT) likely reflect the effects of cold temperatures and/or low organic carbon availability (Crawford et al., 2015; Burns et al., 2018) while low values at floodplain (FP) sites may be attributable to their more characteristically lentic

conditions, which allows higher rates of $CH_4$ oxidation in the water column. Indeed, oxidation has been shown to represent a significant $CH_4$ sink in floodplain lakes associated with the Amazon River (Barbosa et al., 2018) and most of the FP sites in GRiMeDB are part of the Amazon system.

As noted in Sect. 3.4, the availability of supporting information is inconsistent, as, for example, only ~25% of data sources provided information on channel order or basin size. However, the growing availability of open-access regional and global

geospatial datasets that provide information about site characteristics (e.g., Linke et al., 2019, Yang et al., 2020) has increased rapidly in the past decade. Recent upscaling efforts analyses (Rosentreter et al., 2021; Liu et al., 2022; Rocher-Ros et al., in review) have, for example, benefited from improved estimates of the surface area of world streams and rivers (Allen and Pavelsky, 2018; Yang et al., 2020), while the diverse datasets in HydroSHEDS (Linke et al., 2019) allowed us to evaluate the global representativeness of GRiMeDB sites. As new global-scale datasets become available, we anticipate that their pairing

with GRiMeDB data will result in significant improvements in the strength and certainty of continental and global-scale models explaining $CH_4$ distribution and quantifying fluvial emissions to the atmosphere.





**5. Data and code availability**


GRiMeDB and its associated metadata are available from the the Environmental Data Initiative (Stanley et al., 2022): https://doi.org/10.6073/pasta/b7d1fba4f9a3e365c9861ac3b58b4a90

Code used for unit conversions, spatial analyses, and general data analysis and visualization will be available from

EDI.


**6. Conclusion**

The data gathered in GRiMeDB highlights many new opportunities, both through analysis of $CH_4$ and supporting data in the database, and by revealing gaps that currently exist across fluvial $CH_4$ studies. The most conspicuous data limitations include deficits in measurements of non-diffusive flux pathways in underrepresented arid, tropical, and arctic biomes. Challenges

associated with quantifying ebullition discussed above also emphasize the need for more intercomparisons among the various flux methods. Regardless of pathway, flux is a difficult process to quantify and can be highly sensitive to methods or gas exchange model choices and there are few methodological comparisons available to inform these decisions (Raymond et al., 2012; Lorke et al., 2015). Finally, we highlight that the expansion of GHG data world streams and rivers over the past decade has proceeded largely across spatial rather than temporal dimensions. While this expansion has vastly improved the geographic

representativeness of the data, long-term datasets are rare despite their power for generating ecological understanding and informing policy/management in the face of environmental change (Hughes et al., 2017). Unfortunately, GHG´s, particularly $CH_4$ and $N_2O$, are rarely included as routine components of water quality monitoring programs. Thus, we emphasize the compelling need to establish such sampling efforts and perpetuate those few that do exist.

Despite highlighting these areas of data limitation in the field, it is important to underscore the opportunities that the growth

in GHG data availability- especially of $CH_4$ data- now provide. Assembly of GRiMeDB was motivated by the goal of having a centralized, standardized resource to facilitate further studies of $CH_4$ pattern and process in flowing water systems. Our strategy in developing this database was to maximize opportunities for identifying patterns and relationships involving this gas in future analyses. Past difficulties with such efforts may well be a product of the common practice of averaging values over time or among sites and/or of including non-fluvial sites in analyses. Thus, we carefully documented the data and resolved

observations to individual sites and dates whenever possible to match the pronounced spatial and temporal variance of this gas. Similarly, while we included a range of habitat types in GRiMeDB, unconventional or targeted sites are easily identifiable. Further, we carefully examined sites to ensure that they were not subject to impounding effects of a dam or were not situated in estuaries where distinct processes such as tidal cycles and elevated sulphate concentrations may obscure or overtake relationships present in inland flowing water systems. Thus, we are optimistic that analysis of GriMeDB data by itself, or in

concert with other complementary datasets will provide new and unprecedented opportunities to examine relationships





between $CH_4$ and environmental drivers or correlates, as well as providing broad contextual information for site-based studies of fluvial carbon and GHG dynamics.





**Appendix A. GRiMeDB tables and variables**

**Table A1. Column titles and description of their content for the GRiMeDB Sources Table.**

| Column Title | Description |
|---|---|
| Title | Title of data source. |
| Author | Lead author last name |
| Source | Identity of the outlet for the data (e.g., journal, data repository, agency that presented the data). For titles with published papers paired with published datasets, the journal is listed in this column |
| Pub_year | Year of publication, data release, or acquisition of an unpublished dataset |
| Source_ID | Unique data source identifier |
| Additional_data | "Yes" in this column indicates that additional data were acquired directly from the author for any field. Additions are described in the Comments field |
| Comments | Additional information or clarification about the data source |
| Paper_DOI | DOI or hyperlink for journal article or other publication based on the $CH_4$ data |
| Data_DOI_primary | DOI or hyperlink for $CH_4$ data posted in a data repository |
| Data_DOI_supporting | DOI or hyperlink for separate datasets providing supporting data |




**Table A2. Column titles and content description for the GRiMeDB Sites Table.**

| Column Title | Definition |
| --- | --- |
| Source_ID | Unique data source identifier from the Sources Table |
| Site_ID | Unique site identifier |
| Site_Name | Unique site name |
| Stream_Name | Stream or river name; taken or modified from the data source or generated *de novo* when a name was not specified in the data source |
| Aggregated | Yes or No; "Yes" if $CH_4$ data entered are averages from >1 site |
| N_sites_aggregated | Number of sites that were averaged for aggregated sites |
| Basin_Region | Name of the larger drainage basin or region that contains the site. This information is included to facilitate site grouping during data analysis |
| Latitude | Decimal degrees, WGS84 ensemble: EPSG:4326 coordinate system |
| Longitude | Decimal degrees, WGS84 ensemble: EPSG:4326 coordinate system |
| Elevation_m | Reported meters above sea level |
| Slope_m_per_m | Reported channel slope expressed as $m\ m^{-1}$ |
| Strahler_order | Reported Strahler stream order |
| Basin_size_km$^2$ | Reported basin size in square kilometers |
| Channel_type | Codes denoting distinct site or channel attributes or presence of specified conditions. See Table 1 for categories and their definitions |
| Latitude_snapped | Latitude in decimal degrees for site location after snapping to the closest channel for elevation determination |
| Longitude_snapped | Longitude in decimal degrees for site location after snapping to the closest channel for elevation determination |
| Elevation_estimated_m | Elevation (meters above sea level) calculated from the DEM. See Sect. 2.2 for details |
| Comments | Additional information or clarification about the site source |



**Table A3. Column titles and definitions for the GRiMeDB Concentration Table**

| Column Title | Definition |
|---|---|
| Source_ID | Unique paper identifier from the Sources Table |
| Site_ID | Unique paper identifier from the Sites Table |
| Site_Name | Unique site name from the Sites Table |
| Conc_Name | Unique name for the sampling event at the site; same as Flux_Name in the Fluxes Table if both concentration and flux data for the same site-date combination are available |
| Date_start | First sampling date |
| Date_end | Last sampling date; this is the same date as the Date_start if data are not aggregated over time |
| Aggregated_Space | Yes or No; "Yes" if $CH_4$ data entered are averages from >1 site |
| Aggregated_Time | Yes or No; "Yes: if $CH_4$ data entered are averages from >1 date |
| FluxYesNo | Yes or No; "Yes" if there is a corresponding flux measurement associated with this site-date combination |
| SampleCount | Number of samples or observations corresponding to the mean or median concentration |
| CH4min | Minimum measured $CH_4$ concentration if data are aggregated spatially or temporally, has multiple within-day measurements (e.g., a diel study), or are from a data-dense spatial study |
| CH4max | Maximum measured $CH_4$ concentration if data are aggregated or temporally, has multiple within-day measurements (e.g., a diel study), or are from a data-dense spatial study |
| CH4mean | Mean or sole reported $CH_4$ concentration for the sampling event |
| CH4_SD | Standard deviation of the mean $CH_4$ concentration |
| CH4median | Median $CH_4$ concentration |
| CO2min | Minimum measured $CO_2$ concentration if data are aggregated or temporally, has multiple within-day measurements (e.g., a diel study), or are from a data-dense spatial study |
| CO2max | Maximum measured $CO_2$ concentration if data are aggregated or temporally, has multiple within-day measurements (e.g., a diel study), or are from a data-dense spatial study |
| CO2mean | Mean or sole reported $CO_2$ concentration for the sampling event |
| CO2_SD | Standard deviation of the mean concentration |
| CO2median | Median $CO_2$ concentration |
| N2Omin | Minimum measured $N_2O$ concentration if data are aggregated or temporally, has multiple within-day measurements (e.g., a diel study), or are from a data-dense spatial study |



| | |
|---|---|
| N2Omax | Maximum measured $N_2O$ concentration if data are aggregated or temporally, has multiple within-day measurements (e.g., a diel study), or are from a data-dense spatial study |
| N2Omean | Mean or sole reported $N_2O$ concentration for the concentration for the sampling event |
| N2O_SD | Standard deviation of the mean $N_2O$ concentration |
| N2Omedian | Median $N_2O$ concentration |
| WaterTemp_degC | Water temperature in degrees C measured concurrently with $_{CH4}$ |
| WaterTemp_degC _estimated | Estimated water temperature in degrees C. This field was populated only for cases in which temperature was needed for gas unit conversion. Most estimates were based on temperatures from adjacent sites, averaging temperatures from prior and proceeding sample dates, or from an adjacent day of the year but from another year. |
| Cond_uScm | Specific conductance in $\mu S\ cm^{-1}$ |
| pH | pH |
| DO_mgL | Dissolved oxygen in mg $L^{-1}$ |
| DO_percentsat | Percent saturation of dissolved oxygen |
| Q | Discharge measured at the time of sample collection |
| NO3 | $NO_3$ or $NO_2+NO_3$ measured concurrently with $CH_4$ |
| NH4 | $NH_4$ measured concurrently with $CH_4$ |
| TN | TN or TDN measured concurrently with $CH_4$ |
| SRP | SRP or $PO_4$ measured concurrently with $CH_4$ |
| TP | TP or TDP measured concurrently with $CH_4$ |
| DOC | DOC or TOC measured concurrently with $CH_4$ |
| Comments | Any additional relevant information regarding data |
| new_CH4_unit | Current common units for all $CH_4$ concentrations |
| new_CO2_unit | Current common units for all $CO_2$ concentrations |
| new_N2O_unit | Current common units for all $N_2O$ concentrations |
| new_NO3_unit | Current common units for all $NO_3$ or $NO_2+NO_3$ concentrations |
| new_NH4_unit | Current common units for all $NH_4$ concentrations |
| new_TN_unit | Current common units for all TN or TDN concentrations |



| | |
|---|---|
| new_SRP_unit | Current common units for all SRP or $PO_4$ concentrations |
| new_TP_unit | Current common units for all TP or TDP concentrations |
| new_DOC_unit | Current common units for all DOC or TOC concentrations |
| new_Q_unit | Current common units for all discharge measurements |
| orig_CH4_unit | Original units for $CH_4$ concentration |
| orig_CO2_unit | Original units for $CO_2$ concentration |
| orig_N2O_unit | Original units for $N_2O$ concentration |
| orig_NO3_unit | Original units for $NO_3$ or $NO_2+NO_3$ concentration |
| orig_NH4_unit | Original units for $NH_4$ concentration |
| orig_TN_unit | Original units for TN concentration |
| orig_SRP_unit | Original units for SRP or $PO_4$ concentration |
| orig_TP_unit | Original units for TP concentration |
| orig_DOC_unit | Original units for DOC concentration |
| orig_Q_unit | Original units of discharge |




**Table A4. Column titles and definitions for the GRiMeDB Flux Table**

| Column Title | Definition |
| --- | --- |
| Source_ID | Unique paper identifier from the Sources Table |
| Site_ID | Unique paper identifier from the Sites Table |
| Site_Name | Unique site name from the Sites Table |
| Flux_Name | Unique name for the sampling event at the site; same as Conc_Name in the Concentrations Table if both concentration and flux data for the same site-date combination are available |
| Date_start | First sampling date |
| Date_end | Last sampling date; this is the same date as the Date_start if data are not aggregated over time |
| Aggregated_Space | Yes or No; "Yes" if $CH_4$ data entered are averages from >1 site |
| Aggregated_Time | Yes or No; "Yes: if $CH_4$ data entered are averages from >1 date |
| Diffusive_CH4_Flux_Min | Minimum measured $CH_4$ diffusive flux if data are aggregated or are from diel or data-dense spatial studies |
| Diffusive_CH4_Flux_Max | Maximum measured $CH_4$ diffusive flux if data are aggregated or are from diel or data-dense spatial studies |
| Diffusive_CH4_Flux_Mean | Mean or sole reported $CH_4$ diffusive flux for the sampling event |
| Diffusive_CH4_Flux_SD | Standard deviation of the mean $CH_4$ diffusive flux |
| Diffusive_CH4_Flux_Median | Median $CH_4$ diffusive flux |
| SampleCount_Diffusive | Number of samples or observations corresponding to the mean or median diffusive $CH_4$ flux |
| Diff_Method | Methodological category used to measure diffusive gas flux. Categories (with brief explanations in italics) are: |

chamber (unspecified)- unspecified response
*use of an unspecified type of chamber (suspended, tethered, or free-floating) and pattern of change gas concentration over time during flux measurements is also not specified*

chamber (unspecified)- linear response
*unspecified type of chamber with a linear increase in chamber gas concentration over time or use of a linear model to calculate flux*

suspended/tethered chamber-unspecified response





*chamber is restrained to maintain its position and not float downstream during flux measurement*

suspended/tethered chamber- linear response

floating chamber- unspecified response
*chamber is unrestrained and is able to float downstream during flux measurement*

floating chamber- linear response

conc+k
*diffusive flux calculated using the equation:*
$$flux = k(C_w\text{-}C_{eq}), where$$
*$k$ = gas exchange coefficient*
*$C_w$ = CH$_4$ concentration measured in water*
*$C_{eq}$ = CH$_4$ concentration in water in equilibrium with the atmosphere*

other

*methods other than those described above*

| | |
|---|---|
| Eb_CH4_Flux_Min | Minimum measured CH$_4$ ebullitive flux if data are aggregated or are from diel or data-dense spatial studies |
| Eb_CH4_Flux_Max | Maximum measured CH$_4$ ebullitive flux if data are aggregated or are from diel or data-dense spatial studies |
| Eb_CH4_Flux_Mean | Mean or sole reported CH$_4$ ebullitive flux for the sampling event |
| Eb_CH4_Flux_SD | Standard deviation of the mean CH$_4$ ebullitive flux |
| Eb_CH4_Flux_Median | Median CH$_4$ ebullition flux |
| SampleCount_Eb | Number of samples or observations corresponding to the mean or median ebullitive CH$_4$ flux |
| Eb_Method | Methodological category used to measure ebullitive gas flux. Categories (with brief explanations in italics) are: |

chamber minus conc+k
*ebullition calculated as chamber-measured flux (assumed to be total CH$_4$ flux) minus diffusive flux calculated from the 'conc+k' method*

bubble trap + bubble analysis
*gas released by ebullition captured in traps to quantify total gas volume; volume data combined with measurement of CH$_4$ content of recently collected bubbles*



| | |
|---|---|
| | echosounder + bubble analysis<br>*gas bubble volume determined using echosounder and combined with $CH_4$ content of recently collected bubbles* |
| | departure from linear increase during measurement<br>*non-linear change in gas concentrations during chamber-based flux measurements taken as evidence of ebullition; various approaches used to quantify ebullition from these departures* |
| | other<br>*methods other than those described above* |
| Total_CH4_Flux_Min | Minimum measured total $CH_4$ flux |
| Total_CH4_Flux_Max | Maximum measured total $CH_4$ flux |
| Total_CH4_Flux_Mean | Mean or sole reported total $CH_4$ flux for the sampling event |
| Total_CH4_Flux_SD | Standard deviation of the mean total $CH_4$ flux |
| Total_CH4_Flux_Median | Median measured total $CH_4$ flux |
| Total_Method | Methodological category used to measure total $CH_4$ flux. Categories (with brief explanations in italics) are:<br><br>conc+k and ebullition<br>*total flux calculated as the sum of separate measurements of diffusion determined by the conc+k method plus ebullition determined from the bubble trap or echosounder approach combined with bubble $CH_4$ analysis*<br><br>floating chamber<br>*free-floating chamber is assumed to capture diffusive flux and ebullitive flux (if present)*<br><br>suspended/tethered chamber<br>*suspended or tethered chamber is assumed to capture diffusive flux and ebullitive flux (if present)*<br><br>chamber and ebullition<br>*total flux calculated as the sum of separate measurements of diffusion determined using a floating or suspended/tethered chamber plus ebullition determined from the bubble trap or echosounder approach combined with bubble $CH_4$ analysis*<br><br>mass balance<br>*total flux represents the difference between all measured inputs to a reach (e.g., dissolved $CH_4$ from upstream flow, groundwater discharge, and methanogenesis) minus all outputs other than efflux to the atmosphere (e.g., downstream export, methane oxidation)* |



|  |  |
| --- | --- |
| other | *methods other than those described above* |
| CO2_Flux_Min | Minimum measured $CO_2$ flux if data are aggregated or are from diel or data-dense spatial studies |
| CO2_Flux_Max | Maximum measured $CO_2$ flux if data are aggregated or are from diel or data-dense spatial studies |
| CO2_Flux_Mean | Mean or sole reported $CO_2$ diffusive flux for the sampling event |
| CO2_Flux_SD | Standard deviation of the mean $CO_2$ flux |
| CO2_Flux_Median | Median $CO_2$ flux |
| N2O_Flux_Min | Minimum measured $N_2O$ flux if data are aggregated or are from diel or data-dense spatial studies |
| N2O_Flux_Max | Maximum measured $N_2O$ flux if data are aggregated or are from diel or data-dense spatial studies |
| N2O_Flux_Mean | Mean or sole reported $N_2O$ diffusive flux for the sampling event |
| N2O_Flux_Stddev | Standard deviation of the mean $N_2O$ flux |
| N2O_Flux_Median | Median $N_2O$ flux |

k_Method    Methodological category used for estimating the gas exchange coefficient, *k*, Categories (with brief explanations in italics) are:

physical model
: *k calculated using equations based on physical variables such as channel slope, water velocity, etc.*

chamber + conc
: *k determined by chamber-based measurements of flux, dissolved gas concentration, and re-arrangement of the flux equation*
  $$flux = k(C_w\text{-}C_{eq})$$
  *to solve for k. Typically, these measurements are made for $CO_2$, and then $k_{CO2}$ is converted to $k_{CH4}$*

tracer addition
: *paired conservative and gas tracer additions used to calculate k from concentration declines along a stream reach*

assigned k value
: *use of k values from other dates or sites in the same study or k values considered to be characteristic of the site*

other
: *methods other than those described above*





|  | unknown *method to determine k is not described* |
| --- | --- |
| k_ref | k method citation reported in the data source |
| Comments | Any additional relevant information regarding data entered in this row |
| new_Diffusive_Flux_unit | Current common units for all diffusive $CH_4$ flux data |
| new_Eb_CH4_Flux_unit | Current common units for all ebullitive $CH_4$ flux data |
| new_Total_Flux_unit | Current common units for all total $CH_4$ flux data |
| new_CO2_Flux_unit | Current common units for all $CO_2$ flux data |
| new_N2O_Flux_unit | Current common units for all $N_2O$ flux data |
| orig_Diffusive_Flux_unit | Original units for diffusive $CH_4$ flux |
| orig_Eb_CH4_Flux_unit | Original units for ebullitive $CH_4$ flux used |
| orig_Total_Flux_unit | Original units for total $CH_4$ flux |
| orig_CO2_Flux_unit | Original units for $CO_2$ flux |
| orig_N2O_Flux_unit | Original units for $N_2O$ flux |




**Appendix B. Citations for data sources in GRiMeDB**. Citations are not provided for unpublished datasets.

Abbott, B. and Jones, J.: Soil respiration, water chemistry, and soil gas data for thermokarst features and undisturbed tundra on the North Slope of Alaska, Arctic Data Center, https://doi.org/10.18739/A23T9D71C, 2013.

Abbott, B. W., Jones, J. B., Godsey, S. E., Larouche, J. R., and Bowden, W. B.: Patterns and persistence of hydrologic carbon and nutrient export from collapsing upland permafrost, Biogeosciences, 12, 3725–3740, https://doi.org/10.5194/bg-12-3725-2015, 2015.

Abril, G., Guérin, F., Richard, S., Delmas, R., Galy-Lacaux, C., Gosse, P., Tremblay, A., Varfalvy, L., Dos Santos, M. A., and Matvienko, B.: Carbon dioxide and methane emissions and the carbon budget of a 10-year old tropical reservoir (Petit Saut,

French Guiana), Global Biogeochem. Cycles, 19, GB4007, https://doi.org/10.1029/2005GB002457, 2005.

Adams, D. D. and Simiyu, G. M.: Greenhouse gas (methane and carbon dioxide) emissions from a tropical river in Kenya: the importance of anthropogenic factors on natural gas flux rates, SIL Proceedings 1922–2010, 30, 887-889, https://doi.org/10.1080/03680770.2009.11902264, 2009.

Aho, K. S. and Raymond, P. A.: Differential response of greenhouse gas evasion to storms in forested and wetland streams, J.

Geophys. Res. Biogeosci., 124, 649–662, https://doi.org/10.1029/2018JG004750, 2019.

Aho, K., Cawley, K., DelVecchia, A., Stanley, E., and Raymond, P.: Dissolved greenhouse gas concentrations derived from the NEON dissolved gases in surface water data product (DP1.20097.001), Environmental Data Initiative, https://doi.org/10.6073/pasta/47d7cb6d374b6662cce98e42122169f8, 2021a

Aho, K. S., Fair, J. H., Hosen, J. D., Kyzivat, E. D., Logozzo, L. A., Rocher-Ros, G., Weber, L. C., Yoon, B., and Raymond,

P. A.: Distinct concentration-discharge dynamics in temperate streams and rivers: $CO_2$ exhibits chemostasis while $CH_4$ exhibits source limitation due to temperature control, Limnol Oceanogr, 66, 3656–3668, https://doi.org/10.1002/lno.11906, 2021b.

Aho, K., Fair, J., Hosen, J., Kyzivat, E., Logozzo, L., Weber, L., Yoon, B., Zarnetske, J., and Raymond, P.: Dissolved $N_2O$ measurements from the Connecticut River Watershed, Environmental Data Initiative, https://doi.org/10.6073/pasta/3494ca49fc3283eea5e4fc2f8a24ce3b, 2021c.

Aho, K., Hoyle, J., Hosen, J., Kyzivat, E., Logozzo, L., Rocher-Ros, G., Weber, L., Bryan, Y., and Raymond, P.: Dissolved $CO_2$ and $CH_4$ concentrations in the Connecticut River Watershed, Environmental Data Initiative, https://doi.org/10.6073/pasta/af4daec813775b7f426a1db574cbebc7, 2021d.



Alshboul, Z., Encinas-Fernández, J., Hofmann, H., and Lorke, A.: Export of dissolved methane and carbon dioxide with effluents from municipal wastewater treatment plants, Environ. Sci. Technol., 50, 5555–5563, https://doi.org/10.1021/acs.est.5b04923, 2016.

Andrews, L. F., Wadnerkar, P. D., White, S. A., Chen, X., Correa, R. E., Jeffrey, L. C., and Santos, I. R.: Hydrological, geochemical and land use drivers of greenhouse gas dynamics in eleven sub-tropical streams, Aquat. Sci., 83, 40, https://doi.org/10.1007/s00027-021-00791-x, 2021.

Anthony, S. E., Prahl, F. G., and Peterson, T. D.: Methane dynamics in the Willamette River, Oregon, Limnol. Oceanogr., 57, 1517–1530, https://doi.org/10.4319/lo.2012.57.5.1517, 2012.

Antweiler, R. C., Smith, R. L., Voytek, M. A., Bohlke, J. K., and Dupré, D. H.: Water quality data from two agricultural drainage basins in Nnrthwestern Indiana and northeastern Illinois: III. Biweekly data, 2000-2002, U.S. Geological Survey, https://pubs.usgs.gov/of/2005/1197/, 2005a.

Antweiler, R. C., Smith, R. L., Voytek, M. A., Böhlke, J.-K., and Richards, K. D.: Water quality data from two agricultural drainage basins in northwestern Indiana and northeastern Illinois: I. Lagrangian and synoptic data, 1999-2002, U.S. Geological Survey, http://pubs.water.usgs.gov/ofr20041317/, 2005b.

Arp, C., Kane, D., Hinzman, L., and Stuefer, S.: Hydrographic data, Imnavait Creek Watershed, Alaska, 1985-2017, Arctic Data Center, https://doi.org/10.18739/A2K649S9D, 2017.

Atkins, M. L., Santos, I. R., and Maher, D. T.: Seasonal exports and drivers of dissolved inorganic and organic carbon, carbon dioxide, methane and δ13C signatures in a subtropical river network, Sci. Total Environ., 575, 545–563, https://doi.org/10.1016/j.scitotenv.2016.09.020, 2017.

Audet, J., Wallin, M. B., Kyllmar, K., Andersson, S., and Bishop, K.: Nitrous oxide emissions from streams in a Swedish agricultural catchment, Agric. Ecosyst. Environ., 236, 295–303, https://doi.org/10.1016/j.agee.2016.12.012, 2017.

Bagnoud, A., Pramateftaki, P., Bogard, M. J., Battin, T. J., and Peter, H.: Microbial ecology of methanotrophy in streams along a gradient of $CH_4$ availability, Front. Microbiol., 11, 771, https://doi.org/10.3389/fmicb.2020.00771, 2020.

Baker, M. A., Dahm, C. N., Valett, H. M., Morrice, J. A., Henry, K. S., Campana, M. E., and Wroblicky, G. J.: Spatial and temporal variation in methane distribution at the ground water/surface water interface in headwater catchments, in: 2nd International Conference on Ground Water Ecology, Atlanta, GA, 29-37, https://www.osti.gov/biblio/37567, 1994.



Ballester, M. V. R. and Santos, J. E. dos: Biogenic gases in tropical floodplain river, Braz. Arch. Biol. Technol., 44, 141–147,
https://doi.org/10.1590/S1516-89132001000200006, 2001.

Bange, H. W., Sim, C. H., Bastian, D., Kallert, J., Kock, A., Mujahid, A., and Müller, M.: Nitrous oxide ($N_2O$) and methane
($CH_4$) in rivers and estuaries of northwestern Borneo, Biogeosciences, 16, 4321–4335, https://doi.org/10.5194/bg-16-4321-
2019, 2019.

Banks, E. W., Hatch, M., Smith, S., Underschultz, J., Lamontagne, S., Suckow, A., and Mallants, D.: Multi-tracer and
hydrogeophysical investigation of the hydraulic connectivity between coal seam gas formations, shallow groundwater and
stream network in a faulted sedimentary basin, J. Hydrol., 578, 124132, https://doi.org/10.1016/j.jhydrol.2019.124132, 2019.

Barbosa, P. M., Melack, J. M., Farjalla, V. F., Amaral, J. H. F., Scofield, V., and Forsberg, B. R.: Diffusive methane fluxes
from Negro, Solimões and Madeira rivers and fringing lakes in the Amazon basin: Diffusive methane fluxes from rivers and
fringing lakes, Limnol. Oceanogr., 61, S221–S237, https://doi.org/10.1002/lno.10358, 2016.

Bastien, J. and Demarty, M.: Spatio-temporal variation of gross $CO_2$ and $CH_4$ diffusive emissions from Australian reservoirs
and natural aquatic ecosystems, and estimation of net reservoir emissions, Lakes Reservoirs Res. Manage.18, 115–127,
https://doi.org/10.1111/lre.12028, 2013.

Baulch, H. M., Dillon, P. J., Maranger, R., and Schiff, S. L.: Diffusive and ebullitive transport of methane and nitrous oxide
from streams: Are bubble-mediated fluxes important?, J. Geophys. Res., 116, G04028, https://doi.org/10.1029/2011JG001656,
735  2011.

Beaulieu, J. J.: Controls on greenhouse gas emissions from headwater streams, Ph.D. thesis, University of Notre Dame, Notre
Dame, IN, USA, 205 pp., https://onesearch.library.nd.edu/permalink/f/1phik6l/ndu_aleph002323859, 2007.

Beaulieu, J. J., Shuster, W. D., and Rebholz, J. A.: Controls on gas transfer velocities in a large river, J. Geophys. Res., 117,
G02007, https://doi.org/10.1029/2011JG001794, 2012.

Bednařík, A., Čáp, L., Maier, V., and Rulík, M.: Contribution of methane benthic and atmospheric fluxes of an experimental
area (Sitka Stream), CLEAN Soil Air Water, 43, 1136–1142, https://doi.org/10.1002/clen.201300982, 2015.

Bednařík, A., Blaser, M., Matoušů, A., Hekera, P., and Rulík, M.: Effect of weir impoundments on methane dynamics in a
river, Sci. Total Environ., 584–585, 164-174, https://doi.org/10.1016/j.scitotenv.2017.01.163, 2017.



Begum, M. S., Bogard, M. J., Butman, D. E., Chea, E., Kumar, S., Lu, X., Nayna, O. K., Ran, L., Richey, J. E., Tareq, S. M., Xuan, D. T., Yu, R., and Park, J.: Localized pollution impacts on greenhouse gas dynamics in three anthropogenically modified Asian river systems, J Geophys. Res. Biogeosci., 126, e2020JG006124, https://doi.org/10.1029/2020JG006124, 2021a.

Begum, M. S., Bogard, M. J., Butman, D. E., Chea, E., Kumar, S., Lu, X., Nayna, O., K, Ran, L., Richey, J E, Tareq, S. M, Xuan, D. T., Yu, R., and Park, J.: Spatiotemporal variation in $p$CO$_2$, CH$_4$, N$_2$O, DOM, and ancillary water quality measured in
the Ganges, Mekong, and Yellow River during 2016 to 2019, PANGAEA, https://doi.org/10.1594/PANGAEA.926582, 2021b.

Billett, M. F. and Harvey, F. H.: Measurements of CO$_2$ and CH$_4$ evasion from UK peatland headwater streams, Biogeochemistry, 114, 165–181, https://doi.org/10.1007/s10533-012-9798-9, 2013.

Billett, M. F. and Moore, T. R.: Supersaturation and evasion of CO$_2$ and CH$_4$ in surface waters at Mer Bleue peatland, Canada, Hydrol. Process., 22, 2044–2054, https://doi.org/10.1002/hyp.6805, 2008.

Bilsley, N.: Processes affecting the spatial and temporal variability of methane in a temperate dammed river system, M.S. thesis, University of California San Diego, San Diego, CA, USA, 94 pp., https://search-library.ucsd.edu/permalink/01UCS_SDI/ld412s/alma991004859049706535, 2012.

Blackburn, S. R. and Stanley, E. H.: Greenhouse gas fluxes and concentrations and associated habitat data in western Dane County, Wisconsin, USA, streams during the 2018 growing season ver. 2., Environmental Data Initiative,
https://doi.org/10.6073/pasta/eeba6193162bf1189a6b78452686c773, 2019.

Blackburn, S. R. and Stanley, E. H.: Floods increase carbon dioxide and methane fluxes in agricultural streams, Freshwater Biol., 66, 62–77, https://doi.org/10.1111/fwb.13614, 2021.

Bodmer, P., Heinz, M., Pusch, M., Singer, G., and Premke, K.: Carbon dynamics and their link to dissolved organic matter quality across contrasting stream ecosystems, Sci. Total Environ., 553, 574–586,
https://doi.org/10.1016/j.scitotenv.2016.02.095, 2016.

Bolpagni, R., Laini, A., Mutti, T., Viaroli, P., and Bartoli, M.: Connectivity and habitat typology drive CO$_2$ and CH$_4$ fluxes across land-water interfaces in lowland rivers: C fluxes across lowland river interfaces, Ecohydrology, 12, e2036, https://doi.org/10.1002/eco.2036, 2019.

Borges, A. V., Abril, G., Darchambeau, F., Teodoru, C. R., Deborde, J., Vidal, L. O., Lambert, T., and Bouillon, S.: Divergent
biophysical controls of aquatic CO$_2$ and CH$_4$ in the World's two largest rivers, Sci. Rep., 5, 15614, https://doi.org/10.1038/srep15614, 2015a.

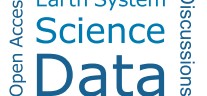

Borges, A. V., Darchambeau, F., Teodoru, C. R., Marwick, T. R., Tamooh, F., Geeraert, N., Omengo, F. O., Guérin, F., Lambert, T., Morana, C., Okuku, E., and Bouillon, S.: Globally significant greenhouse-gas emissions from African inland waters, Nat., Geosci, 8, 637–642, https://doi.org/10.1038/ngeo2486, 2015b.

Borges, A. V., Abril, G., and Bouillon, S.: Carbon dynamics and $CO_2$ and $CH_4$ outgassing in the Mekong Delta, Biogeosciences, 15, 1093–1114, https://doi.org/10.5194/bg-15-1093-2018, 2018a.

Borges, A. V., Darchambeau, F., Lambert, T., Bouillon, S., Morana, C., Brouyère, S., Hakoun, V., Jurado, A., Tseng, H.-C., Descy, J.-P., and Roland, F. A. E.: Effects of agricultural land use on fluvial carbon dioxide, methane and nitrous oxide concentrations in a large European river, the Meuse (Belgium), Sci. Total Environ., 610–611, 342–355,
https://doi.org/10.1016/j.scitotenv.2017.08.047, 2018b.

Borges, A. V. and Bouillon, S.: data-base of $CO_2$, $CH_4$, $N_2O$ and ancillary data in the Congo River (latest), Zenodo, https://doi.org/10.5281/ZENODO.3413449, 2019.

Bouillon, S., Abril, G., Borges, A. V., Dehairs, F., Govers, G., Hughes, H. J., Merckx, R., Meysman, F. J. R., Nyunja, J., Osburn, C., and Middelburg, J. J.: Distribution, origin and cycling of carbon in the Tana River (Kenya): a dry season basin-
scale survey from headwaters to the delta, Biogeosciences, 6, 2475–2493, https://doi.org/10.5194/bg-6-2475-2009, 2009.

Bouillon, S., Yambélé, A., Spencer, R. G. M., Gillikin, D. P., Hernes, P. J., Six, J., Merckx, R., and Borges, A. V.: Organic matter sources, fluxes and greenhouse gas exchange in the Oubangui River (Congo River basin), Biogeosciences, 9, 2045–2062, https://doi.org/10.5194/bg-9-2045-2012, 2012.

Bouillon, S., Yambélé, A., Gillikin, D. P., Teodoru, C., Darchambeau, F., Lambert, T., and Borges, A. V.: Contrasting
biogeochemical characteristics of the Oubangui River and tributaries (Congo River basin), Sci. Rep., 4, 5402, https://doi.org/10.1038/srep05402, 2015.

Bowden, W. B.: Arctic LTER streams chemistry Toolik Field Station, Alaska 1978 to 2019., Environmental Data Initiative, https://doi.org/10.6073/pasta/3faacd18b63b3bacc5a0dbd6f09660e1, 2021.

Bower, S.: Effects of storms on nitrate removal and greenhouse gas emissions from fluvial wetland dominated surface water
flow paths. University of New Hampshire., M.S. thesis, University of New Hampshire, Durham, NH, USA, https://scholars.unh.edu/thesis/1376/, 2020.

Bresney, S. R., Moseman-Valtierra, S., and Snyder, N. P.: Observations of greenhouse gases and nitrate concentrations in a Maine river and fringing wetland, Northeast. Nat., 22, 120–143, https://doi.org/10.1656/045.022.0125, 2015.




Bretz, K. A., Jackson, A. R., Rahman, S., Monroe, J. M., and Hotchkiss, E. R.: Integrating ecosystem patch contributions to
stream corridor carbon dioxide and methane fluxes, J Geophys Res Biogeosci, 126, e2021JG006313,
https://doi.org/10.1029/2021JG006313, 2021a.

Bretz, K. A., Jackson, A. R., Rahman, S., Monroe, J. M., and Hotchkiss, E. R.:
IntegratingEcosystemPatchContributionsStreamCO2CH4_Data,                                    HydroShare,
https://doi.org/10.4211/hs.ab2b33f27b3b4a0ca9a8ce4b8936753f, 2021b.

Brown, R. S. and Hershey, A. E.: Potential effects of the invasive bivalve *Corbicula fluminea* on methane cycling processes
in an urban stream, Biogeochemistry, 144, 181–195, https://doi.org/10.1007/s10533-019-00578-1, 2019.

Bugna, G. C., Chanton, J. P., Cable, J. E., Burnett, W. C., and Cable, P. H.: The importance of groundwater discharge to the
methane budgets of nearshore and continental shelf waters of the northeastern Gulf of Mexico, Geochim. Cosmochim. Acta,
60, 4735–4746, https://doi.org/10.1016/S0016-7037(96)00290-6, 1996.

Buriánková, I., Brablcová, L., Mach, V., Hýblová, A., Badurová, P., Cupalová, J., Čáp, L., and Rulík, M.: Methanogens and
methanotrophs distribution in the hyporheic sediments of a small lowland stream, Fundam. Appl. Limnol., 181, 87–102,
https://doi.org/10.1127/1863-9135/2012/0283, 2012.

Burns, R., Wynn, P. M., Barker, P., McNamara, N., Oakley, S., Ostle, N., Stott, A. W., Tuffen, H., Zhou, Z., Tweed, F. S.,
Chesler, A., and Stuart, M.: Direct isotopic evidence of biogenic methane production and efflux from beneath a temperate
glacier, Sci. Rep., 8, 17118, https://doi.org/10.1038/s41598-018-35253-2, 2018.

Burrows, R. M., van de Kamp, J., Bodrossy, L., Venarsky, M., Coates-Marnane, J., Rees, G., Jumppanen, P., and Kennard, M.
J.: Methanotroph community structure and processes in an inland river affected by natural gas macro-seeps, FEMS Microbiol.
Ecol., 97, fiab130, https://doi.org/10.1093/femsec/fiab130, 2021.

Bussmann, I.: Distribution of methane in the Lena Delta and Buor-Khaya Bay, Russia, Biogeosciences, 10, 4641–4652,
https://doi.org/10.5194/bg-10-4641-2013, 2013a.

Bussmann, I.: Methane concentration and isotopic composition ($\delta^{13}$C) in the waters of the Lena River and the Laptev Sea, in
the years 2008, 2009 and 2010, PANGAEA, https://doi.org/10.1594/PANGAEA.817302, 2013b.

Bussmann, I. and Fedorova, I. V: Dissolved methane concentrations under ice cover in the Lena Delta area, PANGAEA,
https://doi.org/10.1594/PANGAEA.905776, 2019.



Bussmann, I., Fedorova, I. V., Juhls, B., Overduin, P. P., and Winkel, M.: Dissolved methane concentrations and oxidation rates in the Lena Delta area, 2016-2018, 2 datasets, PANGAEA, https://doi.org/10.1594/PANGAEA.920015, 2020.

Bussmann, I., Fedorova, I. V., Juhls, B., Overduin, P. P., and Winkel, M.: Methane dynamics in three different Siberian water bodies under winter and summer conditions, Biogeosciences, 18, 2047–2061, https://doi.org/10.5194/bg-18-2047-2021, 2021.

Call, M., Sanders, C. J., Enrich-Prast, A., Sanders, L., Marotta, H., Santos, I. R., and Maher, D. T.: Metadata: Radon-traced
pore-water as a potential source of $CO_2$ and $CH_4$ to receding black and clear water environments in the Amazon Basin: Black and clear water environments in the Amazon Basin, figshare, https://figshare.com/s/95945208c33e8800f02c 2018a.

Call, M., Sanders, C. J., Enrich-Prast, A., Sanders, L., Marotta, H., Santos, I. R., and Maher, D. T.: Radon-traced pore-water as a potential source of $CO_2$ and $CH_4$ to receding black and clear water environments in the Amazon Basin: Black and clear water environments in the Amazon Basin, Limnol. Oceanogr. Lett., 3, 375–383, https://doi.org/10.1002/lol2.10089, 2018b.

Campeau, A.: Dataset for manuscript: Stable carbon isotopes reveal soil-stream DIC linkages in contrasting headwater catchments (Version 1.0), Uppsala University Publications, http://urn.kb.se/resolve?urn=urn:nbn:se:uu:diva-335488 2018.

Campeau, A., Lapierre, J.-F., Vachon, D., and del Giorgio, P. A.: Regional contribution of $CO_2$ and $CH_4$ fluxes from the fluvial network in a lowland boreal landscape of Québec: $CO_2$ and $CH_4$ emission from boreal rivers, Global Biogeochem. Cycles, 28, 57–69, https://doi.org/10.1002/2013GB004685, 2014.

Campeau, A., Bishop, K. H., Billett, M. F., Garnett, M. H., Laudon, H., Leach, J. A., Nilsson, M. B., Öquist, M. G., and Wallin, M. B.: Aquatic export of young dissolved and gaseous carbon from a pristine boreal fen: Implications for peat carbon stock stability, Global Change Biol, 23, 5523–5536, https://doi.org/10.1111/gcb.13815, 2017.

Campeau, A., Bishop, K., Nilsson, M. B., Klemedtsson, L., Laudon, H., Leith, F. I., Öquist, M., and Wallin, M. B.: Stable carbon isotopes reveal soil-stream DIC linkages in contrasting headwater catchments, J. Geophys. Res. Biogeosci., 123, 149–
167, https://doi.org/10.1002/2017JG004083, 2018.

Carey, C. C., Woelmer, W. M., Maze, J. T., and Hounshell, A. G.: Manually-collected discharge data for multiple inflow tributaries entering Falling Creek Reservoir and Beaverdam Reservoir, Vinton, Virginia, USA in 2019, Environmental Data Initiative, https://doi.org/10.6073/pasta/4d8e7b7bedbc6507b307ba2d5f2cf9a2, 2019.

Carey, C. C., Hounshell, A. G., Lofton, M. E., Birgand, F., Bookout, B. J., Corrigan, R. S., Gerling, A. B., McClure, R. P., and
Woelmer, W. M.: Discharge time series for the primary inflow tributary entering Falling Creek Reservoir, Vinton, Virginia, USA 2013-2021, Environmental Data Initiative, https://doi.org/10.6073/pasta/8d22a432aac5560b0f45aa1b21ae4746, 2021a.





Carey, C. C., Wander, H. L., Woelmer, W. M., Lofton, M. E., Breef-Pilz, A., Doubek, J. P., Gerling, A. B., Hounshell, A. G., McClure, R. P., and Niederlehner, B. R.: Water chemistry time series for Beaverdam Reservoir, Carvins Cove Reservoir, Falling Creek Reservoir, Gatewood Reservoir, and Spring Hollow Reservoir in southwestern Virginia, USA 2013-2020,
Environmental Data Initiative, https://doi.org/10.6073/pasta/8d83ef7ec202eca9192e3da6dd34a4e0, 2021b.

Carey, C. C., Hounshell, A. G., McClure, R. P., Gerling, A. B., Lewis, A. S. L., and Niederlehner, B. R.: Time series of dissolved methane and carbon dioxide concentrations for Falling Creek Reservoir and Beaverdam Reservoir in southwestern Virginia,    USA    during    2015-2021,    Environmental    Data    Initiative,
https://doi.org/10.6073/pasta/2fb836492aace4c13b7962f2718be8e5, 2022.

Carpenter, C., Wickland, K. P., Dornblaser, M. M., Clow, D. W., Koch, J. C., and Dantoin, E. D.: Wetland stream water quality data for West Twin Creek, AK, Allequash Creek, WI, and Big Thompson River, CO, 2010-2020: U.S. Geological Survey data release, https://doi.org/10.5066/p9wys23u, 2021.

Castro-Morales, K., Canning, A., Körtzinger, A., Göckede, M., Küsel, K., Overholt, W.A., Wichard, T., Redlich, S., Arzberger, S., Kolle, O., and Zimov, N.: Data published in manuscript "Effects of reversal of water flow in an Arctic floodplain river on
fluvial emissions of $CO_2$ and $CH_4$" by Castro-Morales et al., Zenodo, https://doi.org/10.5281/zenodo.5758728, 2021.

Castro-Morales, K., Canning, A., Körtzinger, A., Göckede, M., Küsel, K., Overholt, W. A., Wichard, T., Redlich, S., Arzberger, S., Kolle, O., and Zimov, N.: Effects of reversal of water flow in an Arctic floodplain river on fluvial emissions of $CO_2$ and $CH_4$, J. Geophys. Res. Biogeosci., 127, e2021JG006485, https://doi.org/10.1029/2021jg006485, 2022.

Chamberlain, S. D., Boughton, E. H., and Sparks, J. P.: Underlying ecosystem emissions exceed cattle-emitted methane from
subtropical lowland pastures, Ecosystems, 18, 933–945, https://doi.org/10.1007/s10021-015-9873-x, 2015.

Chan, C.-N., Shi, H., Liu, B., and Ran, L.: $CO_2$ and $CH_4$ emissions from an arid fluvial network on the Chinese Loess Plateau, Water, 13, 1614, https://doi.org/10.3390/w13121614, 2021.

Chen, C.-T. A., Wang, S.-L., Lu, X.-X., Zhang, S.-R., Lui, H.-K., Tseng, H.-C., Wang, B.-J., and Huang, H.-I.: Hydrogeochemistry and greenhouse gases of the Pearl River, its estuary and beyond, Quat. Int., 186, 79–90,
https://doi.org/10.1016/j.quaint.2007.08.024, 2008.

Chen, S., Wang, D., Ding, Y., Yu, Z., Liu, L., Li, Y., Yang, D., Gao, Y., Tian, H., Cai, R., and Chen, Z.: Ebullition controls on $CH_4$ emissions in an urban, eutrophic river: A potential time-scale bias in determining the aquatic $CH_4$ flux, Environ. Sci. Technol., 55, 7287–7298, https://doi.org/10.1021/acs.est.1c00114, 2021.



Clarizia, P. E.: Seasonal methane and carbon dioxide emissions along a temperate fluvial wetland dominated river continuum,
M.S. thesis, University of New Hampshire, Durham, NH, USA, 40 pp., https://scholars.unh.edu/thesis/1322/, 2019.

Clayer, F., Thrane, J. -E., Brandt, U., Dörsch, P., and Wit, H. A.: Boreal headwater catchment as hot spot of carbon processing
from headwater to fjord, J. Geophys. Res. Biogeosci., 126, e2021JG006359, https://doi.org/10.1029/2021JG006359, 2021a.

Clayer, F., Thrane, J. -E., Brandt, U., Dörsch, P., and de Wit, H.: Dataset for "Boreal headwater catchment as hot spot of
carbon     processing     from     headwater     to     fjord"     Clayer     et     al.,     HydroShare,
https://doi.org/10.4211/hs.b143432cfe72462fac90376631c9a2b3, 2021b.

Clilverd, H. M., Jones, J. B., and Kielland, K.: Nitrogen retention in the hyporheic zone of a glacial river in interior Alaska,
Biogeochemistry, 88, 31–46, https://doi.org/10.1007/s10533-008-9192-9, 2008.

Comer-Warner, S., Krause, S., Gooddy, D. C., Ullah, S., and Wexler, S. K.: Seasonal streambed carbon and nitrogen cycling
(including greenhouse gases) in an agriculturally-impacted stream. Measured at Wood Brook UK, 2016-2017, NERC
Environmental Information Data Centre, https://doi.org/10.5285/00601260-285e-4ffa-b381-340b51a7ec50, 2018.

Comer-Warner, S. A., Gooddy, D. C., Ullah, S., Glover, L., Percival, A., Kettridge, N., and Krause, S.: Seasonal variability of
sediment   controls   of   carbon   cycling   in   an   agricultural   stream,   Sci.   Total   Environ.,   688,   732–741,
https://doi.org/10.1016/j.scitotenv.2019.06.317, 2019.

Cotovicz, L. C., Ribeiro, R. P., Régis, C. R., Bernardes, M., Sobrinho, R., Vidal, L. O., Tremmel, D., Knoppers, B. A., and
Abril, G.: Greenhouse gas emissions ($CO_2$ and $CH_4$) and inorganic carbon behavior in an urban highly polluted tropical coastal
lagoon (SE, Brazil), Environ. Sci. Pollut. Res., 28, 38173–38192, https://doi.org/10.1007/s11356-021-13362-2, 2021.

Cowley, K., Looman, A., Maher, D. T., and Fryirs, K.: Geomorphic controls on fluvial carbon exports and emissions from
upland swamps in eastern Australia, Sci. Total Environ., 618, 765–776, https://doi.org/10.1016/j.scitotenv.2017.08.133, 2018.

Crawford, J. T. and Stanley, E. H.: Controls on methane concentrations and fluxes in streams draining human-dominated
landscapes, Ecol. Appl., 26, 1581–1591, https://doi.org/10.1890/15-1330, 2016.

Crawford, J. and Stanley, E.: Greenhouse gas emissions from streams at North Temperate Lakes LTER 2012, Environmental
Data Initiative, https://doi.org/10.6073/pasta/000c46ce5c2ded2cefbddb596fca3ce2, 2020.

Crawford, J. T., Striegl, R. G., Wickland, K. P., Dornblaser, M. M., and Stanley, E. H.: Emissions of carbon dioxide and
methane   from   a   headwater   stream   network   of   interior   Alaska,   J.   Geophys.   Res.   Biogeosci.,   118,   482–494,
https://doi.org/10.1002/jgrg.20034, 2013.



Crawford, J. T., Lottig, N. R., Stanley, E. H., Walker, J. F., Hanson, P. C., Finlay, J. C., and Striegl, R. G.: $CO_2$ and $CH_4$ emissions from streams in a lake-rich landscape: Patterns, controls, and regional significance, Global Biogeochem. Cycles, 28, 197–210, https://doi.org/10.1002/2013GB004661, 2014a.

Crawford, J. T., Stanley, E. H., Spawn, S. A., Finlay, J. C., Loken, L. C., and Striegl, R. G.: Ebullitive methane emissions from oxygenated wetland streams, Global Change Biol., 20, 3408–3422, https://doi.org/10.1111/gcb.12614, 2014b.

Crawford, J. T., Dornblaser, M. M., Stanley, E. H., Clow, D. W., and Striegl, R. G.: Source limitation of carbon gas emissions in high-elevation mountain streams and lakes, J. Geophys. Res. Biogeosci., 120, 952–964, https://doi.org/10.1002/2014JG002861, 2015.

Crawford, J. T., Loken, L. C., Stanley, E. H., Stets, E. G., Dornblaser, M. M., and Striegl, R. G.: Basin scale controls on $CO_2$ and $CH_4$ emissions from the Upper Mississippi River: Mississippi River Greenhouse Gases, Geophys. Res. Lett., 43, 1973–1979, https://doi.org/10.1002/2015GL067599, 2016.

Crawford, J., Stanley, E., and Loken, L.: Ebullitive methane emissions from oxygenated wetland streams at North Temperate Lakes LTER 2013, Environmental Data Initiative, https://doi.org/10.6073/pasta/7842765e6520b62af6fd4d95c838f177, 2020.

Dahm, C. N., Carr, D. L., and Coleman, R. L.: Anaerobic carbon cycling in stream ecosystems, SIL Proceedings, 1922-2010, 24, 1600-1604, https://doi.org/10.1080/03680770.1989.11899028, 1991.

Dawson, J. J. C., Billett, M. F., Hope, D., Palmer, S. M., and Deacon, C. M.: Sources and sinks of aquatic carbon in a peatland stream continuum, Biogeochemistry, 70, 71–92, https://doi.org/10.1023/B:BIOG.0000049337.66150.f1, 2004.

de Angelis, M. A. and Lilley, M. D.: Methane in surface waters of Oregon estuaries and rivers, Limnol. Oceanogr., 32, 716–925 722, https://doi.org/10.4319/lo.1987.32.3.0716, 1987.

de Angelis, M. A. and Scranton, M. I.: Fate of methane in the Hudson River and Estuary, Global Biogeochem. Cycles, 7, 509–523, https://doi.org/10.1029/93GB01636, 1993.

Dean, J. F., Billett, M. F., Baxter, R., Dinsmore, K. J., Lessels, J. S., Street, L. E., Subke, J.-A., Tetzlaff, D., Washbourne, I., and Wookey, P. A.: Biogeochemistry of "pristine" freshwater stream and lake systems in the western Canadian Arctic, 930 Biogeochemistry, 130, 191–213, https://doi.org/10.1007/s10533-016-0252-2, 2016.



Dean, J. F., Meisel, O. H., Martyn Rosco, M., Marchesini, L. B., Garnett, M. H., Lenderink, H., van Logtestijn, R., Borges, A. V., Bouillon, S., Lambert, T., Röckmann, T., Maximov, T., Petrov, R., Karsanaev, S., Aerts, R., van Huissteden, J., Vonk, J. E., and Dolman, A. J.: East Siberian Arctic inland waters emit mostly contemporary carbon, Nat. Commun, 11, 1627, https://doi.org/10.1038/s41467-020-15511-6, 2020.

Deirmendjian, L., Anschutz, P., Morel, C., Mollier, A., Augusto, L., Loustau, D., Cotovicz, L. C., Buquet, D., Lajaunie, K., Chaillou, G., Voltz, B., Charbonnier, C., Poirier, D., and Abril, G.: Importance of the vegetation-groundwater-stream continuum to understand transformation of biogenic carbon in aquatic systems – A case study based on a pine-maize comparison in a lowland sandy watershed (Landes de Gascogne, SW France), Sci. Total Environ., 661, 613–629, https://doi.org/10.1016/j.scitotenv.2019.01.152, 2019.

DelSontro, T., Perez, K. K., Sollberger, S., and Wehrli, B.: Methane dynamics downstream of a temperate run-of-the-river reservoir: CH$_4$ dynamics below a run-of-the-river reservoir, Limnol. Oceanogr., 61, S188–S203, https://doi.org/10.1002/lno.10387, 2016.

Deng, O., Li, X., Deng, L., Zhang, S., Gao, X., Lan, T., Zhou, W., Tian, D., Xiao, Y., Yang, J., Ou, D., and Luo, L.: Emission of CO$_2$ and CH$_4$ from a multi-ditches system in rice cultivation region: Flux, temporal-spatial variation and effect factors, J. 945     Environ. Manage., 270, 110918, https://doi.org/10.1016/j.jenvman.2020.110918, 2020.

Descloux, S., Chanudet, V., Serça, D., and Guérin, F.: Methane and nitrous oxide annual emissions from an old eutrophic temperate reservoir, Sci. Total Environ., 598, 959–972, https://doi.org/10.1016/j.scitotenv.2017.04.066, 2017.

Deshmukh, C., Guérin, F., Pighini, S., Vongkhamsao, A., Guédant, P., Rode, W., Godon, A., Chanudet, V., Descloux, S., and Serça, D.: Low methane (CH$_4$) emissions downstream of a monomictic subtropical hydroelectric reservoir (Nam Theun 2, Lao 950     PDR), Biogeosciences Discuss., 12, 11313–11347, https://doi.org/10.5194/bgd-12-11313-2015, 2015.

Dinsmore, K. J., Billett, M. F., Skiba, U. M., Rees, R. M., Drewer, J., and Helfter, C.: Role of the aquatic pathway in the carbon and greenhouse gas budgets of a peatland catchment, Global Change Biol., 16, 2750–2762, https://doi.org/10.1111/j.1365-2486.2009.02119.x, 2010.

Dinsmore, K. J., Billett, M. F., Dyson, K. E., Harvey, F., Thomson, A. M., Piirainen, S., and Kortelainen, P.: Stream water 955     hydrochemistry as an indicator of carbon flow paths in Finnish peatland catchments during a spring snowmelt event, Sci. Total Environ., 409, 4858–4867, https://doi.org/10.1016/j.scitotenv.2011.07.063, 2011.



Dinsmore, K. J., Billett, M. F., and Dyson, K. E.: Five year record of aquatic carbon and greenhouse gas concentrations from Auchencorth Moss, NERC Environmental Information Data Centre, https://doi.org/10.5285/3f0820a7-a8c8-4dd7-a058-8db79ba9c7fe, 2013a.

Dinsmore, K. J., Billett, M. F., and Dyson, K. E.: Temperature and precipitation drive temporal variability in aquatic carbon and GHG concentrations and fluxes in a peatland catchment, Global Change Biol, 19, 2133–2148, https://doi.org/10.1111/gcb.12209, 2013b.

Dinsmore, K. J., Murphey, O., Leith, F., and Carfrae, J.: Aquatic carbon and greenhouse gas concentrations in the Auchencorth Moss catchment following drain blocking, NERC Environmental Information Data Centre, https://doi.org/10.5285/88ffbf44-0ec0-41d6-9814-04bc3535cd84, 2016.

Dodd, A.: Flow regime influences on stream and riparian soil carbon dynamics in the Ozark Highlands and Boston Mountains of Arkansas, Ph.D., University of Arkansas, Fayetteville, AR, USA, 143 pp., https://scholarworks.uark.edu/etd/2911/, 2018.

Dornblaser, M. M. and Halm, D. R. (Eds.): Water and sediment quality of the Yukon River and its tributaries, from Eagle to St. Marys, Alaska, 2002–2003, U.S. Geological Survey Open-File Report 2006-1228, Reston, VA, 202 pp., https://doi.org/10.3133/ofr20061228, 2006.

Druschke, C. G.: Geomorphic and ecological effects of restoration and flood events on stream in the Driftless Area of Wisconsin, M.S. thesis, University of Wisconsin-Madison, Madison, WI, USA, 59 pp., https://search.library.wisc.edu/catalog/9913456953802121, 2021.

Druschke, C. G., Booth, E. G., and Stanley, E. H.: Stream restoration and flood impacts in the Kickapoo River Watershed, Wisconsin, 2019, Environmental Data Initiative, https://doi.org/10.6073/pasta/4905369228d80920974f555a2dd12229, 2022.

Dyson, K. E., Billett, M. F., Dinsmore, K. J., Harvey, F., Thomson, A. M., Piirainen, S., and Kortelainen, P.: Release of aquatic carbon from two peatland catchments in E. Finland during the spring snowmelt period, Biogeochemistry, 103, 125–142, https://doi.org/10.1007/s10533-010-9452-3, 2011.

Einarsdottir, K., Wallin, M. B., and Sobek, S.: High terrestrial carbon load via groundwater to a boreal lake dominated by surface water inflow: Carbon input to a lake via Groundwater, J. Geophys. Res. Biogeosci., 122, 15–29, https://doi.org/10.1002/2016JG003495, 2017.



Emmerton, C. A., St. Louis, V. L., Lehnherr, I., Humphreys, E. R., Rydz, E., and Kosolofski, H. R.: The net exchange of methane with high Arctic landscapes during the summer growing season, Biogeosciences, 11, 3095–3106, https://doi.org/10.5194/bg-11-3095-2014, 2014.

Evans, C. D., Peacock, M., Green, S. M., Holden, J., Chapman, P. J., Lebron, I., Callaghan, N., Grayson, R., and Baird, A. J.: The impact of ditch blocking on fluvial carbon export from a UK blanket bog, Hydrol. Process., 32, 2141–2154, https://doi.org/10.1002/hyp.13158, 2018.

Federov, Y. A., Khoroshevskaya, V. O., and Tambieva, N. S.: Variations in methane concentrations in the water of the Don River and Taganrog Bay under the effect of natural factors, Water Resour., 30, 89–93,
https://doi.org/10.1023/A:1022059903167, 2003.

Flanagan, L. B., Nikkel, D. J., Scherloski, L. M., Tkach, R. E., Smits, K. M., Selinger, L. B., and Rood, S. B.: Multiple processes contribute to methane emission in a riparian cottonwood forest ecosystem, New Phytol., 229, 1970-1982, https://doi.org/10.1111/nph.16977, 2020.

Flessa, H., Rodionov, A., Guggenberger, G., Fuchs, H., Magdon, P., Shibistova, O., Zrazhevskaya, G., Mikheyeva, N.,
Kasansky, O. A., and Blodau, C.: Landscape controls of $CH_4$ fluxes in a catchment of the forest tundra ecotone in northern Siberia, Global Change Biol.,, 14, 2040–2056, https://doi.org/10.1111/j.1365-2486.2008.01633.x, 2008.

Flury, S. and Ulseth, A. J.: Exploring the sources of unexpected high methane concentrations and fluxes from Alpine headwater streams, Geophys. Res. Lett., 46, 6614–6625, https://doi.org/10.1029/2019GL082428, 2019.

Foks, S. S., Stackpoole, S. M., Dornblaser, M. M., Whiddon, E. T., Breitmeyer, S. E., Campbell, D. A., Crawford, J. T.,
Metcalf, E. B., Stets, E. G., Uhle, B. A., Voss, B. M., Wickland, K. P., and Striegl, R. G.: Water quality, quantity, and gas fluxes of the Upper Mississippi River basin (WY 2012-2016), U.S. Geological Survey data release, https://doi.org/10.5066/F7P849S4, 2018.

Foks, S. S., Dornblaser, M. M., Butman, D. E., Campbell, D. A., Koch, J. C., Li, Z., Whiddon, E. T., Wickland, K. P., Striegl, R., Bogard, M. J., Spencer, R. G. M., Textor, S. R., and Johnston, S. E.: Water quality and gas fluxes of Interior Alaska (2014-
2018), U.S. Geological Survey data release, https://doi.org/10.5066/P9C6BDBQ, 2020.

Ford, T. E. and Naiman, R. J.: Alteration of carbon cycling by beaver: methane evasion rates from boreal forest streams and rivers, Can. J. Zool., 66, 529–533, https://doi.org/10.1139/z88-076, 1988.



Galantini, L., Lapierre, J.-F., and Maranger, R.: How are greenhouse gases coupled across seasons in a large temperate river with differential land use?, Ecosystems, https://doi.org/10.1007/s10021-021-00629-5, 2021.

Gareis, J. A. L. and Lesack, L. F. W.: Ice-out and freshet fluxes of $CO_2$ and $CH_4$ across the air–water interface of the channel network of a great Arctic delta, the Mackenzie, Polar Res., 39, https://doi.org/10.33265/polar.v39.3528, 2020.

Gar'kusha, D. N. and Fedorov, Y. A.: Methane in the water and bottom sediments of the mouth area of the Severnaya Dvina River during the winter time, Oceanology, 54, 160–169, https://doi.org/10.1134/S000143701402009X, 2014.

Gar'kusha, D. N. and Fedorov, Y. A.: Distribution of methane concentration in coastal areas of the Gulf of Petrozavodsk, Lake 1015 Onega, Water Resour, 42, 331–339, https://doi.org/10.1134/S0097807815030045, 2015.

Gar'kusha, D. N., Fedorov, Y. A., and Khromov, M. I.: Methane in the water and bottom sediments of the mouth area of the Severnaya Dvina River (White Sea), Oceanology, 50, 498–512, https://doi.org/10.1134/S0001437010040065, 2010a.

Gar'kusha, D. N., Fedorov, Y., and Khromov, M. I.: Methane in waters and bottom sediments of the North (Severnaya) Dvina mouth area in 2004-2006, PANGAEA, https://doi.pangaea.de/10.1594/PANGAEA.763838, 2010b.

Garnier, J., Vilain, G., Silvestre, M., Billen, G., Jehanno, S., Poirier, D., Martinez, A., Decuq, C., Cellier, P., and Abril, G.: Budget of methane emissions from soils, livestock and the river network at the regional scale of the Seine basin (France), Biogeochemistry, 116, 199–214, https://doi.org/10.1007/s10533-013-9845-1, 2013.

Gatland, J. R., Santos, I. R., Maher, D. T., Duncan, T. M., and Erler, D. V.: Carbon dioxide and methane emissions from an artificially drained coastal wetland during a flood: Implications for wetland global warming potential, J. Geophys. Res. 1025 Biogeosci., 119, 1698–1716, https://doi.org/10.1002/2013JG002544, 2014.

Gatti, R. C., Callaghan, T. V., Rozhkova-Timina, I., Dudko, A., Lim, A., Vorobyev, S. N., Kirpotin, S. N., and Pokrovsky, O. S.: The role of Eurasian beaver (*Castor fiber*) in the storage, emission and deposition of carbon in lakes and rivers of the River Ob flood plain, western Siberia, Sci. Total Environ., 644, 1371–1379, https://doi.org/10.1016/j.scitotenv.2018.07.042, 2018.

Gómez-Gener, L., Obrador, B., von Schiller, D., Marcé, R., Casas-Ruiz, J. P., Proia, L., Acuña, V., Catalán, N., Muñoz, I., 1030 and Koschorreck, M.: Hot spots for carbon emissions from Mediterranean fluvial networks during summer drought, Biogeochemistry, 125, 409–426, https://doi.org/10.1007/s10533-015-0139-7, 2015.

Gómez-Gener, L., Gubau, M., von Schiller, D., Marcé, R., and Obrador, B.: Effect of small water retention structures on diffusive $CO_2$ and $CH_4$ emissions along a highly impounded river, Inland Waters, 8, 449–460, https://doi.org/10.1080/20442041.2018.1457846, 2018.





Gondwe, M. J. and Masamba, W. R. L.: Spatial and temporal dynamics of diffusive methane emissions in the Okavango Delta, northern Botswana, Africa, Wetlands Ecol. Manage., 22, 63–78, https://doi.org/10.1007/s11273-013-9323-5, 2014.

Gong, X. J., Wang, X. F., Yuan, X. Z., Liu, T. T., and Hou, C. L.: Effects of field town developments on the dissolved and diffusion fluxes of greenhouse gasses in Heishuitan River basin, Chongqing, Acta Ecol. Sin., 39, 8425–8441, https://www.ecologica.cn/html/2019/22/stxb201810112205.htm, 2019.

Grieve, P. L., Hynek, S. A., Heilweil, V., Sowers, T., Llewellyn, G., Yoxtheimer, D., Solomon, D. K., and Brantley, S. L.: Using environmental tracers and modelling to identify natural and gas well-induced emissions of methane into streams, Appl. Geochem., 91, 107–121, https://doi.org/10.1016/j.apgeochem.2017.12.022, 2018.

Gu, C., Waldron, S., and Bass, A. M.: Carbon dioxide, methane, and dissolved carbon dynamics in an urbanized river system, Hydrol., Process., 35, https://doi.org/10.1002/hyp.14360, 2021.

Guo, Y., Song, C., Wang, L., Tan, W., Wang, X., Cui, Q., and Wan, Z.: Concentrations, sources, and export of dissolved $CH_4$ and $CO_2$ in rivers of the permafrost wetlands, northeast China, Ecol. Eng., 90, 491–497, https://doi.org/10.1016/j.ecoleng.2015.10.004, 2016.

Halm, D. R. and Dornblaser, M. M. (Eds.): Water and sediment quality in the Yukon River and its tributaries, between Atlin, British Columbia, Canada, and Eagle, Alaska, 2004, U.S. Geological Survey Open-File Report 2007-1197, Reston, VA, 120 1050 pp., https://doi.org/10.3133/ofr20071197, 2007.

Hamilton, S. K., Sippel, S., and Melack, J. M.: Oxygen depletion and carbon dioxide and methane production in waters of the Pantanal wetland of Brazil, Biogeochemistry, 30, 115–141, https://doi.org/10.1007/bf00002727, 1995.

Han, Y.: Greenhouse gases emission characteristics of rivers in Nanjing and the influencing factors, M.S. thesis, Nanjing University of Information Science and Technology, Nanjing, China, https://kns.cnki.net/kcms/detail/detail 1055 .aspx?dbname=cmfd201401&filename=1013340872.nh, 2013.

Han, Y., Zhang, G., and Zhao, Y.: Distribution and fluxes of methane in tropical rivers and lagoons of eastern Hainan, J. Trop. Oceanogr., 31, 87–95, http://www.jto.ac.cn/CN/10.11978/j.issn.1009-5470.2012.02.012, 2012.

Hao, X., Ruihong, Y., Zhuangzhuang, Z., Zhen, Q., Xixi, L., Tingxi, L., and Ruizhong, G.: Greenhouse gas emissions from the water–air interface of a grassland river: a case study of the Xilin River, Sci. Rep., 11, 2659, https://doi.org/10.1038/s41598-1060 021-81658-x, 2021.





Harms, T. K. and Godsey, S. E.: Collaborative Research: Climate-mediated coupling of hydrology and biogeochemistry in arctic hillslopes, Arctic Data Center, https://doi.org/10.18739/a23r0ps5p, 2018.

Harms, T. K., Rocher-Ros, G., and Godsey, S. E.: Emission of greenhouse gases from water tracks draining Arctic hillslopes, J. Geophys. Res. Biogeosci., 125, e2020JG005889, https://doi.org/10.1029/2020JG005889, 2020.

Harrison, J. A., Matson, P. A., and Fendorf, S. E.: Effects of a diel oxygen cycle on nitrogen transformations and greenhouse gas emissions in a eutrophied subtropical stream, Aquat. Sci., 67, 308–315, https://doi.org/10.1007/s00027-005-0776-3, 2005.

Hattenberger, D. M.: Comparing carbon dioxide and methane emissions from restored and unrestored sections of three North Carolina streams, M.S. thesis, North Carolina State University, Raliegh, N.C., USA, 103 pp., http://www.lib.ncsu.edu/resolver/1840.20/35725, 2018.

Hayakawa, A., Ikeda, S., Tsushima, R., Ishikawa, Y., and Hidaka, S.: Spatial and temporal variations in nutrients in water and riverbed sediments at the mouths of rivers that enter Lake Hachiro, a shallow eutrophic lake in Japan, Catena, 133, 486–494, https://doi.org/10.1016/j.catena.2015.04.009, 2015.

He, B., He, J., Wang, J., Li, J., and Wang, F.: Characteristics of GHG flux from water-air interface along a reclaimed water intake area of the Chaobai River in Shunyi, Beijing, Atmos. Environ., 172, 102–108, 1075 https://doi.org/10.1016/j.atmosenv.2017.10.060, 2018.

Heilweil, V. M., Stolp, B. J., Kimball, B. A., Susong, D. D., Marston, T. M., and Gardner, P. M.: A stream-based methane monitoring approach for evaluating groundwater impacts associated with unconventional gas development, Groundwater, 51, 511–524, https://doi.org/10.1111/gwat.12079, 2013.

Heilweil, V. M., Risser, D. W., Conger, R. W., Grieve, P. L., and Hyneck, S. A.: Estimation of methane concentrations and 1080 loads in groundwater discharge to Sugar Run, Lycoming County, Pennsylvania, U.S. Geological Survey Open File Report 2014–1126., Reston, VA, 31 p., https://doi.org/10.3133/ofr20141126, 2014.

Heilweil, V. M., Solomon, D. K., Darrah, T. H., Gilmore, T. E., and Genereux, D. P.: Gas-tracer experiment for evaluating the fate of methane in a coastal plain stream: Degassing versus in-stream oxidation, Environ. Sci. Technol., 50, 10504–10511, https://doi.org/10.1021/acs.est.6b02224, 2016.

Hendriks, D. M. D., van Huissteden, J., and Dolman, A. J.: Multi-technique assessment of spatial and temporal variability of methane fluxes in a peat meadow, Agric. For. Meteorol., 150, 757–774, https://doi.org/10.1016/j.agrformet.2009.06.017, 2010.



Heppell, C. M. and Binley, A.: Hampshire Avon: Daily discharge, stage and water chemistry data from four tributaries (Sem, Nadder, West Avon, Ebble), NERC Environmental Information Data Centre, https://doi.org/10.5285/0dd10858-7b96-41f1-8db5-e7b4c4168af5, 2016a.

Heppell, C. M. and Binley, A.: Hampshire Avon: Vertical head gradient, saturated hydraulic conductivity and pore water chemistry data from six river reaches, NERC Environmental Information Data Centre, https://doi.org/10.5285/d82a04ce-f04d-40b4-9750-1a2bf7dc29a3, 2016b.

Herman-Mercer, N. M.: Water-quality data from the Yukon River Basin in Alaska and Canada, U.S. Geological Survey data release, https://doi.org/10.5066/f77d2s7b, 2016.

Herreid, A. M., Wymore, A. S., Varner, R. K., Potter, J. D., and McDowell, W. H.: Divergent controls on stream greenhouse gas concentrations across a land-use gradient, Ecosystems, https://doi.org/10.1007/s10021-020-00584-7, 2020a.

Herreid, A. M., Wymore, A. S., Varner, R. K., Potter, J. D., and McDowell, W. H.: Divergent controls on stream greenhouse gas concentrations across a land use gradient, HydroShare, https://doi.org/10.4211/hs.2679ce1a6d514b30a54459893557dfe7, 2020b.

Herrero Ortega, S., Romero González-Quijano, C., Casper, P., Singer, G. A., and Gessner, M. O.: Methane emissions from contrasting urban freshwaters: Rates, drivers, and a whole-city footprint, Global Change Biol., 25, 4234–4243, https://doi.org/10.1111/gcb.14799, 2019.

Hlaváčová, E., Rulík, M., and Čáp, L.: Anaerobic microbial metabolism in hyporheic sediment of a gravel bar in a small lowland stream, River Res. Appl., 21, 1003–1011, https://doi.org/10.1002/rra.866, 2005.

Hlaváčová, E., Rulík, M., Čáp, L., and Mach, V.: Greenhouse gas ($CO_2$, $CH_4$, $N_2O$) emissions to the atmosphere from a small lowland stream in Czech Republic, Arch. Hydrobiol., 165, 339–353, https://doi.org/10.1127/0003-9136/2006/0165-0339, 2006.

Ho, L., Jerves-Cobo, R., Barthel, M., Six, J., Bode, S., Boeckx, P., and Goethals, P.: Dissolved gas concentrations in Cuenca river systems (Ecuador), Environmental Data Initiative, https://doi.org/10.6073/pasta/545502bf79b3e4e03bad2c5816375b01, 1110 2021.

Ho, L., Jerves-Cobo, R., Barthel, M., Six, J., Bode, S., Boeckx, P., and Goethals, P.: Greenhouse gas dynamics in an urbanized river system: influence of water quality and land use, Environ. Sci. Pollut. Res., https://doi.org/10.1007/s11356-021-18081-2, 2022.



Hope, D., Palmer, S. M., Billett, M. F., and Dawson, J. J. C.: Carbon dioxide and methane evasion from a temperate peatland stream, Limnol. Oceanogr., 46, 847–857, https://doi.org/10.4319/lo.2001.46.4.0847, 2001.

Hope, D., Palmer, S. M., Billett, M. F., and Dawson, J. J. C.: Variations in dissolved $CO_2$ and $CH_4$ in a first-order stream and catchment: an investigation of soil-stream linkages, Hydrol. Process., 18, 3255–3275, https://doi.org/10.1002/hyp.5657, 2004.

Hu, B., Wang, D., Zhou, J., Meng, W., Li, C., Sun, Z., Guo, X., and Wang, Z.: Greenhouse gases emission from the sewage draining rivers, Sci. Total Environ., 612, 1454–1462, https://doi.org/10.1016/j.scitotenv.2017.08.055, 2018.

Hu, J.: Greenhouse gas emissions and spatiotemporal variation from rivers in the Zoige Plateau, M.S. thesis, Northwest Agriculture & Forestry University, Xianyang, China, 56 pp., https://kns.cnki.net/KCMS/detail/detail.aspx?dbname=CMFD201601&filename=1015333482.nh, 2015.

Huang, Y., Yasarer, L. M. W., Li, Z., Sturm, B. S. M., Zhang, Z., Guo, J., and Shen, Y.: Air–water $CO_2$ and $CH_4$ fluxes along a river–reservoir continuum: Case study in the Pengxi River, a tributary of the Yangtze River in the Three Gorges Reservoir, China, Environ. Monit. Assess., 189, 223, https://doi.org/10.1007/s10661-017-5926-2, 2017.

Huotari, J., Nykänen, H., Forsius, M., and Arvola, L.: Effect of catchment characteristics on aquatic carbon export from a boreal catchment and its importance in regional carbon cycling, Global Change Biol., 19, 3607–3620, https://doi.org/10.1111/gcb.12333, 2013.

Hutchins, R. H. S., Prairie, Y. T., and del Giorgio, P. A.: Large-scale landscape drivers of $CO_2$, $CH_4$, DOC, and DIC in boreal river networks, Global Biogeochem. Cycles, 33, 125–142, https://doi.org/10.1029/2018GB006106, 2019.

Hutchins, R. H. S., Tank, S. E., Olefeldt, D., Quinton, W. L., Spence, C., Dion, N., Estop-Aragonés, C., and Mengistu, S. G.: Fluvial $CO_2$ and $CH_4$ patterns across wildfire-disturbed ecozones of subarctic Canada: Current status and implications for future change, Global Change Biol., 26, 2304–2319, https://doi.org/10.1111/gcb.14960, 2020a.

Hutchins, R. H. S., Casas-Ruiz, J. P., Prairie, Y. T., and del Giorgio, P. A.: Magnitude and drivers of integrated fluvial network greenhouse gas emissions across the boreal landscape in Québec, Water Res., 173, 115556, https://doi.org/10.1016/j.watres.2020.115556, 2020b.

Hutchins, R. H. S., Prairie, Y. T., and del Giorgio, P. A.: The relative importance of seasonality versus regional and network-specific properties in determining the variability of fluvial $CO_2$, $CH_4$ and dissolved organic carbon across boreal Québec, Aquat. Sci., 83, 72, https://doi.org/10.1007/s00027-021-00830-7, 2021.



Hyvonen, N. P., Huttunen, J. T., Shurpali, N. J., Lind, S. E., Marushchek, M. E., Heito, L., and Martikainen, P. J.: The role of drainage ditches in greenhouse gas emissions and surface leaching losses from a cutaway peatland cultivated with a perennial bioenergy crop, Boreal Environ. Res., http://hdl.handle.net/10138/229311, 18, 109–126, 2013.

Jauhiainen, J. and Silvennoinen H: Diffusion GHG fluxes at tropical peatland drainage canal water surfaces, Suo, 63, 93–105, http://www.suo.fi/article/9882, 2012.

Jin, H., Yoon, T. K., Begum, M. S., Lee, E.-J., Oh, N.-H., Kang, N., and Park, J.-H.: Longitudinal discontinuities in riverine greenhouse gas dynamics generated by dams and urban wastewater, Biogeosciences, 15, 6349–6369, https://doi.org/10.5194/bg-15-6349-2018, 2018.

Jones, J. B. and Mulholland, P. J.: Influence of drainage basin topography and elevation on carbon dioxide and methane supersaturation of stream water, Biogeochemistry, 40, 57–72, https://doi.org/10.1023/A:1005914121280, 1998a.

Jones, J. B. and Mulholland, P. J.: Methane input and evasion in a hardwood forest stream: Effects of subsurface flow from shallow and deep pathway, Limnol. Oceanogr., 43, 1243–1250, https://doi.org/10.4319/lo.1998.43.6.1243, 1998b.

Jurado, A., Borges, A. V., Pujades, E., Briers, P., Nikolenko, O., Dassargues, A., and Brouyère, S.: Dynamics of greenhouse gases in the river–groundwater interface in a gaining river stretch (Triffoy catchment, Belgium), Hydrogeol. J., 26, 2739–2751, https://doi.org/10.1007/s10040-018-1834-y, 2018.

Juutinen, S., Väliranta, M., Kuutti, V., Laine, A. M., Virtanen, T., Seppä, H., Weckström, J., and Tuittila, E.-S.: Short-term and long-term carbon dynamics in a northern peatland-stream-lake continuum: A catchment approach, J. Geophys. Res. Biogeosci., 118, 171–183, https://doi.org/10.1002/jgrg.20028, 2013.

Karlsson, J., Serikova, S., Vorobyev, S. N., Rocher-Ros, G., Denfeld, B. A., and Pokrovsky, O. S.: Carbon emission from Western Siberian inland waters, Zenodo, https://doi.org/10.5281/ZENODO.4153050, 2020.

Kemenes, A., Forsberg, B. R., and Melack, J. M.: Methane release below a tropical hydroelectric dam, Geophys. Res. Lett., 34, L12809, https://doi.org/10.1029/2007GL029479, 2007.

Kemenes, A., Forsberg, B. R., and Melack, J. M.: LBA-ECO LC-07 Methane and Carbon Dioxide emissions from Balbina Reservoir, Brazil, ORNL DAAC, http://daac.ornl.gov/cgi-bin/dsviewer.pl?ds_id=1143, 2013.

Kemenes, A., Forsberg, B. R., and Melack, J. M.: Downstream emissions of $CH_4$ and $CO_2$ from hydroelectric reservoirs (Tucuruí, Samuel, and Curuá-Una) in the Amazon basin, Inland Waters, 6, 295–302, https://doi.org/10.1080/IW-6.3.980, 2016.





Klaus, M., Geibrink, E., Jonsson, A., Bergström, A.-K., Bastviken, D., Laudon, H., Klaminder, J., and Karlsson, J.: Greenhouse gas emissions from boreal inland waters unchanged after forest harvesting, Biogeosciences, 15, 5575–5594, https://doi.org/10.5194/bg-15-5575-2018, 2018.

Kling, G.: Chemistry from thermokarst impacted soils, lakes, and streams near Toolik Lake Alaska, 2008-2011., Environmental Data Initiative, https://doi.org/10.6073/pasta/2e55d1587290e642938ac1a6caed6ec6, 2013.

Kling, G.: Toolik Inlet discharge data collected in summer 1993, Arctic LTER, Toolik Research Station, Alaska., Environmental Data Initiative, https://doi.org/10.6073/pasta/ae3cf97a2496946fa8ba0cf964271e56, 2016a.

Kling, G.: Toolik Inlet discharge data collected in summer 1994, Arctic LTER, Toolik Research Station, Alaska.,
Environmental Data Initiative, https://doi.org/10.6073/pasta/8cc384d957477d5ad48e926ed26dc89b, 2016b.

Kling, G.: Toolik Inlet discharge data collected in summer 1995, Arctic LTER, Toolik Research Station, Alaska., Environmental Data Initiative, https://doi.org/10.6073/pasta/20e10e53cc8b68cffbe98ed0b234d26a, 2016c.

Kling, G.: Toolik Inlet discharge data collected in summer 1996, Arctic LTER, Toolik Research Station, Alaska., Environmental Data Initiative, https://doi.org/10.6073/pasta/6e9d9bd807d8ec133e91d0e665a1550d, 2016d.

Kling, G.: Toolik Inlet discharge data collected in summer 1997, Arctic LTER, Toolik Research Station, Alaska., Environmental Data Initiative, https://doi.org/10.6073/pasta/33f027ad109d650964a0a084e5df7b11, 2016e.

Kling, G.: Toolik Inlet discharge data collected in summer 1998, Arctic LTER, Toolik Research Station, Alaska., Environmental Data Initiative, https://doi.org/10.6073/pasta/4b78d41f1462c952140b6d2bd4c5d3e4, 2016f.

Kling, G.: Toolik Inlet discharge data collected in summer 1999, Arctic LTER, Toolik Research Station, Alaska.,
Environmental Data Initiative, https://doi.org/10.6073/pasta/37c5b37970b78525819480aa7e4db43a, 2016g.

Kling, G.: Toolik Inlet discharge data collected in summer 2000, Arctic LTER, Toolik Research Station, Alaska., Environmental Data Initiative, https://doi.org/10.6073/pasta/48d71932248e540223bd5650902dd7a4, 2016h.

Kling, G.: Toolik Inlet discharge data collected in summer 2001, Arctic LTER, Toolik Research Station, Alaska., Environmental Data Initiative, https://doi.org/10.6073/PASTA/4EA8FA2D3B89F4BF2B5DE7B98B6A772C, 2016i.

Kling, G.: Toolik Inlet discharge data collected in summer 2002, Arctic LTER, Toolik Research Station, Alaska., Environmental Data Initiative, https://doi.org/10.6073/pasta/aa535873109be90a8a1cb133b45dbc67, 2016j.



Kling, G.: Toolik Inlet discharge data collected in summer 2003, Arctic LTER, Toolik Research Station, Alaska., Environmental Data Initiative, https://doi.org/10.6073/pasta/07d2ff982627a2a73343c1785358d0a6, 2016k.

Kling, G.: Toolik Inlet discharge data collected in summer 2004, Arctic LTER, Toolik Research Station, Alaska., Environmental Data Initiative, https://doi.org/10.6073/pasta/05f608cdb85f2e558febd0fd399da5cf, 2016l.

Kling, G.: Toolik Inlet discharge data collected in summer 2005, Arctic LTER, Toolik Research Station, Alaska., Environmental Data Initiative, https://doi.org/10.6073/pasta/9dde811179666deedd0ecf911be39f65, 2016m.

Kling, G.: Toolik Inlet discharge data collected in summer 2006, Arctic LTER, Toolik Research Station, Alaska., Environmental Data Initiative, https://doi.org/10.6073/pasta/bd8a06d5dab8691912524db28cc24bcd, 2016n.

Kling, G.: Toolik Inlet discharge data collected in summer 2007, Arctic LTER, Toolik Research Station, Alaska., Environmental Data Initiative, https://doi.org/10.6073/pasta/3af4cbab73c38f76b2829c3abff8f703, 2016o.

Kling, G.: Toolik Inlet discharge data collected in summer 2008, Arctic LTER, Toolik Research Station, Alaska., Environmental Data Initiative, https://doi.org/10.6073/pasta/48e780b581b1071f19c7e5f4b165035d, 2016p.

Kling, G.: Toolik Inlet discharge data collected in summer 2009, Arctic LTER, Toolik Research Station, Alaska., Environmental Data Initiative, https://doi.org/10.6073/pasta/94bb7d7a93a46ab5363033de6ee7d603, 2016q.

Kling, G.: Tussock Watershed stream discharge, electrical conductivity, and temperature measurements from 1992, Environmental Data Initiative, https://doi.org/10.6073/pasta/1e224958e278841f9a7a035007c65f21, 2016r.

Kling, G.: Tussock Watershed stream discharge, electrical conductivity, and temperature measurements from 1993, Environmental Data Initiative, https://doi.org/10.6073/pasta/f14f444ce51fa77d5f577db4cdbb0564, 2016s.

Kling, G.: Tussock Watershed stream discharge, electrical conductivity, and temperature measurements from 1994, Environmental Data Initiative, https://doi.org/10.6073/pasta/88124e3e8b4a8bbbd49fbb64d64b62d3, 2016t.

Kling, G.: Tussock Watershed stream discharge, electrical conductivity, and temperature measurements from 1995, Environmental Data Initiative, https://doi.org/10.6073/pasta/7e79c3adc44e965240f1c9d75ea676fb, 2016u.

Kling, G.: Tussock Watershed stream discharge, electrical conductivity, and temperature measurements from 1996, Environmental Data Initiative, https://doi.org/10.6073/pasta/6bd568dba3bfaa58181cfb8abff4d639, 2016v.





Kling, G.: Tussock Watershed stream discharge, electrical conductivity, and temperature measurements from 1997, Environmental Data Initiative, https://doi.org/10.6073/pasta/4c9e9b2bb4861e73dfeaa6bb5e8fb9cd, 2016w.

Kling, G.: Tussock Watershed stream discharge, electrical conductivity, and temperature measurements from 1999, Environmental Data Initiative, https://doi.org/10.6073/pasta/4b943b5a2de08aca8b7dd48542476f12, 2016x.

Kling, G.: Tussock Watershed stream discharge, electrical conductivity, and temperature measurements from 2000, Environmental Data Initiative, https://doi.org/10.6073/pasta/53a45c5a110f0af13c5ae0ed3154b8ca, 2016y.

Kling, G.: Tussock Watershed stream discharge, electrical conductivity, and temperature measurements from 2002, Environmental Data Initiative, https://doi.org/10.6073/pasta/5c3e5f2495561903c027c6b06544cf70, 2016z.

Kling, G.: Tussock Watershed stream discharge, electrical conductivity, and temperature measurements from 2003, 1225    Environmental Data Initiative, https://doi.org/10.6073/pasta/b24b8bb901a4b1b825e09c7ab494b39d, 2016aa.

Kling, G.: Tussock Watershed stream discharge, electrical conductivity, and temperature measurements from 2004, Environmental Data Initiative, https://doi.org/10.6073/pasta/459c62f862e1724005eb7d91648bfb44, 2016ab.

Kling, G. W.: Biogeochemistry data set for Imnavait Creek Weir on the North Slope of Alaska 2002-2018, Environmental Data Initiative,https://doi.org/10.6073/pasta/733c73c6ebffeaec6970b2b0f4dddfe6, 2019a.

Kling, G.: Biogeochemistry data set for soil waters, streams, and lakes near Toolik on the North Slope of Alaska., Environmental Data Initiative, https://doi.org/10.6073/pasta/574fd24522eee7a0c07fc260ccc0e2fa, 2019b.

Kling, G.: Toolik Lake Inlet discharge data collected during summers of 2010 to 2018, Arctic LTER, Toolik Research Station, Alaska., Environmental Data Initiative, https://doi.org/10.6073/pasta/169d1bae55373c44a368727573ef70eb, 2019c

Kling, G. 2022. Biogeochemistry data set for soil waters, streams, and lakes near Toolik Lake on the North Slope of Alaska, 2012 through 2020 ver. 2, Environmental Data Initiative, https://doi.org/10.6073/pasta/4e25db9ae9372f5339f2795792814845, 2022.

Kling, G. W., Kipphut, G. W., and Miller, M. C.: The flux of $CO_2$ and $CH_4$ from lakes and rivers in arctic Alaska, Hydrobiologia, 240, 23–36, https://doi.org/10.1007/BF00013449, 1992.

Köhn, D., Welpelo, C., Günther, A., and Jurasinski, G.: Drainage ditches contribute considerably to the $CH_4$ budget of a drained and a rewetted temperate fen, Wetlands, 41, 71, https://doi.org/10.1007/s13157-021-01465-y, 2021.





Kokic, J., Sahlée, E., Sobek, S., Vachon, D., and Wallin, M. B.: High spatial variability of gas transfer velocity in streams revealed by turbulence measurements, Inland Waters, 8, 461–473, https://doi.org/10.1080/20442041.2018.1500228, 2018.

Koné, Y. J. M., Abril, G., Delille, B., and Borges, A. V.: Seasonal variability of methane in the rivers and lagoons of Ivory Coast (West Africa), Biogeochemistry, 100, 21–37, https://doi.org/10.1007/s10533-009-9402-0, 2010.

Krickov, I. V., Serikova, S., Pokrovsky, O. S., Vorobyev, S. N., Lim, A. G., Siewert, M. B., and Karlsson, J.: Dataset for Ob River floodplain, Zenodo, https://doi.org/10.5281/ZENODO.4563905, 2021a.

Krickov, I. V., Serikova, S., Pokrovsky, O. S., Vorobyev, S. N., Lim, A. G., Siewert, M. B., and Karlsson, J.: Sizable carbon emission from the floodplain of Ob River, Ecol. Indic., 131, 108164, https://doi.org/10.1016/j.ecolind.2021.108164, 2021b.

Kuhn, C., Bettigole, C., Glick, H. B., Seegmiller, L., Oliver, C. D., and Raymond, P.: Patterns in stream greenhouse gas dynamics from mountains to plains in northcentral Wyoming: Aquatic GHG emissions in Wyoming, J. Geophys. Res. Biogeosci., 122, 2173–2190, https://doi.org/10.1002/2017JG003906, 2017.

LaboMaranger: LaboMaranger/RdN_GHG: data available, Zenodo, https://doi.org/10.5281/ZENODO.4566377, 2021.

Laini, A., Bartoli, M., Castaldi, S., Viaroli, P., Capri, E., and Trevisan, M.: Greenhouse gases ($CO_2$, $CH_4$ and $N_2O$) in lowland springs within an agricultural impacted watershed (Po River Plain, northern Italy), Chem. Ecol., 27, 177–187, https://doi.org/10.1080/02757540.2010.547489, 2011.

Lamarche-Gagnon, G., Wadham, J. L., Sherwood Lollar, B., Arndt, S., Fietzek, P., Beaton, A. D., Tedstone, A. J., Telling, J., Bagshaw, E. A., Hawkings, J. R., Kohler, T. J., Zarsky, J. D., Mowlem, M. C., Anesio, A. M., and Stibal, M.: Greenland melt drives continuous export of methane from the ice-sheet bed, Nature, 565, 73–77, https://doi.org/10.1038/s41586-018-0800-0, 2019.

Lamontagne, R. A., Swinnerton, J. W., Linnenbom, V. J., and Smith, W. D.: Methane concentrations in various marine environments, J. Geophys. Res., 78, 5317–5324, https://doi.org/10.1029/JC078i024p05317, 1973.

Leng, P., Kamjunke, N., Li, F., and Koschorreck, M.: Long-term monitoring $CO_2$ and $CH_4$ concentrations from two German streams, figshare, https://doi.org/10.6084/M9.FIGSHARE.12866945.V1, 2020.

Leng, P., Kamjunke, N., Li, F., and Koschorreck, M.: Temporal patterns of methane emissions from two streams with different riparian connectivity, J. Geophys. Res. Biogeosci., 126, e2020JG006104, https://doi.org/10.1029/2020jg006104, 2021.



Li, L., Yan, R., and Xue, B.: Methane levels of a river network in Wuxi City, China and response to water governance, Water, 12, 2617, https://doi.org/10.3390/w12092617, 2020a.

Li, X., Yao, H., Yu, Y., Cao, Y., and Tang, C.: Greenhouse gases in an urban river: Trend, isotopic evidence for underlying processes, and the impact of in-river structures, J. Hydrol., 591, 125290, https://doi.org/10.1016/j.jhydrol.2020.125290, 2020b.

Liang, X., Zhang, X., Sun, Q., He, C., Chen, X., Liu, X., and Chen, Z.: The role of filamentous algae *Spirogyra* spp. in methane production and emissions in streams, Aquat. Sci., 78, 227–239, https://doi.org/10.1007/s00027-015-0419-2, 2016.

Lilley, M. D., de Angelis, M. A., and Olson, E. J.: Methane concentrations and estimated fluxes from Pacific Northwest rivers, SIL Communications, 1953-1996, 25, 187–196, https://doi.org/10.1080/05384680.1996.11904080, 1996.

Liu, K., Hu, Z., Wei, X., Jiang, Z., Lu, H., and Wang, C.: Research on methane flux production flux in urban black and odourous rivers in summer: Taking Chaoyang Creek in Nanning City as an example, Earth Environ., 43, 415–419, doi: (China): 10.14050/j.cnki1672-9250/2015.04.005, 2015.

Loken, L., Crawford, J., Stanley, E., Butman, D., and Striegl, R.: Columbia River spatial water chemistry, Environmental Data Initiative, https://doi.org/10.6073/pasta/e881070c9e8f6b7f774d3c65b27a9f69, 2018a.

Loken, L., Crawford, J., and Stanley, E.: Mississippi River spatial water chemistry Environmental Research Letters datasets, https://doi.org/10.6073/pasta/c1b9dbd9a96edfb5e39a94cfef2982b9, 2018b.

Looman, A., Maher, D. T., Pendall, E., Bass, A., and Santos, I. R.: The carbon dioxide evasion cycle of an intermittent first-order stream: contrasting water–air and soil–air exchange, Biogeochemistry, 132, 87–102, https://doi.org/10.1007/s10533-016-0289-2, 2017.

Looman, A., Santos, I. R., Tait, D. R., Webb, J., Holloway, C., and Maher, D. T.: Dissolved carbon, greenhouse gases, and δ13C dynamics in four estuaries across a land use gradient, Aquat. Sci., 81, 22, https://doi.org/10.1007/s00027-018-0617-9, 2019.

Lottig, N. and Stanley, E.: North Temperate Lakes LTER: Northern Highlands stream chemistry survey 2006 ver. 18, Environmental Data Initiative, https://doi.org/10.6073/pasta/a73255a67041a580601711c9ac761f26, 2022.

Lottig, N. R., Stanley, E. H., and Maxted, J. T.: Assessing the influence of upstream drainage lakes on fluvial organic carbon in a wetland-rich region, J. Geophys. Res., 117, G03011, https://doi.org/10.1029/2012JG001983, 2012.





Ludwig, S., Holmes, R. M., Natali, S., Mann, P., Schade, J., Jimmie, J., Bristol, E., Peter, D., and Dabrowski, J.: Polaris Project 2017: Aquatic isotopes, carbon, and nitrogen, Yukon-Kuskokwim Delta, Alaska, Arctic Data Center, https://doi.org/10.18739/A20298, 2018a.

Ludwig, S., Holmes, R. M., Natali, S., Schade, J., and Mann, P.: Yukon-Kuskokwim Delta fire: aquatic data, Yukon-Kuskokwim Delta Alaska, 2015-2016, Arctic Data Center, https://doi.org/10.18739/A2HG45, 2018b.

Ludwig, S. M., Natali, S. M., Mann, P. J., Schade, J. D., Holmes, R. M., Powell, M., Fiske, G., and Commane, R.: Using machine learning to predict inland aquatic $CO_2$ and $CH_4$ concentrations and the effects of wildfires in the Yukon-Kuskokwim Delta, Alaska, Global Biogeochem. Cycles, 36, e2021GB007146, https://doi.org/10.1029/2021GB007146, 2022.

1300 Ma, P., Li, Y., Qi, L., Tian, X., and Yang, L.: Characteristics of $CH_4$ emission from Shaying River and its influence factors, Environ. Sci. Technol. (China), 40, 26–30, doi (China): 10.3969/j.issn.1003-6504.2017.02.005, 2017.

Mach, V., Bednařík, A., Čáp, L., Šipoš, J., and Rulík, M.: Seasonal measurement of greenhouse gas concentrations and emissions along the longitudinal profile of a small stream, Pol. J. Environ. Stud., 25, 2047–2056, https://doi.org/10.15244/pjoes/61668, 2016.

Maeck, A., DelSontro, T., McGinnis, D. F., Fischer, H., Flury, S., Schmidt, M., Fietzek, P., and Lorke, A.: Sediment trapping by dams creates methane emission hot spots, Environ. Sci. Technol., 47, 8130–8137, https://doi.org/10.1021/es4003907, 2013.

Maier, M.-S., Teodoru, C. R., and Wehrli, B.: Spatio-temporal variations of lateral and atmospheric carbon fluxes from the Danube Delta (dataset): A 2-year dataset of measured concentrations and fluxes, ETH Research Collection, https://doi.org/10.3929/ethz-b-000416925, 2020.

Maier, M.-S., Teodoru, C. R., and Wehrli, B.: Spatio-temporal variations in lateral and atmospheric carbon fluxes from the Danube Delta, Biogeosciences, 18, 1417–1437, https://doi.org/10.5194/bg-18-1417-2021, 2021.

Manning, C. C. M., Preston, V. L, Jones, S. F, Michel, A. P. M., Nicholson, D. P., Duke, P. J., Ahmed, M. M. M., Manganini, K., Else, B. G. T., and Tortell, P. D.: Dissolved methane and nitrous oxide concentrations measured on discrete bottle samples in Cambridge Bay, Nunavut, Canada from 2017, PANGAEA, https://doi.org/10.1594/PANGAEA.907149, 2019a.

Manning, C. C. M., Preston, V. L, Jones, S. F, Michel, A. P. M., Nicholson, D. P., Duke, P. J., Ahmed, M. M. M., Manganini, K., Else, B. G. T., and Tortell, P. D.: Dissolved methane and nitrous oxide concentrations measured on discrete bottle samples in Cambridge Bay, Nunavut, Canada from 2018, PANGAEA, https://doi.org/10.1594/PANGAEA.907150, 2019b.



Manning, C. C., Preston, V. L., Jones, S. F., Michel, A. P. M., Nicholson, D. P., Duke, P. J., Ahmed, M. M. M., Manganini, K., Else, B. G. T., and Tortell, P. D.: River inflow dominates methane emissions in an Arctic coastal system, Geophys. Res. Lett., 47, e2020GL087669, https://doi.org/10.1029/2020gl087669, 2020.

Manning, F. C.: Carbon dynamics in oil palm agro-ecosystems, Ph.D. thesis, University of Aberdeen, Aberdeen, UK, 435 pp., https://abdn.alma.exlibrisgroup.com/discovery/delivery/44ABE_INST:44ABE_VU1/12152929810005941, 2019a.

Manning, F. C., Kho, L. K., Hill, T. C., Cornulier, T., and Teh, Y. A.: Carbon emissions from oil palm plantations on peat soil, Front. For. Glob. Change, 2, 37, https://doi.org/10.3389/ffgc.2019.00037, 2019b.

Marescaux, A., Thieu, V., and Garnier, J.: Carbon dioxide, methane and nitrous oxide emissions from the human-impacted Seine watershed in France, Sci. Total Environ., 643, 247–259, https://doi.org/10.1016/j.scitotenv.2018.06.151, 2018.

Martin, J. B., Pain, A. J., Martin, E. E., Rahman, S., and Ackerman, P.: Comparisons of nutrients exported from Greenlandic glacial and deglaciated watersheds, Global Biogeochem. Cycles, 34, e2020GB006661, https://doi.org/10.1029/2020GB006661, 2020.

Martinez-Cruz, K., Gonzalez-Valencia, R., Sepulveda-Jauregui, A., Plascencia-Hernandez, F., Belmonte-Izquierdo, Y., and Thalasso, F.: Methane emission from aquatic ecosystems of Mexico City, Aquat. Sci., 79, 159–169, https://doi.org/10.1007/s00027-016-0487-y, 2017.

Marwick, T. R., Tamooh, F., Ogwoka, B., Teodoru, C., Borges, A. V., Darchambeau, F., and Bouillon, S.: Dynamic seasonal nitrogen cycling in response to anthropogenic N loading in a tropical catchment, Athi–Galana–Sabaki River, Kenya, Biogeosciences, 11, 443–460, https://doi.org/10.5194/bg-11-443-2014, 2014.

Marwick, T. R., Tamooh, F., Ogwoka, B., Borges, A. V., Darchambeau, F., and Bouillon, S.: A comprehensive biogeochemical record and annual flux estimates for the Sabaki River (Kenya), Biogeosciences, 15, 1683–1700, https://doi.org/10.5194/bg-15-1683-2018, 2018.

Matoušů, A., Rulík, M., Tušer, M., Bednařík, A., Šimek, K., and Bussmann, I.: Methane dynamics in a large river: a case study of the Elbe River, Aquat. Sci., 81, 12, https://doi.org/10.1007/s00027-018-0609-9, 2019.

Matveev, A., Blais, M. A., Laurion, I., and Vincent, W. F.: Dissolved methane, carbon dioxide and limnological data from subarctic rivers, northern Québec, Canada, v. 1.0 (2019-2019), Nordicana D, https://doi.org/10.5885/45660CE-8B92339884C146D0, 2020.



McGinnis, D. F., Bilsley, N., Schmidt, M., Fietzek, P., Bodmer, P., Premke, K., Lorke, A., and Flury, S.: Deconstructing
methane emissions from a small northern European river: Hydrodynamics and temperature as key drivers, Environ. Sci.
Technol., 50, 11680–11687, https://doi.org/10.1021/acs.est.6b03268, 2016.

Mei, D., Ni, M., Liang, X., Hou, L., Wang, F., and He, C.: Filamentous green algae *Spirogyra* regulates methane emissions
from eutrophic rivers, Environ. Sci. Pollut. Res., 28, 3660–3671, https://doi.org/10.1007/s11356-020-10754-8, 2021.

Middelburg, J. J., Nieuwenhuize, J., Iversen, N., Høgh, N., de Wilde, H., Helder, W., Seifert, R., and Christof, O.: Methane
distribution in European tidal estuaries, Biogeochemistry, 59, 95–119, https://doi.org/10.1023/A:1015515130419, 2002.

Minkkinen, K. and Laine, J.: Vegetation heterogeneity and ditches create spatial variability in methane fluxes from peatlands
drained for forestry, Plant Soil, 285, 289–304, https://doi.org/10.1007/s11104-006-9016-4, 2006.

Minkkinen, K., Laine, J., Nykänen, H., and Martikainen, P. J.: Importance of drainage ditches in emissions of methane from
mires drained for forestry, Can. J. For. Res., 27, 949–952, https://doi.org/10.1139/x97-016, 1997.

Morozumi, T., Shingubara, R., Murase, J., Nagai, S., Kobayashi, H., Takano, S., Tei, S., Fan, R., Maximov, T. C., and
Sugimoto, A.: Usability of water surface reflectance for the determination of riverine dissolved methane during extreme
flooding in northeastern Siberia, Polar Sci., 21, 186–194, https://doi.org/10.1016/j.polar.2019.01.005, 2019.

Morrice, J. A., Dahm, C. N., Valett, H. M., Unnikrishna, P. V., and Campana, M. E.: Terminal electron accepting processes in
the alluvial sediments of a headwater stream, J. N. Amer. Benthol. Soc., 19, 593–608, https://doi.org/10.2307/1468119, 2000.

Mosher, J., Fortner, A., Phillips, J., Bevelhimer, M., Stewart, A., and Troia, M.: Spatial and temporal correlates of greenhouse
gas diffusion from a hydropower reservoir in the southern United States, Water, 7, 5910–5927,
https://doi.org/10.3390/w7115910, 2015.

Mulholland, P. J., Helton, A. M., Poole, G. C., Hall, R. O., Hamilton, S. K., Peterson, B. J., Tank, J. L., Ashkenas, L. R.,
Cooper, L. W., Dahm, C. N., Dodds, W. K., Findlay, S. E. G., Gregory, S. V., Grimm, N. B., Johnson, S. L., McDowell, W.
H., Meyer, J. L., Valett, H. M., Webster, J. R., Arango, C. P., Beaulieu, J. J., Bernot, M. J., Burgin, A. J., Crenshaw, C. L.,
Johnson, L. T., Niederlehner, B. R., O'Brien, J. M., Potter, J. D., Sheibley, R. W., Sobota, D. J., and Thomas, S. M.: Stream
denitrification across biomes and its response to anthropogenic nitrate loading, Nature, 452, 202–205,
https://doi.org/10.1038/nature06686, 2008.



Murphy, J. F., Arnold, A., Duerdoth, C. P., Hawczak, A., Pacioglu, O., Pretty, J. L., and Jones, J. I.: Temporal variation in temperature and light availability in the Hampshire Avon, United Kingdom [Macronutrient Cycling], NERC Environmental Information Data Centre, https://doi.org/10.5285/9B6A6233-85AD-44F4-BA83-4905B8C48713, 2017.

Natchimuthu, S., Wallin, M. B., Klemedtsson, L., and Bastviken, D.: Spatio-temporal patterns of stream methane and carbon dioxide emissions in a hemiboreal catchment in Southwest Sweden, Sci. Rep., 7, 39729, https://doi.org/10.1038/srep39729, 2017.

National Ecological Observatory Network (NEON): Chemical properties of surface water (DP1.20093.001): RELEASE-2021 (RELEASE-2021), https://doi.org/10.48443/05K7-E011, 2021a.

National Ecological Observatory Network (NEON): Discharge field collection (DP1.20048.001): RELEASE-2021 (RELEASE-2021), https://doi.org/10.48443/8TD9-4Z94, 2021b.

Neu, V., Neill, C., and Krusche, A. V.: Gaseous and fluvial carbon export from an Amazon forest watershed, Biogeochemistry, 105, 133–147, https://doi.org/10.1007/s10533-011-9581-3, 2011.

Okuku, E. O., Bouillon, S., Tole, M., and Borges, A. V.: Diffusive emissions of methane and nitrous oxide from a cascade of tropical hydropower reservoirs in Kenya, Lakes & Reserv., 24, 127–135, https://doi.org/10.1111/lre.12264, 2019.

Olid, C., Rodellas, V., Rocher-Ros, G., Garcia-Orellana, J., Diego-Feliu, M., Alorda-Kleinglass, A., Bastviken, D., and Karlsson, J.: Groundwater discharge as a driver of methane emissions from Arctic lakes, Nat. Commun., 13, 3667, https://doi.org/10.1038/s41467-022-31219-1, 2022.

Osudar, R., Liebner, S., Alawi, M., Yang, S., Bussmann, I., and Wagner, D.: Methane turnover and methanotrophic communities in arctic aquatic ecosystems of the Lena Delta, Northeast Siberia, FEMS Microbiol. Ecol., 92, fiw116, https://doi.org/10.1093/femsec/fiw116, 2016.

Oviedo-Vargas, D., Genereux, D. P., Dierick, D., and Oberbauer, S. F.: The effect of regional groundwater on carbon dioxide and methane emissions from a lowland rainforest stream in Costa Rica, J. Geophys. Res. Biogeosci., 120, 2579–2595, https://doi.org/10.1002/2015JG003009, 2015.

Pain, A., Martin, J., Martin, E., and Rahman, S.: Hydrogeochemistry of Greenlandic proglacial and nonglacial streams, 2017-2018, Arctic Data Center, https://doi.org/10.18739/A2PC2T94T, 2019.

Peacock, M.: Forest ditch GHG data, figshare, https://doi.org/10.6084/m9.figshare.15152253.V3, 2021a.





Peacock, M.: Peacock et al, 2021, GCB data figshare, https://doi.org/10.6084/m9.figshare.14784852.V2, 2021b.

Peacock, M., Ridley, L. M., Evans, C. D., and Gauci, V.: Management effects on greenhouse gas dynamics in fen ditches, Sci. Total Environ., 578, 601–612, https://doi.org/10.1016/j.scitotenv.2016.11.005, 2017.

Peacock, M., Gauci, V., Baird, A. J., Burden, A., Chapman, P. J., Cumming, A., Evans, J. G., Grayson, R. P., Holden, J., Kaduk, J., Morrison, R., Page, S., Pan, G., Ridley, L. M., Williamson, J., Worrall, F., and Evans, C. D.: The full carbon balance
of a rewetted cropland fen and a conservation-managed fen, Agric. Ecosys. Environ., 269, 1–12, https://doi.org/10.1016/j.agee.2018.09.020, 2019.

Peacock, M., Audet, J., Bastviken, D., Cook, S., Evans, C. D., Grinham, A., Holgerson, M. A., Högbom, L., Pickard, A. E., Zieliński, P., and Futter, M. N.: Small artificial waterbodies are widespread and persistent emitters of methane and carbon dioxide, Global Change Biol., 27, 5109–5123, https://doi.org/10.1111/gcb.15762, 2021a.

Peacock, M., Granath, G., Wallin, M. B., Högbom, L., and Futter, M. N.: Significant emissions from forest drainage ditches— An unaccounted term in anthropogenic greenhouse gas inventories?, J. Geophys. Res. Biogeosci., 126, e2021JG006478, https://doi.org/10.1029/2021JG006478, 2021b.

Pickard, A. E., Skiba, U. M., Carvalho, L., Heal, K. V., Rees, R. M., and Harley, J. F.: River Tay, Scotland, water chemistry and greenhouse gas measurements 2009-2010, NERC Environmental Information Data Centre,
https://doi.org/10.5285/a61da7da-b7ef-40b7-a324-c3711ef81207, 2019.

Pickard, A., Dinsmore, K. J., Billett, M. F., and Branagan, M.: Aquatic carbon and greenhouse gas concentrations in headwater streams draining from natural, drained and restored peatland catchments in the Flow Country, Scotland, September 2008-August 2010, NERC Environmental Information Data Centre, https://doi.org/10.5285/7525088d-e504-456a-bc55-e48d8ca85303, 2021.

Pickard, A. E., Branagan, M., Billett, M. F., Andersen, R., and Dinsmore, K. J.: Effects of peatland management on aquatic carbon concentrations and fluxes, Biogeosciences, 19, 1321–1334, https://doi.org/10.5194/bg-19-1321-2022, 2022.

Pulliam, W. M.: Carbon dioxide and methane exports from a southeastern floodplain swamp, Ecol. Monogr., 63, 29–53, https://doi.org/10.2307/2937122, 1993.

Qin, X., Li, Y., Wan, Y., Fan, M., Liao, Y., Li, Y., Wang, B., and Gao, Q.: Diffusive flux of $CH_4$ and $N_2O$ from agricultural
river networks: Regression tree and importance analysis, Sci. Total Environ., 717, 137244, https://doi.org/10.1016/j.scitotenv.2020.137244, 2020a.





Qin, X., Li, Y., Wan, Y., Fan, M., Liao, Y., Li, Y., Wang, B., Gao, Q., Wu, H., and Chen, X.: Multiple stable isotopic signatures corroborate the predominance of acetoclastic methanogenesis during $CH_4$ formation in agricultural river networks, Agric. Ecosys. Environ., 296, 106930, https://doi.org/10.1016/j.agee.2020.106930, 2020b.

Qin, Y., Wang, Z., Li, Z., and Yang, B.: $CO_2$ and $CH_4$ flux across water-air interface and environmental factors in Pengxi River of the Three Gorges Reservoir, J. Earth Environ., 10, 177–189, doi (China): 10.7515./JEE182071, 2019a.

Qin, Y., Wang, Z., Li, Z., and Yang, B.: $CO_2$ and $CH_4$ partial press and flux across water-air interface in the downstream of the Jinsha River, China, Appl. Ecol. Env. Res., 17, https://doi.org/10.15666/aeer/1703_58235839, 2019b.

Qu, B., Aho, K. S., Li, C., Kang, S., Sillanpää, M., Yan, F., and Raymond, P. A.: Greenhouse gases emissions in rivers of the Tibetan Plateau, Sci. Rep., 7, 16573, https://doi.org/10.1038/s41598-017-16552-6, 2017.

Rajkumar, N. A., Barnes, J., Ramesh, R., Purvaja, R., and Upstill-Goddard, R. C.: Methane and nitrous oxide fluxes in the polluted Adyar River and estuary, SE India, Mar. Pollut. Bull., 56, 2043–2051, https://doi.org/10.1016/j.marpolbul.2008.08.005, 2008.

Ran, L., Shi, H., and Yang, X.: Magnitude and drivers of $CO_2$ and $CH_4$ emissions from an arid/semiarid river catchment on the Chinese Loess Plateau, J. Hydrol., 598, 126260, https://doi.org/10.1016/j.jhydrol.2021.126260, 2021.

Reeburgh, W. S., King, J. Y., Regli, S. K., Kling, G. W., Auerbach, N. A., and Walker, D. A.: A $CH_4$ emission estimate for the Kuparuk River basin, Alaska, J. Geophys. Res., 103, 29005–29013, https://doi.org/10.1029/98JD00993, 1998.

Richey, J. E., Devol, A. H., Wofsy, S. C., Victoria, R., and Riberio, M. N. G.: Biogenic gases and the oxidation and reduction of carbon in Amazon River and floodplain waters: Amazon dissolved gases, Limnol. Oceanogr., 33, 551–561, https://doi.org/10.4319/lo.1988.33.4.0551, 1988.

Robison, A. and Wolheim, W.: PIE LTER dissolved methane and water temperature from four headwater streams in Massachusetts and New Hampshire., Environmental Data Initiative, http://dx.doi.org/10.6073/pasta/789147c37923aeecc924ab33b35595eb, 2021a.

Robison, A. and Wollheim, W.: PIE LTER time series of methane, $CO_2$ and $N_2O$ ebullition measurements at four headwater streams in Massachusetts and New Hampshire., Environmental Data Initiative, https://doi.org/10.6073/pasta/9b7fb5afa7b55f3c198f37ade701a542, 2021b.





Robison, A. L., Wollheim, W. M., Turek, B., Bova, C., Snay, C., and Varner, R. K.: Spatial and temporal heterogeneity of methane ebullition in lowland headwater streams and the impact on sampling design, Limnol. Oceanogr., 66, 4063–4076, https://doi.org/10.1002/lno.11943, 2021.

Roulet, N. T. and Moore, T. R.: The effect of forestry drainage practices on the emission of methane from northern peatlands, Can. J. For. Res., 25, 491–499, https://doi.org/10.1139/x95-055, 1995.

Rovelli, L., Olde, L. A., Heppell, C. M., Binley, A., Yvon-Durocher, G., Glud, R. N., and Trimmer, M.: High-resolution time series of day and night outgassing rates of carbon dioxide and methane for six tributaries of Hampshire River Avon (UK) collected with an automated floating chamber in late spring 2015., figshare, https://doi.org/10.6084/m9.figshare.16545954.v1,
2021a.

Rovelli, L., Olde, L. A., Heppell, C. M., Binley, A., Yvon-Durocher, G., Glud, R. N., and Trimmer, M.: Summary data of chamber-based oxygen and methane consumption and production in the streambed and outgassing of carbon dioxide and methane to the atmosphere collected seasonally for six tributaries of Hampshire River Avon (UK) during 2013-2014., figshare, https://doi.org/10.6084/m9.figshare.16545846.v1, 2021b.

Rovelli, L., Olde, L. A., Heppell, C. M., Binley, A., Yvon-Durocher, G., Glud, R. N., and Trimmer, M.: Contrasting biophysical controls on carbon dioxide and methane outgassing from streams, J. Geophys. Res. Biogeosci., e2021JG006328, 127, https://doi.org/10.1029/2021JG006328, 2022.

Rulík, M., Čáp, L., and Hlaváčová, E.: Methane in the hyporheic zone of a small lowland stream (Sitka, Czech Republic), Limnologica, 30, 359–366, https://doi.org/10.1016/S0075-9511(00)80029-8, 2000.

Sanders, I. A., Heppell, C. M., Cotton, J. A., Wharton, G., Hildrew, A. G., Flowers, E. J., and Trimmer, M.: Emission of methane from chalk streams has potential implications for agricultural practices, Freshwater Biol., 52, 1176–1186, https://doi.org/10.1111/j.1365-2427.2007.01745.x, 2007.

Sansone, F. J., Holmes, M. E., and Popp, B. N.: Methane stable isotopic ratios and concentrations as indicators of methane dynamics in estuaries, Global Biogeochem. Cycles, 13, 463–474, https://doi.org/10.1029/1999gb900012, 1999.

Sawakuchi, H. O., Bastviken, D., Sawakuchi, A. O., Krusche, A. V., Ballester, M. V. R., and Richey, J. E.: Methane emissions from Amazonian rivers and their contribution to the global methane budget, Global Change Biol., 20, 2829–2840, https://doi.org/10.1111/gcb.12646, 2014.



Sawakuchi, H. O., Bastviken, D., Enrich-Prast, A., Ward, N. D., Camargo, P. B., and Richey, J. E.: Low diffusive methane emissions from the main channel of a large Amazonian run-of-the-river reservoir attributed to high methane oxidation, Front. Environ. Sci., 9, 655455, https://doi.org/10.3389/fenvs.2021.655455, 2021.

Schade, J. D., Bailio, J., and McDowell, W. H.: Greenhouse gas flux from headwater streams in New Hampshire, USA: Patterns and drivers: Greenhouse gas flux from headwater streams, Limnol. Oceanogr., 61, S165–S174, https://doi.org/10.1002/lno.10337, 2016.

Schrier-Uijl, A. P., Kroon, P. S., Leffelaar, P. A., van Huissteden, J. C., Berendse, F., and Veenendaal, E. M.: Methane emissions in two drained peat agro-ecosystems with high and low agricultural intensity, Plant Soil, 329, 509–520, https://doi.org/10.1007/s11104-009-0180-1, 2010.

Schrier-Uijl, A. P., Veraart, A. J., Leffelaar, P. A., Berendse, F., and Veenendaal, E. M.: Release of $CO_2$ and $CH_4$ from lakes and drainage ditches in temperate wetlands, Biogeochemistry, 102, 265–279, https://doi.org/10.1007/s10533-010-9440-7, 2011.

Schuster, P. F. (Ed.): Water and sediment quality in the Yukon River Basin, Alaska, during Water Year 2001, U.S. Geological Survey, Denver, CO, 120 pp., https://doi.org/10.3133/ofr03427, 2003.

Schuster, P. F. (Ed.): Water and sediment quality in the Yukon River Basin, Alaska, during Water Year 2002, U.S. Geological Survey, Denver, CO, 82 pp., https://doi.org/10.3133/ofr20051199, 2006a.

Schuster, P. F. (Ed.): Water and sediment quality in the Yukon River Basin, Alaska, during Water Year 2003, U.S. Geological Survey, Boulder, CO, 74 pp., https://doi.org/10.3133/ofr20051397, 2006b.

Schuster, P. F. (Ed.): Water and sediment quality in the Yukon River Basin, Alaska, during Water Year 2004, U.S. Geological Survey, Boulder, CO, 67 pp., https://doi.org/10.3133/ofr20061258, 2006c.

Schuster, P. F. (Ed.): Water and sediment quality in the Yukon River Basin, Alaska, during Water Year 2005, U.S. Geological Survey, Boulder, CO, 65 pp., https://doi.org/10.3133/ofr20071037, 2007.

Schuster, P. F., Maracle, K. B., and Herman-Mercer, N. (Eds.): Water quality in the Yukon River Basin, Alaska, during Water Years 2006 - 2008, U.S. Geological Survey, Reston, VA, 220 pp., https://doi.org/10.3133/ofr20101241, 2010.

Selvam, B. P., Natchimuthu, S., Arunachalam, L., and Bastviken, D.: Methane and carbon dioxide emissions from inland waters in India - implications for large scale greenhouse gas balances, Global Change Biol., 20, 3397–3407, https://doi.org/10.1111/gcb.12575, 2014.





Shelley, F., Grey, J., and Trimmer, M.: Widespread methanotrophic primary production in lowland chalk rivers, Proc. R. Soc. B, 281, 20132854, https://doi.org/10.1098/rspb.2013.2854, 2014.

Shi, W., Chen, Q., Yi, Q., Yu, J., Ji, Y., Hu, L., and Chen, Y.: Carbon Emission from Cascade Reservoirs: Spatial heterogeneity and mechanisms, Environ. Sci. Technol., 51, 12175–12181, https://doi.org/10.1021/acs.est.7b03590, 2017.

Sieczko, A. K., Demeter, K., Singer, G. A., Tritthart, M., Preiner, S., Mayr, M., Meisterl, K., and Peduzzi, P.: Aquatic methane
dynamics in a human-impacted river-floodplain of the Danube: Floodplain methane emission, Limnol. Oceanogr., 61, S175–S187, https://doi.org/10.1002/lno.10346, 2016.

Silvennoinen, H., Liikanen, A., Rintala, J., and Martikainen, P. J.: Greenhouse gas fluxes from the eutrophic Temmesjoki River and its Estuary in the Liminganlahti Bay (the Baltic Sea), Biogeochemistry, 90, 193–208, https://doi.org/10.1007/s10533-008-9244-1, 2008.

Smith, L. K., Lewis, Jr., W. M., Chanton, J. P., Cronin, G., and Hamilton, S. K.: Methane emissions from the Orinoco River floodplain, Venezuela, Biogeochemistry, 51, 113–140, https://doi.org/10.1023/A:1006443429909, 2000.

Smith, R. L. and Bohlke, J. K.: Methane and nitrous oxide temporal and spatial concentrations in the Iroquois River and Sugar Creek in Northwestern Indiana and Northeastern Illinois, 1999-2003 (ver. 2.0, November 2020), U.S. Geological Survey data release, https://doi.org/10.5066/f7th8kwz, 2018.

Smith, R. L. and Böhlke, J. K.: Methane and nitrous oxide temporal and spatial variability in two midwestern USA streams containing high nitrate concentrations, Sci. Total Environ., 685, 574–588, https://doi.org/10.1016/j.scitotenv.2019.05.374, 2019.

Smith, R. M., Kaushal, S. S., Beaulieu, J. J., Pennino, M. J., and Welty, C.: Influence of infrastructure on water quality and greenhouse gas dynamics in urban streams, Biogeosciences, 14, 2831–2849, https://doi.org/10.5194/bg-14-2831-2017, 2017.

Soja, G., Kitzler, B., and Soja, A.-M.: Emissions of greenhouse gases from Lake Neusiedl, a shallow steppe lake in Eastern Austria, Hydrobiologia, 731, 125–138, https://doi.org/10.1007/s10750-013-1681-8, 2014.

Soued, C. and Prairie, Y. T.: The carbon footprint of a Malaysian tropical reservoir: measured versus modelled estimates highlight the underestimated key role of downstream processes, Biogeosciences, 17, 515–527, https://doi.org/10.5194/bg-17-515-2020, 2020.

Spawn, S., Dunn, S., Fiske, G., Natali, S., Schade, J., and Zimov, N.: Summer methane ebullition from a headwater catchment in Northeastern Siberia, Inland Waters, 5, 224–230, https://doi.org/10.5268/IW-5.3.845, 2015.





Stets, E. G.: Synoptic carbon gas fluxes from streams, rivers, and lakes in the conterminous US, USGS Landcarbon, 2012 to 2014, U.S. Geological Survey data release, https://doi.org/10.5066/F72B8W2C, 2016.

Striegl, R. G., Dornblaser, M. M., McDonald, C. P., Rover, J. R., and Stets, E. G.: Carbon dioxide and methane emissions from the Yukon River system, Global Biogeochem. Cycles, 26, GB0E05, https://doi.org/10.1029/2012gb004306, 2012.

Sturm, K., Grinham, A., Werner, U., and Yuan, Z.: Sources and sinks of methane and nitrous oxide in the subtropical Brisbane River estuary, South East Queensland, Australia, Estuarine Coastal Shelf Sci., 168, 10–21, https://doi.org/10.1016/j.ecss.2015.11.002, 2016.

Taillardat, P., Bodmer, P., Deblois, C. P., Ponçot, A., Prijac, A., Riahi, K., Gandois, L., del Giorgio, P. A., Bourgault, M. A., Tremblay, A., and Garneau, M.: Carbon dioxide and methane dynamics in a peatland headwater stream: origin, processes and implications, Zenodo, https://zenodo.org/record/6073957, 2021.

Taillardat, P., Bodmer, P., Deblois, C. P., Ponçot, A., Prijac, A., Riahi, K., Gandois, L., del Giorgio, P. A., Bourgault, M. A., Tremblay, A., and Garneau, M.: Carbon dioxide and methane dynamics in a peatland headwater stream: Origins, processes and implications, J. Geophys. Res. Biogeosci., e2022JG006855, https://doi.org/10.1029/2022JG006855, 2022.

Taniwaki, R. H.: Methane concentrations and fluxes in agricultural and preserved tropical headwater streams, Environmental Data Initiative, https://doi.org/10.6073/pasta/82b1aa19ad5b4ef88939303f79c1e74c, 2022.

Taniwaki, R. H., Cunha, D. G. F., Bento, C. B., Martinelli, L. A., Stanley, E. H., Filoso, S., Ferreira, M. de S., França, M. V., Ribeiro Júnior, J. W., Schiesari, L. C., and do Carmo, J. B.: Methane concentrations and fluxes in agricultural and preserved tropical headwater streams, Sci. Total Environ., 844, 157238, https://doi.org/10.1016/j.scitotenv.2022.157238, 2022.

Teodoru, C. R., Nyoni, F. C., Borges, A. V., Darchambeau, F., Nyambe, I., and Bouillon, S.: Dynamics of greenhouse gases ($CO_2$, $CH_4$, $N_2O$) along the Zambezi River and major tributaries, and their importance in the riverine carbon budget, Biogeosciences, 12, 2431–2453, https://doi.org/10.5194/bg-12-2431-2015, 2015.

Trimmer, M., Hildrew, A. G., Jackson, M. C., Pretty, J. L., and Grey, J.: Evidence for the role of methane-derived carbon in a free-flowing, lowland river food web, Limnol. Oceanogr., 54, 1541–1547, https://doi.org/10.4319/lo.2009.54.5.1541, 2009.

Upstill-Goddard, R. C., Barnes, J., Frost, T., Punshon, S., and Owens, N. J. P.: Methane in the southern North Sea: Low-salinity inputs, estuarine removal, and atmospheric flux, Global Biogeochem. Cycles, 14, 1205–1217, https://doi.org/10.1029/1999GB001236, 2000.



Upstill-Goddard, R. C., Salter, M. E., Mann, P. J., Barnes, J., Poulsen, J., Dinga, B., Fiske, G. J., and Holmes, R. M.: The riverine source of $CH_4$ and $N_2O$ from the Republic of Congo, western Congo Basin, Biogeosciences, 14, 2267–2281, https://doi.org/10.5194/bg-14-2267-2017, 2017.

Vermaat, J. E., Hellmann, F., Dias, A. T. C., Hoorens, B., van Logtestijn, R. S. P., and Aerts, R.: Greenhouse gas fluxes from Dutch peatland water bodies: Importance of the surrounding landscape, Wetlands, 31, 493–498, https://doi.org/10.1007/s13157-011-0170-y, 2011.

Vidon, P. and Serchan, S.: Impact of stream geomorphology on greenhouse gas concentration in a New York Mountain stream, Water Air Soil Pollut., 227, 428, https://doi.org/10.1007/s11270-016-3131-5, 2016.

Villa, J. A., Ju, Y., Smith, G. J., Angle, J. C., Renteria, L., Arntzen, E., Harding, S. F., Stegen, J. C., Wrighton, K. C., and Bohrer, G.: Chamber flux and porewater concentration of $CH_4$, $CO_2$ and $N_2O$, 2018, Columbia River bank at the Hanford site, WA, USA, ESS-DIVE Data Archive, https://doi.org/10.15485/1595105, 2020a.

Villa, J. A., Smith, G. J., Ju, Y., Renteria, L., Angle, J. C., Arntzen, E., Harding, S. F., Ren, H., Chen, X., Sawyer, A. H., Graham, E. B., Stegen, J. C., Wrighton, K. C., and Bohrer, G.: Methane and nitrous oxide porewater concentrations and surface fluxes of a regulated river, Sci. Total Environ., 715, 136920, https://doi.org/10.1016/j.scitotenv.2020.136920, 2020b.

Vorobyev, S. N., Karlsson, J., Kolesnichenko, Y. Y., Korets, M. A., and Pokrovsky, O. S.: Fluvial carbon dioxide emission from the Lena River basin during the spring flood, Biogeosciences, 18, 4919–4936, https://doi.org/10.5194/bg-18-4919-2021, 2021.

Wallin, M. B., Löfgren, S., Erlandsson, M., and Bishop, K.: Representative regional sampling of carbon dioxide and methane concentrations in hemiboreal headwater streams reveal underestimates in less systematic approaches: $CO_2$ and $CH_4$ in headwater streams, Global Biogeochem. Cycles, 28, 465–479, https://doi.org/10.1002/2013GB004715, 2014.

Wallin, M. B., Campeau, A., Audet, J., Bastviken, D., Bishop, K., Kokic, J., Laudon, H., Lundin, E., Löfgren, S., Natchimuthu, S., Sobek, S., Teutschbein, C., Weyhenmeyer, G. A., and Grabs, T.: Carbon dioxide and methane emissions of Swedish low-order streams-a national estimate and lessons learnt from more than a decade of observations: Carbon dioxide and methane emissions, Limnol. Oceanogr. Lett., 3, 156–167, https://doi.org/10.1002/lol2.10061, 2018.

Wang, D., Chen, Z., Sun, W., Hu, B., and Xu, S.: Methane and nitrous oxide concentration and emission flux of Yangtze Delta plain river net, Sci. China Ser. B-Chem., 52, 652–661, https://doi.org/10.1007/s11426-009-0024-0, 2009.



Wang, G., Xia, X., Liu, S., Zhang, L., Zhang, S., Wang, J., Xi, N., and Zhang, Q.: Intense methane ebullition from urban inland waters and its significant contribution to greenhouse gas emissions, Water Research, 189, 116654, https://doi.org/10.1016/j.watres.2020.116654, 2021.

Wang, R., Zhang, H., Zhang, W., Zheng, X., Butterbach-Bahl, K., Li, S., and Han, S.: An urban polluted river as a significant hotspot for water–atmosphere exchange of $CH_4$ and $N_2O$, Environ. Pollut., 264, 114770, https://doi.org/10.1016/j.envpol.2020.114770, 2020.

Wang, X., He, Y., Chen, H., Yuan, X., Peng, C., Yue, J., Zhang, Q., and Zhou, L.: $CH_4$ concentrations and fluxes in a subtropical metropolitan river network: Watershed urbanization impacts and environmental controls, Sci. Total Environ., 622–623, 1079–1089, https://doi.org/10.1016/j.scitotenv.2017.12.054, 2018.

Webb, J. R., Santos, I. R., Maher, D. T., Macdonald, B., Robson, B., Isaac, P., and McHugh, I.: Terrestrial versus aquatic carbon fluxes in a subtropical agricultural floodplain over an annual cycle, Agric. For. Meteorol., 260–261, 262–272, https://doi.org/10.1016/j.agrformet.2018.06.015, 2018.

Wendt, A. K., Sowers, T., Hynek, S., Lemon, J., Beddings, E., Zheng, G., Li, Z., Williams, J. Z., and Brantley, S. L.: Scientist-nonscientist teams explore methane sources in streams near oil/gas development, J. Contemp. Water Res. Educ., 164, 80–111, https://doi.org/10.1111/j.1936-704X.2018.03286.x, 2018.

Wilcock, R. J. and Sorrell, B. K.: Emissions of greenhouse gases CH4 and N2O from low-gradient streams in agriculturally developed catchments, Water Air Soil Pollut, 188, 155–170, https://doi.org/10.1007/s11270-007-9532-8, 2008.

Wilkniss, P. E., Lamontagne, R. A., Larson, R. E., and Swinnerton, J. W.: Atmospheric trace gases and land and sea breezes at the Sepik River coast of Papua New Guinea, J. Geophys. Res., 83, 3672-3674, https://doi.org/10.1029/jc083ic07p03672, 1978.

Woda, J., Wen, T., Lemon, J., Marcon, V., Keeports, C. M., Zelt, F., Steffy, L. Y., and Brantley, S. L.: Methane concentrations in streams reveal gas leak discharges in regions of oil, gas, and coal development, Sci. Total Environ., 737, 140105, https://doi.org/10.1016/j.scitotenv.2020.140105, 2020.

Wu, H., Zhao, Q., Gao, Q., Li, Y., Wan, Y., Li, Y., Tian, D., Liao, Y., Fan, M., Ganjurjav, H. Hu, G., Wang, B., Chen, X., and Qin, X.: Human activities inducing high $CH_4$ diffusive fluxes in an agricultural river catchment in subtropical China, Sustainability, 12, 2114, https://doi.org/10.3390/su12052114, 2020.



Wu, S., Li, S., Zou, Z., Hu, T., Hu, Z., Liu, S., and Zou, J.: High methane emissions largely attributed to ebullitive fluxes from a subtropical river draining a rice paddy watershed in China, Environ. Sci. Technol., 53, 3499–3507, https://doi.org/10.1021/acs.est.8b05286, 2019.

Wu, W.: Characteristics and influencing factors of greenhouse gas emissions from water bodies of Zhongtianshe River Basin
in Tianmu Lake area, M.S. thesis, Nanjing Normal University, Nanjing, China, doi (China): 10.27245/d.cnki.gnjsu.2020.001588, https://kns.cnki.net/KCMS/detail/detail.aspx?dbname=CMFD202101&filename=1021516788.nh, 2020.

Wu, Y., Li, H., Chen, W.: Effects and emission characteristics of greenhouse gases from Wenyu River in summer, Environ. Sci. Technol. (China), doi (China): 10.3969/j.issn.1003-6504.2016.05.002, 39, 8-16, 2016.

Wynn, P.: Direct isotopic evidence of biogenic methane production and efflux from beneath a temperate glacier, Lancaster University, https://doi.org/10.17635/lancaster/researchdata/246, 2018b.

Xiao, Q., Hu, Z., Hu, C., Islam, A. R. M. T., Bian, H., Chen, S., Liu, C., and Lee, X.: A highly agricultural river network in Jurong Reservoir watershed as significant $CO_2$ and $CH_4$ sources, Sci. Total Environ., 769, 144558,
https://doi.org/10.1016/j.scitotenv.2020.144558, 2021.

Xu, H., Chen, M., Xiao, S., Yu, Z., and Zheng, X.: Temporal and spatial pattern of dissolved methane concentration in the river of a small karst watershed in Western Hubei, China Environ. Sci., https://kns.cnki.net/kcms/detail/11.2201.x.20210408.0910.002.html, 2020.

Yang, L.: Contrasting methane emissions from upstream and downstream rivers and their associated subtropical reservoir in
eastern China, Sci. Rep., 9, 8072, https://doi.org/10.1038/s41598-019-44470-2, 2019.

Yang, L., Li, X., Yan, W., Ma, P., and Wang, J.: $CH_4$ concentrations and emissions from three rivers in the Chaohu Lake watershed in southeast China, J. Integr. Agric., 11, 665–673, https://doi.org/10.1016/S2095-3119(12)60054-9, 2012.

Yang, S.-S., Chen, I.-C., Ching-Pao, L., Liu, L.-Y., and Chang, C.-H.: Carbon dioxide and methane emissions from Tanswei River in Northern Taiwan, Atmos. Pollut, Res., 6, 52–61, https://doi.org/10.5094/APR.2015.007, 2015.

Yavitt, J. B., Lang, G. E., and Sexstone, A. J.: Methane fluxes in wetland and forest soils, beaver ponds, and low-order streams of a temperate forest ecosystem, J. Geophys. Res., 95, 22463, https://doi.org/10.1029/JD095iD13p22463, 1990.





Ye, R., Wu, Q., Zhao, Z., Hu, J., Cui, L., and Ding, H.: Concentrations and emissions of dissolved $CH_4$ and $N_2O$ in the Yarlung Tsangpo River, Chinese J. Ecol., 38, 791–798, doi (China): 10.13292/j.1000-4890.201903.001, 2019.

Yu, Z., Wang, D., Li, Y., Deng, H., Hu, B., Ye, M., Zhou, X., Da, L., Chen, Z., and Xu, S.: Carbon dioxide and methane dynamics in a human-dominated lowland coastal river network (Shanghai, China): $CO_2$ and $CH_4$ in Shanghai River Network, J. Geophys. Res. Biogeosci., 122, 1738–1758, https://doi.org/10.1002/2017JG003798, 2017.

Zhang, F.: Spatio-temporal dynamics of dissolved concentrations of $N_2O$ and $CH_4$ and their emission flux in Weihe river of Xinxiang, M.S. thesis, Henan Normal University, 81 pp., Xinxiang, China, https://kns.cnki.net/KCMS/detail/detail.aspx?dbname=CMFD201701&filename=1016231948.nh, 2016.

Zhang, L., Xia, X., Liu, S., Zhang, S., Li, S., Wang, J., Wang, G., Gao, H., Zhang, Z., Wang, Q., Wen, W., Liu, R., Yang, Z., Stanley, E. H., and Raymond, P. A.: Significant methane ebullition from alpine permafrost rivers on the East Qinghai–Tibet Plateau, Nat. Geosci., 13, 349–354, https://doi.org/10.1038/s41561-020-0571-8, 2020.

Zhang, L., Xia, X., Liu, S., Yang, Z., Stanley, E. H., and Raymond, P. A.: A dataset for methane concentrations and fluxes for alpine permafrost streams and rivers on the East Qinghai-Tibet Plateau, Environmental Data Initiative, https://doi.org/10.6073/pasta/3e9ed02d7d89a31eba4c36481255084c, 2021a.

Zhang, L., Xia, X., Battin, T. J., and Stanley, E. H.: Nitrous oxide dataset for East Qinghai-Tibet Plateau waterways, Environmental Data Initiative, https://doi.org/10.6073/pasta/ba9340800403c450e7d942d450237dc4, 2021b.

Zhang, W., Li, H., Pueppke, S. G., and Pang, J.: Restored riverine wetlands in a headwater stream can simultaneously behave as sinks of $N_2O$ and hotspots of $CH_4$ production, Environ. Pollut., 284, 117114, https://doi.org/10.1016/j.envpol.2021.117114, 2021a.

Zhang, W., Li, H., Xiao, Q., and Li, X.: Urban rivers are hotspots of riverine greenhouse gas ($N_2O$, $CH_4$, $CO_2$) emissions in the mixed-landscape chaohu lake basin, Water Res., 189, 116624, https://doi.org/10.1016/j.watres.2020.116624, 2021b.

Zhang, Y., Kang, S., Wei, D., Luo, X., Wang, Z., and Gao, T.: Sink or source? Methane and carbon dioxide emissions from cryoconite holes, subglacial sediments, and proglacial river runoff during intensive glacier melting on the Tibetan Plateau, Fundam. Res., 1, 232–239, https://doi.org/10.1016/j.fmre.2021.04.005, 2021.

Zhao, J.: Distribution and fluxes of methane and nitrous oxide in the Changjiang (Yangtze River) and its estuary, M.S. thesis, Ocean University of China, Qingdao, China, 81 pp., https://kns.cnki.net/KCMS/detail/detail.aspx?dbname=CMFD2012&filename=1011230510.nh, 2011.





Zhao, J., Zhang, G.-L., Wu, Y., and Yang, J.: Distribution and emission of methane from the Changjiang, Environ. Sci., 32,
18–25, doi (China): 10.13227/j.hjkx.2011.01.003, 2011.

Zhao, X.: Study on methane and nitrous oxide emissions from polluted water in Nanjing, M.S. thesis, Nanjing University of
Information Science and Technology, Nanjing, China, 55 pp., https://kns.cnki.net/KCMS/detail/detail.aspx
?dbname=CMFD2012&filename=1012369155.nh, 2012.

Zolkos, S., Tank, S. E., Striegl, R. G., and Kokelj, S. V.: Thermokarst effects on carbon dioxide and methane fluxes in streams
on the Peel Plateau (NWT, Canada), J. Geophys. Res. Biogeosci., 124, 1781–1798, https://doi.org/10.1029/2019JG005038,
2019.


**Supplement.**

The supplement related to this article is available at

**Author contributions.**

EHS conceived of the project idea, and led data entry, manuscript preparation, and data curation. LCL developed the code used for unit conversions, was responsible for data conversion and QA/QC, and contributed to data visualization, data analysis,

and code curation. GRR was responsible for spatial analyses, and contributed to data visualization, code curation, and manuscript preparation. The structure and composition of the manuscript were the result from collaborative discussions among EHS, GRR, LCL, NJC, SKO, and RAS. All authors contributed to data acquisition, data entry, data checking, and substantial manuscript revising and editing.

**Competing Interests.**

The authors declare that they have no conflict of interest.

**Acknowledgements.**

We are grateful to all the authors who generously shared data or made their data public; our science is better or this

transparency. We are also grateful to several of these investigators for patiently and thoughtfully responding to emails from E. Stanley. Corinna Gries provided advice and assistance with data publication. Stanley thanks K. Forest and K. McMahon for support during manuscript preparation

**Financial Support.**

Emily H. Stanley acknowledges support from the National Science Foundation #DEB-2025982, NTL LTER. Nora J. Casson was supported in part by the Canada Research Chair program. Gerard Rocher-Ros was supported by the Swedish Research Council (2021-06667) and the Department of Ecology and Environmental Science, Umeå University. Marcus B. Wallin was supported by the Swedish Research Council's VR (2021-04058) and Formas (2019-01105).






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
