# Peer review of "GRiMeDB: The global river database of methane concentrations and fluxes"

_Earth System Science Data, 2022_

## Referee Comment (RC2)

**"GRiMeDB: The global river database of methane concentrations and fluxes"** by Stanley et al.

GriMeDB is a valuable compilation of published and unpublished global data of methane ($CH_4$) concentrations and fluxes in flowing waters (rivers and streams), and the add on of values for $CO_2$ and $N_2O$. This valuable data set can support improving global and regional carbon and nitrogen budgets.

It is well known that rivers are generally emitters of relevant greenhouse gases to the atmosphere (i.e., $CO_2$, $CH_4$ and $N_2O$), and an updated global compilation in fluvial systems was very much needed, especially as the authors point out, with the increasing density of measurements over the last years in response to improved technologies. Such compilation was necessary in order to better quantify the role and contribution of fluvial systems to greenhouse gas emissions, as well as to better identify current geographic gaps.
The authors not only compiled these data sets into one single source, but also carefully curated the data in order to standardize the data base (e.g., use of same units in concentrations, daily aggregated values). Additionally, they provide interesting information as a result of statistical analysis showing trends in the data available regarding spatial coverage and temporal trends, over- and under-sampled areas and values with respect to channel types. These results certainly bring new opportunities and recommendations for future studies.
The GriMeDB data base includes detailed information on the methods used for quantification of the three methane emission pathways (diffusion, ebullition and plant-mediated transport), pointing out the large dominance of diffusive fluxes, followed by total fluxes (sum of diffusion and ebullition). Hence, they also provide information regarding gaps in current methodologies.
The data is easy to access and handle, organized in tables with clear description in headers for any user. The manuscript is well written and contains clear flow charts that explain to the reader the protocol followed for the preparation and curation of the data sets. The graphic work is also clear and contains summarized information of the global data base analysis. I have only few recommendations/comments and minor comments in the hope that these can be useful to the authors.
I can only thank the authors for the great effort put in this work. I am sure that the great effort made by the authors to compile this data set will be greatly appreciated by the community, and therefore I recommend the publication of this work in ESSD.

**General comments:**

-        The authors mentioned that the diffusive fluxes have large uncertainties due to the various methods used to calculate the gas transfer velocity *k* in the different studies. This is a well-known issue, and k is the most uncertain parameter, and what brings the largest uncertainty to any diffusive gas calculation. I wonder if the authors are able to provide in this context:
   1) An estimate of uncertainty in the flux calculation to estimate the error of the comparison between fluxes?

2) This might be out of the scope of this study, but having an overview of all the studies and summarizing the >25 different references for *k* model sources, can the authors provide an own view/recommendation of the "best model", or at least "most commonly used" model, for estimation of *k* in fluvial systems? This can benefit greatly to future studies to at least make use of a common model, allowing for a reduction in the uncertainty between studies for comparison purposes. This might vary in channel type though, but a distinction between e.g., hydraulic models, wind speed-based models, might hint at a best approach.

- Can the authors assess the uncertainty reduction that the data compiled in GriMeDB provides when compared to the estimate contribution of fluvial methane emissions or wetland streams (WS category) in the current global methane budget (Saunois et al., 2020)? In fact, the dominant bottom-up inventory of inland waters in the global methane budget is from wetlands, and including rivers and streams to these budgets might contribute to reduce the difference between top-down and bottom-up approaches.

A comparison to current budgets e.g., global methane budget should be assessed in order to compare the previous knowledge and added value that GriMeDB gives in reducing the uncertainty of carbon fluvial emissions.

- The authors do not mention in the manuscript if GriMeDB has the potential to be a living data base with a call to scientist to inform and/or deposit new (self-curated) data and to keep this data set growing.

The current GriMeDB contains I believe the majority of studies published with CH4 data in fluvial systems, but this list is not exhaustive, especially with missing recent published works (some in spotted under sampled regions), such as:

Canning et al., 2021, https://doi.org/10.5194/bg-18-3961-2021
Castro-Morales et al., 2022, https://doi.org/10.5194/bg-19-5059-2022
Patel et al., 2022, https://doi.org/10.1016/j.watres.2022.119380
Wesley et al., 2022 (still work on discussion, unpublished),
    https://doi.org/10.5194/egusphere-2022-549
Zhao et al., 2022, https://doi.org/10.1016/j.envpol.2021.118769
Zhu et al., 2022, https://doi.org/10.1038/s41467-022-31559-y

Maybe some of these studies, do not meet the requirements for GriMeDB, but it is just an example of current works that could be potentially added to the data base.

Additionally, and as part of my comment, can the authors provide some sort of protocol for sampling and curation of future or current available data set that are not yet part of GriMeDB and can help the authors to include and expand GriMeDB with community contributions. An example of such living data base is SOCAT (https://www.socat.info/), which provides clear protocols for scientists that wish to contribute with their data sets to this growing data base. This might not be on the scope of the authors at this stage, but given that they host the method it would be interesting if a call to the community can be done on the interest of keeping GriMeDB growing with a certain quality control.

Even if there is no current perspective by the authors to include more data sets to the current data base, an added short protocol as part of this manuscript would be extremely useful. Such protocol can include, e.g., parameters to measure and report (e.g., concentration in certain units), method used to determine $k$, measurement or information of other parameters (e.g., channel type following Strahler scale, channel slope), in a way that it can facilitate future additions of new data sets to GriMeDB.

Already the authors provide certain general recommendations (e.g., determine and report detection limits and include samples falling below these limits, include information on habitat conditions, studies expanding temporal dimensions are encouraged, routine $CH_4$ measurements with as part of water quality monitoring programs are encouraged), that can also be included and summarized more clearly in a protocol, with potential of including recommendations for e.g., $k$ parameterization to be used. If the authors cannot provide a protocol at this stage, I encourage them to add a section 4.3. in the discussion section with "Recommendations for future studies".

**Minor comments:**

P4,
L4 – How the authors assessed the quality control in unpublished data sets?
L84 – EU Zenodo is missing

P7,
L141 - it is better at this stage to mention that R package was used and not until section 2.5
L159 - It is missing the code NORM in Table 1, it is only mentioned in the caption of Fig. 14

P10,
L222 - it is necessary to add explicitly the units of concentrations and fluxes in each column, where it corresponds, in Tables A3 and A4, otherwise it is only possible to visualize the units by accessing the headers at the tables directly.

P21, L371 - there is a dot instead of a comma after the parenthesis

P22,
L396 – I believe the words "between $CH_4$ physical site attributes" need to be removed, so the sentence can only read "As with relationships between $CH_4$ concentration of flux and water chemistry parameters … "
L398 – Did the authors try to calculate correlations of Figs. 12 and 13 per latitude bands? They can be biased due to density of observations but at least some meaningful correlations might be seen between the selected parameters and methane.
L401 – refer here to Fig. 13a
L403 – refer here to Fig. 13b

P23, L408 – define here IMP as (impounded reaches), as all the other listed channel types were defined in this paragraph, the same for TH (thermogenic $CH_4$ inputs) in L415.

P25, Fig. 14 – These site-averaged concentrations need to be normalized to sample size so variations can be reduced due to the varying sizes and a better comparison between channel types can be done.

P29, L551 – "compared **TO** our previous efforts"

P30, L610 – additionally data assimilation models will strongly benefit from the GriMeDB database

P31, L628 – "… the expansion of GHG data **FOR** world streams and rivers …"

---

## Author Comment (AC1)

**Citation**: https://doi.org/10.5194/essd-2022-346-RC1

**RC1**: 'Comment on essd-2022-346', Yuanzhi Yao, 13 Nov 2022

The manuscript entitled " GRiMeDB: The global river database of methane concentrations and fluxes" proposes an a comprehensive database for riverine CH4 and the associate drivers. The proposed databased is based on the earlier work by Stanley et al. (2016). The authors present the flow chart for generating the database, and also the data analysis.

The topic of the manuscript is interesting and relevant to the earth system science community, as methane emission is a potent source of greenhouse gas. Overall, this is a well written manuscript without any apparent flaws. I can recommend the publication of this manuscript with minor revisions.

- We very much appreciate these positive remarks.

I also have a minor remark about the 'first database' stated in the abstract. I must confess, though, that I did not quite understand the difference between this database and the previous one (MethDB). I think MethDB is the first comprehensive database for river CH4. This work is an extension with significant efforts.

- We changed the wording in the abstract to simply say 'we present a comprehensive database…'

Figs. 5 and 6: Can you differentiate the sites for ebullitive and diffusive flux, respectively. It is very important for modelers.

- We created new versions of Figs. 5 and 6 for the different flux measurements and included it in the Supplement (Fig. S1 and S2).

The figures are very nice and useful. Not all of them need to be in color, though. I also struggled a little bit with the legends: I think Figures 11, 12 and 13 should have legends to show the meaning of the colors.

- For clarity, we used the orange-green color scheme throughout the manuscript (with the exception of Fig. 12 (which is now Fig. 13 in the revised ms) and Fig. 13b (now Fig. 14b) to denote concentration (orange) and flux (green) data. Similarly, brown is used for figures dealing with sites with both concentration and flux data. We would like to retain this color scheme for all plots for consistency. For Fig. 11 (now Fig. 12), because there is just one point type per plot and the y-axis and figure caption also define the contents of each plot, we have not added a legend to this figure. For Fig. 12 (now 13), we added legends, and for consistency, the flux plots were converted to a density plot format. For. Fig. 13b (now Fig. 14b), the colors in this figure have no specific significance- only that they are different- to represent different sites, as is explained in the figure caption. We believe that adding a legend would add confusion rather than clarity, so it was not added.

---

## Author Comment (AC2)

*Review of manuscript essd-2022-346*
**"GRiMeDB: The global river database of methane concentrations and fluxes"** by Stanley et al.

GriMeDB is a valuable compilation of published and unpublished global data of methane ($CH_4$) concentrations and fluxes in flowing waters (rivers and streams), and the add on of values for $CO_2$ and $N_2O$. This valuable data set can support improving global and regional carbon and nitrogen budgets.

It is well known that rivers are generally emitters of relevant greenhouse gases to the atmosphere (i.e., $CO_2$, $CH_4$ and $N_2O$), and an updated global compilation in fluvial systems was very much needed, especially as the authors point out, with the increasing density of measurements over the last years in response to improved technologies. Such compilation was necessary in order to better quantify the role and contribution of fluvial systems to greenhouse gas emissions, as well as to better identify current geographic gaps.

The authors not only compiled these data sets into one single source, but also carefully curated the data in order to standardize the data base (e.g., use of same units in concentrations, daily aggregated values). Additionally, they provide interesting information as a result of statistical analysis showing trends in the data available regarding spatial coverage and temporal trends, over- and under-sampled areas and values with respect to channel types. These results certainly bring new opportunities and recommendations for future studies.

The GriMeDB data base includes detailed information on the methods used for quantification of the three methane emission pathways (diffusion, ebullition and plant- mediated transport), pointing out the large dominance of diffusive fluxes, followed by total fluxes (sum of diffusion and ebullition). Hence, they also provide information regarding gaps in current methodologies.

The data is easy to access and handle, organized in tables with clear description in headers for any user. The manuscript is well written and contains clear flow charts that explain to the reader the protocol followed for the preparation and curation of the data sets. The graphic work is also clear and contains summarized information of the global data base analysis. I have only few recommendations/comments and minor comments in the hope that these can be useful to the authors.

I can only thank the authors for the great effort put in this work. I am sure that the great effort made by the authors to compile this data set will be greatly appreciated by the community, and therefore I recommend the publication of this work in ESSD.

Thank you for these kind comments and your positive endorsement of our efforts.

**General comments:**

-        The authors mentioned that the diffusive fluxes have large uncertainties due to the various methods used to calculate the gas transfer velocity *k* in the different studies. This is a well-known issue, and k is the most uncertain parameter, and what brings the largest uncertainty to any diffusive gas calculation. I wonder if the authors are able to provide in this context:

1) An estimate of uncertainty in the flux calculation to estimate the error of the comparison between fluxes?

2) This might be out of the scope of this study, but having an overview of all the studies and summarizing the >25 different references for *k* model sources, can the authors provide an own view/recommendation of the "best model", or at least "most commonly used" model, for estimation of *k* in fluvial systems? This can benefit greatly to future studies to at least make use of a common model, allowing for a reduction in the uncertainty between studies for comparison purposes. This might vary in channel type though, but a distinction between e.g., hydraulic models, wind speed-based models, might hint at a best approach.

We are hesitant to go down this path of recommending a "best model" for *k*, as this would take quite a bit of additional effort (that is, we feel it is beyond the scope of this already lengthy paper). Further, there are two excellent papers (Raymond et al. 2012; Hall & Ulseth 2020) that delve into this topic in substantial detail, analyze performance of different models, and address topics such as when/where wind speed models may become relevant in rivers. Nonetheless, we added text to the discussion noting the large and concerning number of approaches used to estimate k, and that considering the consequences of different choices may be facilitated by methodological information included in GRiMeDB.

-       Can the authors assess the uncertainty reduction that the data compiled in GriMeDB provides when compared to the estimate contribution of fluvial methane emissions or wetland streams (WS category) in the current global methane budget (Saunois et al., 2020)? In fact, the dominant bottom-up inventory of inland waters in the global methane budget is from wetlands, and including rivers and streams to these budgets might contribute to reduce the difference between top-down and bottom-up approaches.

A comparison to current budgets e.g., global methane budget should be assessed in order to compare the previous knowledge and added value that GriMeDB gives in reducing the uncertainty of carbon fluvial emissions.

We agree with this comment. Estimates of uncertainty and improved estimates of fluvial emissions have been made. However, because of the significance of these topics, addressing them will be presented in a separate manuscript.

-       The authors do not mention in the manuscript if GriMeDB has the potential to be a living data base with a call to scientist to inform and/or deposit new (self-curated) data and to keep this data set growing.

The current GriMeDB contains I believe the majority of studies published with CH4 data in fluvial systems, but this list is not exhaustive, especially with missing recent published works (some in spotted under sampled regions), such as:

Canning et al., 2021, https://doi.org/10.5194/bg-18-3961-2021
Castro-Morales et al., 2022, https://doi.org/10.5194/bg-19-5059-2022
Patel et al., 2022, https://doi.org/10.1016/j.watres.2022.119380 Wesley
et al., 2022 (still work on discussion, unpublished),
https://doi.org/10.5194/egusphere-2022-549
Zhao et al., 2022, https://doi.org/10.1016/j.envpol.2021.118769 Zhu
et al., 2022, https://doi.org/10.1038/s41467-022-31559-y

Maybe some of these studies, do not meet the requirements for GriMeDB, but it is just an example of current works that could be potentially added to the data base.

We thank the reviewer for pointing us toward these references. We were familiar with all but one of them; however, most appeared or became known to us after we had submitted our data package to EDI for publication and thus were not included. We needed to stop entering data at some point in time! But we have been amassing these and other recent relevant papers for a later update and they will definitely be part of this update (discussed below).

Additionally, and as part of my comment, can the authors provide some sort of protocol for sampling and curation of future or current available data set that are not yet part of GriMeDB and can help the authors to include and expand GriMeDB with community contributions. An example of such living data base is SOCAT (https://www.socat.info/), which provides clear protocols for scientists that wish to contribute with their data sets to this growing data base. This might not be on the scope of the authors at this stage, but given that they host the method it would be interesting if a call to the community can be done on the interest of keeping GriMeDB growing with a certain quality control.

Even if there is no current perspective by the authors to include more data sets to the current data base, an added short protocol as part of this manuscript would be extremely useful. Such protocol can include, e.g., parameters to measure and report (e.g., concentration in certain units), method used to determine $k$, measurement or information of other parameters (e.g., channel type following Strahler scale, channel slope), in a way that it can facilitate future additions of new data sets to GriMeDB.

Already the authors provide certain general recommendations (e.g., determine and report detection limits and include samples falling below these limits, include information on habitat conditions, studies expanding temporal dimensions are encouraged, routine $CH_4$ measurements with as part of water quality monitoring programs are encouraged), that can also be included and summarized more clearly in a protocol, with potential of including recommendations for e.g., $k$ parameterization to be used. If the authors cannot provide a protocol at this stage, I encourage them to add a section 4.3. in the discussion section with "Recommendations for future studies".

This comment about the future of GRiMeDB has been very helpful and triggered substantial reflection about how to manage this resource from this point on. It is our hope to make this a living database, at least for now, so we added text to the conclusion regarding our intent to provide future updates. We have also taken the step of generating a data submission form based on the current structure of GRiMeDB and a 'cookbook' (*sensu* SOCAT) describing the fields, defining fields required for data to be included, how to enter data, identifying preferred units, and how to submit the information. As is stated in the manuscript, this information is now available at stanley.limnology.wisc.edu.grimedb.

We also added some suggestions for core data that would be most helpful for future analyses to the Conclusion section. I do not think we are at the stage of developing a more formal sampling protocol yet, and believe that doing so would be best done with broader input from the research community (a discussion we hope to have in the future). But we hope that these new additions inspired by the reviewer's comments are steps in the right direction.

**Minor comments:**

P4,
L4 – How the authors assessed the quality control in unpublished data sets?
There are 7 listed unpublished datasets, 5 of which were provided by authors of this paper, and 2 were provided by colleagues we know and trust. Thus, there was no formal QA/QC process beyond making sure the data made sense during the data entry process.

L84 – EU Zenodo is missing
Zenodo omission corrected.

P7,

L141 - it is better at this stage to mention that R package was used and not until section 2.5

We added in the citation to R here, but also left further details about other packages in section 2.5.

L159 - It is missing the code NORM in Table 1, it is only mentioned in the caption of Fig. 14

We have added the 'NORM' category to Table 1.

P10,

L222 - it is necessary to add explicitly the units of concentrations and fluxes in each column, where it corresponds, in Tables A3 and A4, otherwise it is only possible to visualize the units by accessing the headers at the tables directly.

Units have been added to rows defining concentrations and fluxes in Tables A3 and A4.

P21, L371 - there is a dot instead of a comma after the parenthesis

Fixed

P22,

L396 – I believe the words "between $CH_4$ physical site attributes" need to be removed, so the sentence can only read "As with relationships between $CH_4$ concentration of flux and water chemistry parameters … "

This sentence is now: "As with relationships between $CH_4$ and physical site attributes…"

L398 – Did the authors try to calculate correlations of Figs. 12 and 13 per latitude bands? They can be biased due to density of observations but at least some meaningful correlations might be seen between the selected parameters and methane.

We only reported regression results for unbinned $CH_4$ concentrations/fluxes vs latitude (and basin area; results in Table S3) and hadn't included results from a similar binned analysis given the weak relationship revealed in Fig. 12. Following this comment, we re-analyzed the data after binning (using bins shown in Fig. 8) and again did not find evidence of a consistent relationship between $CH_4$ and latitude.

L401 – refer here to Fig. 13a L403
– refer here to Fig. 13b

Done

P23, L408 – define here IMP as (impounded reaches), as all the other listed channel types were defined in this paragraph, the same for TH (thermogenic $CH_4$ inputs) in L415.

Done

P25, Fig. 14 – These site-averaged concentrations need to be normalized to sample size so variations can be reduced due to the varying sizes and a better comparison between channel types can be done.

We used the non-parametric Kruskal-Wallis test because it is robust in cases of unequal sample size. And in general, the intent of this and other figures is to be exploratory and suggestive of what can be done with the data rather than providing rigorous analyses to identify drivers/predictors. However, we did re-run these tests after dropping poorly-represented sites (those with <10 observations) to reduce substantial differences in sample size. This did not substantively change the test outcomes.

P29, L551 – "compared **TO** our previous efforts"

Done

P30, L610 – additionally data assimilation models will strongly benefit from the GriMeDB database
Suggestion incorporated

P31, L628 – "… the expansion of GHG data **FOR** world streams and rivers …"
Done

Hall, R.O. Jr. and A.J. Ulseth. 2020. Gas exchange in streams and rivers. Wiley Interdisciplinary Reviews: Water 7 (1), e1391.

Raymond, P. A., et al. 2012. Scaling the gas transfer velocity and hydraulic geometry in streams and small rivers. Limnology and Oceanography: Fluids & Environments, 2, 41–53.

---

## Author Comment (AC3)

**Comment on essd-2022-346**

Bridget Deemer (Referee)
* * *
Referee comment on "GRiMeDB: The global river database of methane concentrations and fluxes" by Emily H. Stanley et al., Earth Syst. Sci. Data Discuss., https://doi.org/10.5194/essd-2022-346-RC3, 2023
* * *
This data paper substantially updates and expands upon a previous fluvial methane database (MethDB) from a similar group of authors that was published in 2016 (Stanley et al. 2016). It will be an important resource for the aquatic biogeochemistry community in general. The paper identifies key gaps in the existing spatial representativeness of stream and river methane emission data (missing data from arid, high altitude, and arctic regions) and also highlights the lack of long time series methane flux data in fluvial ecosystems. The dataset is structured in a unique way that allows users to explore spatio-temporal variation. Instead of reporting mean emissions for a given system, the database provides within-system site and date specific emissions (and summary statistics across the day when relevant). The dataset will be a key resource for those interested in upscaling aquatic greenhouse gas emissions.

The paper contains helpful visualizations and tables that orient the reader to key aspects of the dataset. One useful addition the authors could consider adding is a database schema diagram that shows how unique IDs can be used to link the four.csv files (concentrations, fluxes, sites, and source). This type of diagram could provide a visual guide for a data user describing which connections are "one to one" versus "one to many". For example, on first look at the data files, I found myself a bit confused as to how I would link the flux data to the concentration data. Is there always a "one to one" link where each row of flux data has a matching row of data in the "concentration" data file?

We created a new figure (Figure 1) that provides an overview of the DB's structure and linkages between the tables to illustrate/explain when a row in the flux table did or did not have a matching row in the concentration table.

Another novel aspect of this database is the inclusion of both diffusive and ebullitive methane emissions and the comprehensive annotation of specific methodological approaches (where the previous MethDB database only reported diffusive emissions). I think this aspect of the dataset should be mentioned in the abstract.
Done

I also think it would be interesting to visualize and/or further describe differences in the emissions that are generated via different methods. There is already some discussion of the potential importance of ebullition in overall fluvial methane flux (lines 522-523 citing certain papers), but the authors do not report any statistics regarding the fraction of emissions that are ebullitive in their own dataset (for sites with independent estimates of both flux pathway). In my work with reservoir methane emissions, the potential predictors of emission became clearer when we stopped combining diffusive-only emission estimates with those that integrated across both flux pathways (Deemer et al. 2016). I think this division may be a bit harder to make in the river literature (since it may be harder to discern which estimates were truly diffusive-only), but I think some basic summary of the flux data by method would be helpful. In looking at the data, it looks like you only have 8 rows of data with total methane flux recorded. You do mention that 85% of the data is diffusive-only, but I'm surprised there are only 8 rows that have estimates of both diffusive and ebullitive emission. This could be explicitly called out.

Investigating the relationship between diffusive and ebullitive fluxes, including possible methodological effects is a great suggestion. However, we have elected not to pursue this suggestion in this paper because we are doing so in a separate effort that has a more in-depth consideration of $CH_4$ fluxes. However, we will note that given the reality that fluxes (total or diffusive) span 6 orders of magnitude, being able to see methods-driven differences at this spatial scale may be difficult.

With respect to having 8 rows (observations with both ebullition and diffusion data), this is slightly off. There are 64 observations in which ebullition and diffusion were measured simultaneously and independently. We suspect that the 8 observations noted in this comment are observations reporting ebullition and diffusion, but not total flux.

Thank you to the authors for this important contribution to the field.

We appreciate your generous words!

Line by Line Comments

Line 35- If you can fit it, I suggest including the ranges and/or standard deviations here
Ranges and SDs were added to the abstract.

Lines 88-95- I assume beaver ponds were not included as "marginal" fluvial systems, but you might explicitly mention this here. Also, what about river reaches upstream of weirs?
We added 'beaver ponds and immediately upstream of small dams' (weirs) to the list of excluded sites.

Line 131-132- Figure 2 doesn't really make this distinction regarding sites that were used in multiple studies. Consider either adding this or annotating it somehow directly in figure legend.
The first step in the process of entering sites into the Site Table for us was to determine if a site being considered was already entered (i.e., it had been studied before in a separate article). This is represented by the first box in Fig. 2 ("Site used previously?"). Comments were in fact entered regarding a site's use across >1 paper, including identifying the other sources using the site for their additional data collections. So we have not made any adjustments to this figure.

Line 175- Consider changing the wording in this title (and/or in the text directly below it) to "Concentrations Table and Fluxes Table" to make extra clear that they are two separate tables.
Done

Line 183- Delete duplicate use of the word "both"
Done

Lines 221-228- So, is it true that in some cases the same concentration data might be applied to many rows of flux estimates (one to many)?
This relates to the earlier general comment about how the tables in GRiMe are linked, and hopefully the new Fig. 1 helps to clarify this situation. To respond to this specific question, no- it is not true that one concentration observation would be applied to many rows of flux estimates. If a site-date combination has concentration data with supporting water chemistry and also flux data, there will be 1 row in the Concentration Table and 1 row in the Flux Table for this site-date combination. If the site-date combination has flux and chemistry but no concentration, there will also be 1 row in the Flux Table and 1 row in the concentration table. In this case, the Concentration Table entry has the supporting water chemistry data, but does not contain $CH_4$ concentration data (since it doesn't exist). If a site-date combination has only flux data (no concentration, no supporting water chemistry), then there will be 1 row in the Flux Table and no corresponding row in the Concentration Table.

Lines 287-289- This pattern is also true for lake and reservoir methane data—65% of the lake/reservoir methane emission estimates in a recent dataset were collected since 2015

(Rosentreter et al. 2021, Deemer and Holgerson 2021)
Not surprising! Perhaps data increases reflect the growing use of portable GHG analyzers.

Lines 292-294- Wow!  I can't believe how short the longest flux record is—much shorter than the lake literature.
Yes- it's very surprising. We hope drawing attention to this brevity will inspire continued collection of records to generate longer time series.

Line 335- Do you mean 4% of the global river surface?  I don't think you mean land surface from looking at the map (more than 4% seems to be shaded darker tones of orange, but some of this is in surface-water poor areas like the Sahara).
Yes- thanks for catching this omission. Wording has been fixed.

Line 408- Include the definition for "IMP" like you do for the other site types.
Done. Also done for "TH" later in the paragraph.

Figure 12- The relationship between flux and total N & P looks stronger than for DOC or dissolved oxygen, but this isn't called out where you discuss drivers (lines 571-590) .  You might consider citing some of the wetland, lake and reservoir literature that has linked methane emission to productivity/chlorophyll a.
Text was added to this paragraph in the discussion about the relationships between TN, TP, and $CH_4$ emissions along with noting papers that have found similar positive relationships between $CH_4$ and eutrophication/nutrient enrichment.

Line 480- You could discuss insights on spatial/temporal resolution from the lake literature here.  Wik et al. 2016 Geophysical Research Letters showed that spatial and temporal under-representation generally led to underestimates of emission in lakes.
We are hesitant to suggest that limited temporal sampling is likely to lead to underestimated emissions in streams and rivers at this point, given the very different controls on gas concentrations and fluxes between lentic and lotic systems.

Line 509- Remove either "few" or "several"
Done

Line 522- It isn't clear if this 30-90% range comes from your entire database, or just from the few papers cited here.  Or maybe there are only three papers that quantify both pathways together?  In the lake and reservoir literature, the fraction of emissions that are ebullitive can range dramatically (undetectable to almost all of the emission), with ebullitive emission contributing a median of 78% of methane emissions in reservoirs and 54% in lakes- Deemer and Holgerson 2021).
The wording was edited to clarify that the 30-90% of total emissions was reported by the papers cited here.

Line 584- Rosentreter et al. 2021 also used latitude to upscale stream and river emissions.
Citation added

Line 599- I thought Burns et al. 2018 reported rather high methane emissions from glacial systems?
Although Burns et al. argue for very high emissions, outflow concentrations are actually below the global average.

Line 628- Add the word "from" between "data" and "world"
Done

Tables A3 and A4- I suggest explicitly clarifying that "new" units are the relevant units for the data you report.
These table describe the 'new units' as the current common units for all concentrations

---

## Author Response (AR2)

We are delighted to make these final updates to our manuscript.

Please address the following issues:
1. For the next revision, please add the citation (Stanley et al., 2022) to your DOI /10.6073/pasta/b7d1fba4f9a3e365c9861ac3b58b4a90 in the "Abstract" section of the *.pdf manuscript file.

Author Response:
We have added the doi to the abstract section. PLEASE NOTE- the dataset DOI was updated during the revision process and is now Stanley et al. 2023. The prior DOI is still functional and will lead to version 1 of GRiMeDB and will include a link to get to version 2 (which is now cited in the abstract). The citation has also been updated throughout the paper.

Author Response
2. Your reference list includes works "in review". Such works can be cited upon submission if being available to the reviewers. They should not be cited in the final, accepted manuscript, unless published, accepted for publication, or available as preprint with a DOI.

The citation to this paper has been updated to "accepted."